# The hybrid lipoplex induces cytoskeletal rearrangement via autophagy/RhoA signaling pathway for enhanced anticancer gene therapy

Xueyi Hu [1,3], Yichun Wang[1,3], Ruohan Wang[1], Yiyao Pu[1], Rongrong Jin[1] ✉, Yu Nie [1] ✉ & Xintao Shuai [2]

Delivering plasmid DNA (pDNA) to solid tumors remains a significant challenge due to the requirement for multiple transport steps and the need to promote delivery efficiency. Herein, we present a virus-mimicking hybrid lipoplex, composed of an arginine-rich cationic lipid, hyaluronic acid derivatives coated gold nanoparticles, and pDNA. This system induces cytoskeletal rearrangements through "outside-in" mechanical and "inside-out" biochemical signaling, overcoming intra- and intercellular barriers to enhance pDNA delivery. By modulating autophagy, RhoA signaling, and cytoskeletal dynamics, we achieve a 20-fold increase in gene expression with high tissue specificity in solid tumors. Furthermore, the system is applied to co-deliver a p53 plasmid and an MDM2 inhibitor, demonstrating significant synergistic antitumor effects in hepatocellular and lung carcinomas.

Gene therapy has gained increasing attention for the prevention and treatment of genetic disorders, refractory diseases, and widespread infectious diseases, emerging as one of the most promising strategies in biological therapies[1]. The rapid deployment of mRNA vaccines during the COVID-19 pandemic, administered to millions of healthy individuals, has made a significant contribution to public health[2,3]. Additionally, 40 gene therapy products have been approved, with over 4,000 human clinical trials currently underway worldwide[4]. Among nucleic acid-based therapeutic strategies, plasmid DNA (pDNA)-based therapies offer unique advantages, including more sustained transgene expression[5,6], greater stability, and lower production costs[7,8]. However, intracellular delivery of pDNA is more challenging than other nucleic acid cargos, as it requires additional steps for nuclear import to enable successful transcription.

Lipid nanoparticles (LNPs) are the only approved nonviral carriers currently used in clinical trials and on the market due to their favorable biocompatibility and high efficiency delivery in mRNA and siRNA delivery. However, none of the lipid-based pDNA delivery systems for cancer treatment has advanced to late-stage clinical trials. This is largely due to the dense extracellular matrix and high interstitial fluid pressure in solid tumors, which result in low cellular transfection efficiency in vivo[6]. To improve pDNA transfection both in vitro and in vivo, researchers have devoted much effort to every extra- and intracellular delivery step, from the beginning of cellular recognition/endocytosis[9], endosome/lysosome escape[10,11], and gene release, to final nuclear import[12]. Similarly, our group has developed a series of nanoparticles that mimic viral nanosized core-shell structures[13], enabling cellular targeting[14], membrane penetration[15,16], and hierarchical stimuli-triggered transformations[17]—all of which promote gene expression. Despite these advantages, a significant gap remains compared to viral carriers, which are highly efficient due to their intricate natural mechanisms.

Besides stimuli-triggered transformations, viruses leverage the host cell's cytoskeleton system (microtubules and actin filaments) as an

[1]National Engineering Research Center for Biomaterials, College of Biomedical Engineering, Sichuan University, Chengdu 610064, P. R. China. [2]Nanomedicine Research Center, The Third Affiliated Hospital of Sun Yat-sen University, Guangzhou 510630, P. R. China. [3]These authors contributed equally: Xueyi Hu, Yichun Wang. ✉e-mail: jinrr2015@scu.edu.cn; nie_yu@scu.edu.cn

"express elevator" to transport their genetic material throughout the infection process[18]. For example, adenoviruses rely on actin filament assembly for endocytosis and use microtubules and kinesin complexes to transport their genome near the nucleus[19,20]. Similarly, baculovirus uses actin filaments as a conduit to deliver their genome to the nucleus[21]. In contrast, inhibition of microtubules or actin filaments has been shown to significantly reduce transfection efficiency by impairing uptake[22] and preventing gene vectors from targeting subcellular organelles or delivering genes to the perinuclear region[21,23]. Moreover, after viral replication and assembly within the host cell, viruses use actin filaments during exocytosis to facilitate their release (e.g., coronaviruses)[18,24], prompting intercellular delivery and further infection.

Recent studies have highlighted the susceptibility of the cytoskeleton to various nanomaterials with different mechanical properties and the initiation of specific biochemical signals triggered by these materials. For instance, variations in nanomaterial rigidity can regulate phagocytosis and cancer cell uptake, both of which are closely related to cytoskeletal dynamics[25,26]. In terms of biochemical signaling, silica nanoparticles have been shown to disrupt tight junctions and alter cytoskeletal arrangements in a blood-brain barrier model, involving oxidative stress and Rho-kinase/JNK signaling pathways[27]. Additionally, nanoparticles such as $TiO_2$, $SiO_2$, and polystyrene induce actomyosin contraction by triggering intracellular calcium elevation via myosin light chain kinase[28]. These insights led us to explore whether LNPs could similarly manipulate the cytoskeleton, like viruses, to transport genetic material and achieve higher pDNA transfection efficiency.

In this work, we hybridize arginine-rich lipids (RLS) with hyaluronic acid derivatives coated gold nanoparticles (HS@Au) to investigate whether gene delivery efficiency can be enhanced by modifying the mechanical properties of the vectors and influencing biochemical signaling. Building on the hybrid lipoplex (RLS/HS@Au/DNA), which demonstrates high transfection efficacy both in vitro and in vivo, we conduct an in-depth exploration of the underlying mechanisms step by step, including endocytosis, nuclear translocation, exocytosis, transcytosis, tumor penetration, and accumulation (Fig. 1). Finally, the enhanced gene transfection is applied to tumor therapy targeting the p53/MDM2 pathway by co-delivering the p53 plasmid and the MDM2 inhibitor SP141.

## Results

### Preparation and characterization of the hybrid lipoplex

Gold nanoparticles (AuNPs) were incorporated into the RLS lipoplex (Supplementary Fig. 1) due to their "harder" properties, which may help investigate the role of cytoskeleton regulation in transfection. Previous studies demonstrated the potential of AuNPs to enhance pDNA delivery efficiency, though the exact mechanism remains unclear[29]. To improve cellular uptake and ensure biocompatibility, a hyaluronic acid coating was applied (Fig. 2a). We compared an in-situ reduction method for preparing AuNPs using thiol-conjugated hyaluronic acid (HS, Supplementary Fig. 2) as the reductant with the traditional small-molecule reduction-ligand exchange method. The in situ prepared HS@Au showed a similar AuNPs morphology and size;

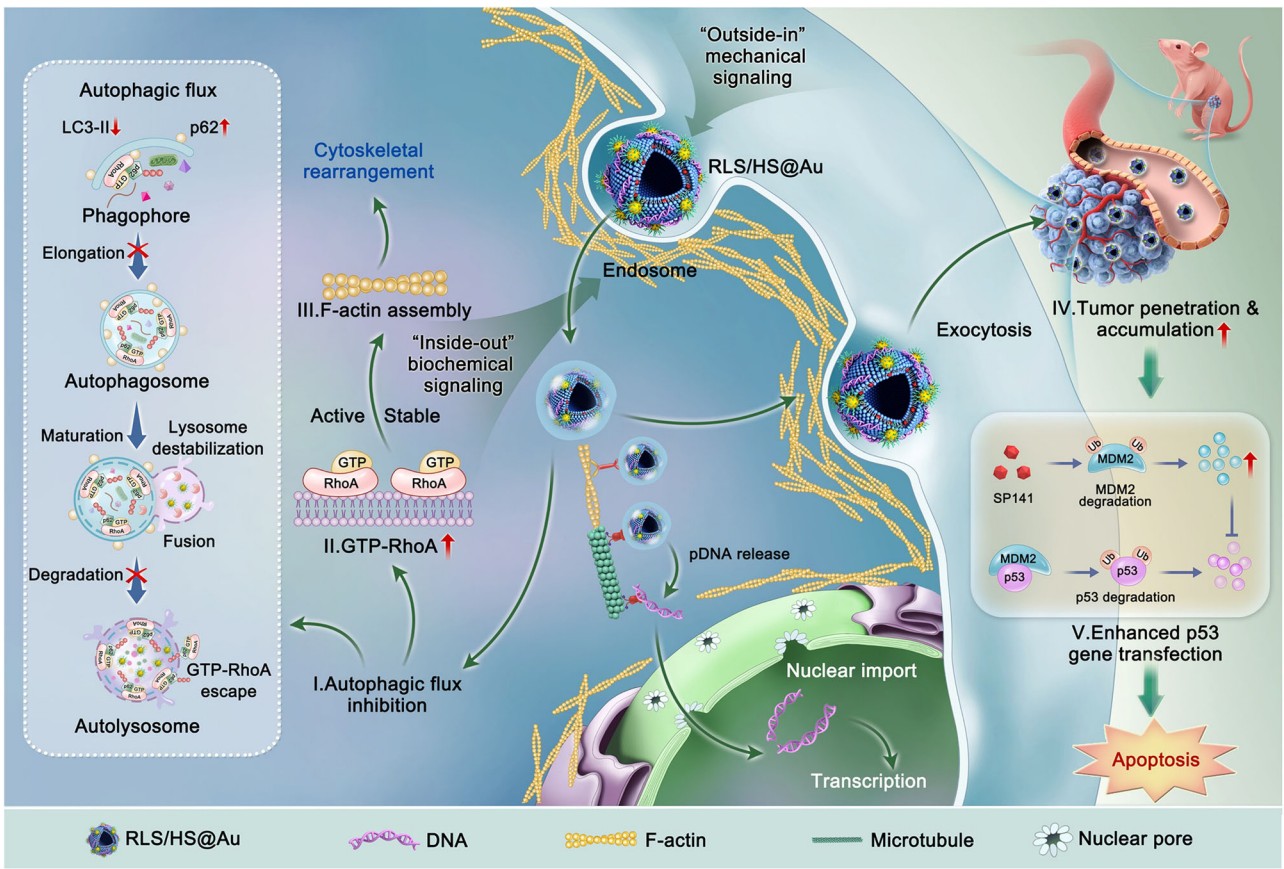

**Fig. 1 | Schematic illustration of facilitated gene transfection by the hybrid lipoplex with therapeutic mechanism exploration.** The hybrid lipoplex enhances pDNA delivery through "outside-in" mechanical signaling (via the "harder" HS@Au) and "inside-out" biochemical signaling (through autophagy/RhoA pathways), inducing cytoskeletal rearrangement and promoting tumor inhibition by targeting the p53-MDM2 pathway. Once endocytosed by cells, the hybrid lipoplex (**I**) inhibits autophagic flux, reducing autophagosome formation and impairing lysosomal degradation of GTP-RhoA. This result in (**II**) increases membrane-associated GTP-RhoA and (**III**) enhances actin polymerization into F-actin, driving cytoskeletal rearrangement. The rearranged cytoskeleton facilitates transcytosis (**IV**), improving tumor penetration and accumulation of the hybrid lipoplex. Finally, (**V**) cytoskeletal rearrangement promotes p53 gene expression, and the co-loaded p53/SP141 hybrid lipoplex synergistically induces tumor apoptosis by targeting the p53-MDM2 pathway.

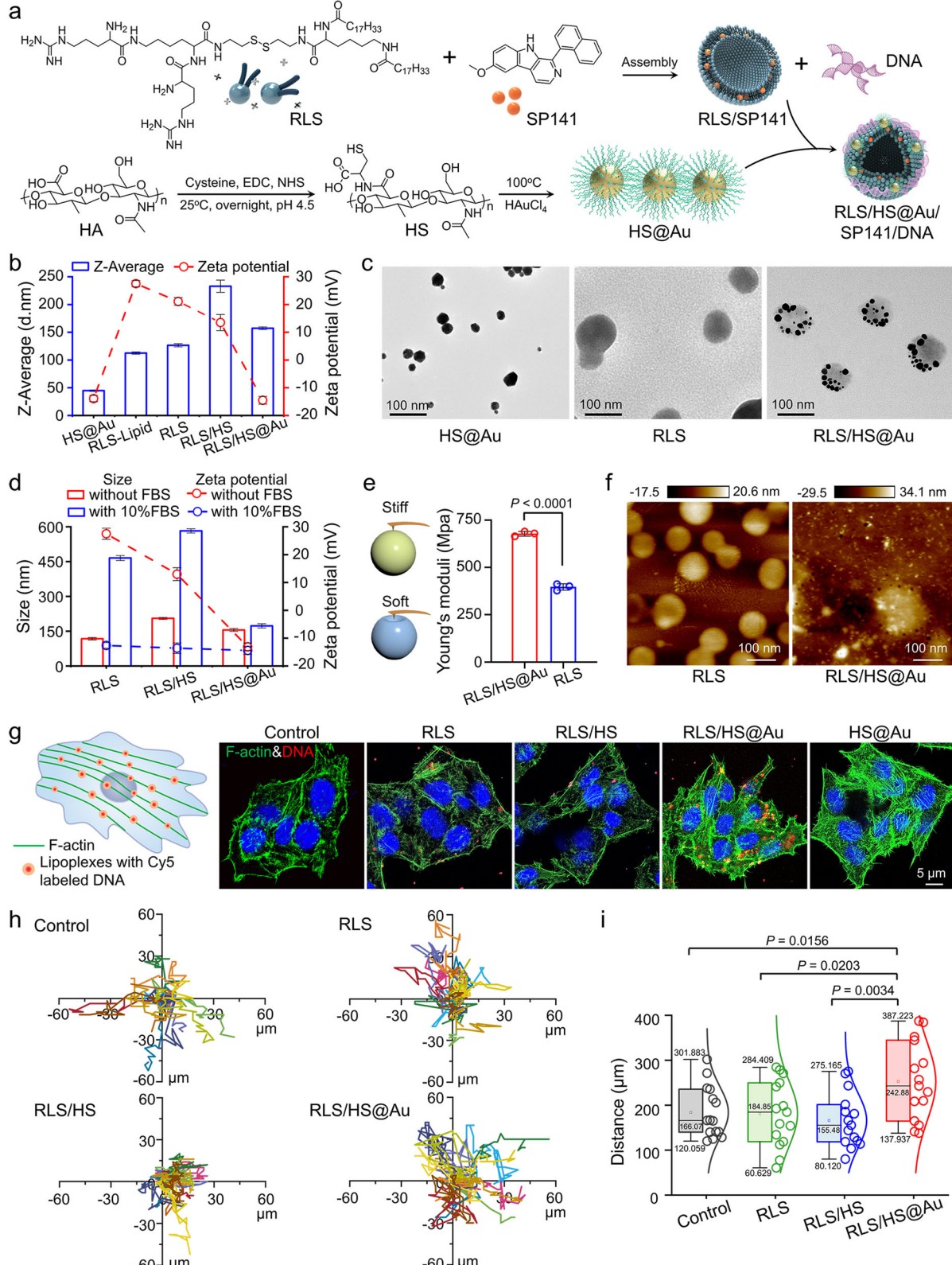

however, it contained significantly more hyaluronic acid (HA, 57.5%) compared to the traditional ligand exchange method (HS@Au, 35.4%) (Fig. 2b, c, and Supplementary Figs. 3–5).

Subsequently, the hybrid lipoplexes were formed by combining HS@Au with the RLS lipoplex (containing pDNA), resulting in nanosized particles (~160.1 nm) with a negative charge (−15.9 mV) and a uniform spherical morphology (Fig. 2b, c). The

incorporation of HS@Au improved the lipoplex's resistance to nuclease degradation without affecting DNA condensation or release at the optimized N/P ratio of 20 and an Au/DNA mass ratio of 2.4 (Supplementary Figs. 6, 7). No significant changes in particle size were observed when RLS/HS@Au was placed in culture media (10% FBS), PBS buffer (pH 7.4), or 10% plasma (Supplementary Fig. 8). The average size remained stable over one month

**Fig. 2 | Preparation, physicochemical characterization, and cytoskeletal remodeling function of the hybrid lipoplex (RLS/HS@Au). a** Preparation steps for HS, HS@Au, and RLS/HS@Au. **b** Size distribution and ζ potential of HS@Au, RLS assemblies, and various lipoplexes, as detected by dynamic light scattering (DLS). $n = 3$ independent experimental units. **c** The representative transmission electron microscopy (TEM) image of HS@Au and various lipoplexes. The scale bar is 100 nm. $n = 3$ independent experimental units with similar results. **d** Changes in particle size and ζ potential of different lipoplexes in culture medium with or without 10% FBS for 24 h. $n = 3$ independent experimental units. **e** Young's modulus and **f** representative morphology images of RLS and RLS/HS@Au with distinctive stiffness measured by atomic force microscope (AFM). $n = 3$ independent experimental units with similar results. The scale bar is 100 nm. **g** Representative fluorescence images of the lipoplexes distributed along F-actin in well-spread cells after

4 h of transfection observed by confocal laser scanning microscope (CLSM). (Blue: DAPI stained nucleus, Green: FITC-phalloidin stained F-actin, Red: Cy5-labeled DNA). $n = 3$ independent experimental cell lines. The scale bar is 5 μm. **h** Typical trajectories of HepG2 cells after treatment with various lipoplexes. Different colored lines represent the motion trajectories of 15 randomly selected cells within 24 h. **i** Migration distances of HepG2 cells with different lipoplexes treatments, calculated from Fig. 2h. $n = 15$ cells migration distances analyzed from 3 independent experimental cell lines. Boxplots show the distribution of expression with the center of the box representing the mean, the center line corresponding to the median, and upper and lower bounds representing 75% and 25% percentiles. Two-sided unpaired Student's t test was used for the comparisons in (**e**) and (**i**). The data in (**b**, **d**, **e**, and **i**) are mean ± SD. $p$ values < 0.05 were considered statistically significant. Source data are provided as a Source Data file.

(Supplementary Fig. 9), indicating the good stability of RLS/HS@Au. Notably, atomic force microscopy (AFM) analysis (Fig. 2e, f) revealed that RLS/HS@Au possessed a higher Young's modulus (~678.3 MPa) compared to RLS alone (~397.0 MPa), confirming the successful preparation of a stiffer lipoplex through the incorporation of inorganic nanoparticles.

### Cytoskeletal remodeling by the hybrid lipoplex

To confirm whether the soft and hard lipoplexes could induce distinct cytoskeletal rearrangements, HepG2 cells were stained with phalloidin after being exposed to various lipoplexes containing Cy5-labeled DNA (Cy5-DNA) for 4 h (Fig. 2g). Both RLS/HS@Au and HS@Au lipoplexes induced the most pronounced actin filament polymerization compared to the unhybridized RLS and RLS/HS lipoplexes (Fig. 2g and Supplementary Fig. 10a, b). The Cy5 signal was largely colocalized with the actin filaments, indicating that the intracellular transport of lipoplexes may depend on the cytoskeletal delivery system. Additionally, as actin filaments play a critical role in pDNA nuclear import[30,31], the amount of perinuclear actin cap was quantitatively analyzed, revealing a significant increase following treatment with RLS/HS@Au and HS@Au (Supplementary Fig. 10c).

Following this, the cell migration behavior primarily governed by the cytoskeleton, was investigated using a live-cell imaging system. Analysis of the cell trajectories of 15 randomly selected cells revealed that the hybrid lipoplex (RLS/HS@Au) induced more extensive cellular movement compared to the softer RLS and RLS/HS lipoplexes with enhancements of 26% and 58% in the average migration distance, respectively (Fig. 2h, i). Collectively, these findings suggest that the incorporation of inorganic nanoparticles into the lipoplex harnesses "outside" mechanical signals, leading to subsequent "inside" cytoskeletal remodeling.

### Promoted gene transfection by the hybrid lipoplex

To investigate whether hybridization could enhance gene delivery efficiency, we assessed the expression of the enhanced green fluorescent protein (EGFP) reporter gene (Fig. 3a and Supplementary Fig. 11). In the presence of HS@Au, RLS consistently demonstrated high transfection efficiencies under 10% FBS conditions (at an N/P ratio of 20), regardless of whether tumor cells (such as HepG2, HeLa, and 4T1) or non-tumor cells (NIH-3T3) were used. This observation indicates a universal phenomenon that is independent of cell type. Notably, transfection efficiency increased with the amount of AuNPs added in HepG2 cells, with the highest EGFP expression observed at an Au/DNA mass ratio of 2.4—approximately 6.5 times higher than that of RLS/HS (Fig. 3b). The optimized RLS/HS@Au lipoplex showed a 12-fold increase in transfection efficiency compared to the commercial Lipofectamine 2000. The advantages of hybridization persisted even when serum concentration increased to 20% and 50% (Supplementary Fig. 12), and the complexes demonstrated acceptable cytocompatibility (Supplementary Figs. 13, 14).

Subsequently, we investigated the intracellular delivery processes of different lipoplexes using confocal laser scanning microscopy (CLSM), focusing on cellular uptake, interaction with endo/lysosomes, and nuclear import. The cells were incubated with different lipoplexes containing Cy5-DNA (red) for 6 h, followed by staining for the cell membrane (green) and nucleus (blue). The hybridized lipoplex showed a mean fluorescent intensity (MFI) that was 3- and 5-fold higher than that of the RLS and RLS/HS lipoplexes after just 1 h of incubation with HepG2 cells (Fig. 3c, d), indicating a more rapid cellular uptake.

Notably, the Cy5 signal in the RLS/HS@Au group consistently increased over the 6-hour period, whereas the signals in the RLS and RLS/HS-treated cells peaked at 2 h and gradually declined thereafter. Typically, the fluorescent signal from lipoplexes decreases upon delivery into lysosomes, which are responsible for degrading foreign materials. This decline highlights the lysosome as a significant barrier to gene transfection. However, the hybrid lipoplex retained the strongest Cy5 signal after 6 h, likely due to hybridization altering the degradation behavior of lipoplexes in lysosomes.

To confirm our hypothesis and investigate the interaction between the lipoplex and lysosomes, we used LysoTracker (green) to monitor the quantity and acidic conditions of lysosomes, conducting co-localization analysis with Cy5-DNA (red) (Fig. 3e, f). Notably, the RLS/HS@Au group exhibited more intense and abundant LysoTracker signals compared to the other two groups during the initial incubation period (2 to 4 h), reflecting greater cellular uptake in the hybridization group, as described earlier (Fig. 3c). Importantly, the co-localization signal of DNA and lysosomes (indicated by yellow fluorescence) and its Pearson's correlation value gradually increased at 4 h post-treatment with the hybrid lipoplex, followed by a significant decrease at 6 h. In contrast, the RLS and RLS/HS groups showed a consistent increase in yellow fluorescent spots until the 6-hour mark. This difference confirms that hybridization accelerates the departure of lipoplexes from lysosomes. This effect may be attributed to the impaired acidic degradation function of lysosomes, as evidenced by the decreased green fluorescent LysoTracker signal in RLS/HS@Au-treated cells from 4 to 6 h, a trend not observed in the nonhybrid RLS and RLS/HS groups (Fig. 3e). These results align with our earlier speculation and provide further evidence that hybridization enhances the escape of lipoplexes from lysosomes.

Next, we compared nuclear import in cells treated with different lipoplexes, which represents the final step in pDNA delivery. The RLS/HS@Au lipoplex consistently demonstrated the shortest distance to the nucleus compared to the other two groups (Fig. 3c, g) throughout the observation period (1 to 6 h). To eliminate the possibilities of pseudolocalization from the 2D scan in CLSM, a 3D scan was conducted. This confirmed that nearly 30% of the Cy5-labeled DNA delivered by RLS/HS@Au had entered the nucleus at 6 h, whereas only 11% and 8% were observed for RLS and RLS/HS, respectively (Fig. 3h, i). These results indicate a faster intracellular transport of the hybrid lipoplex, enhancing the potential for transfection within the nucleus.

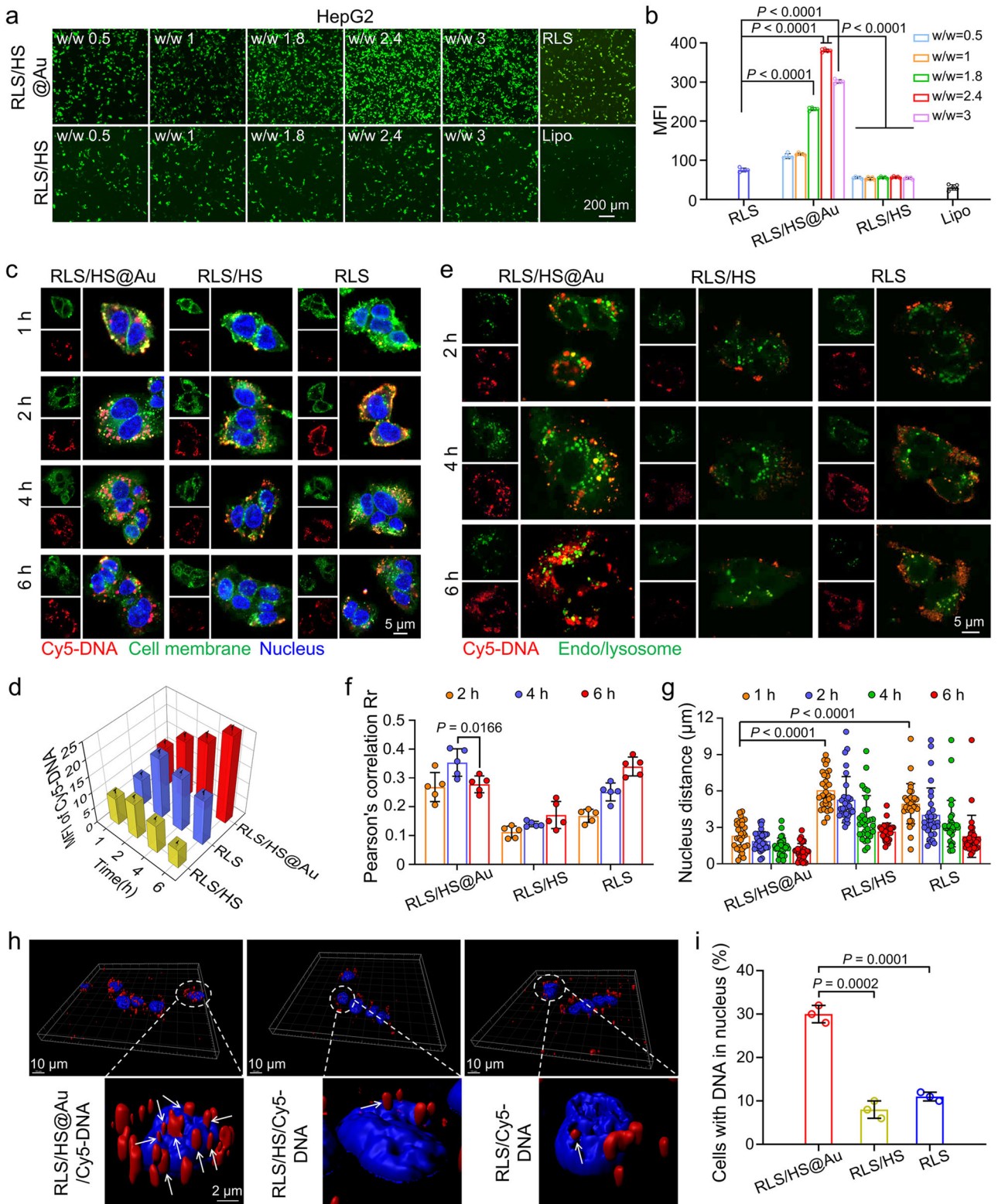

**Cy5-DNA Cell membrane Nucleus** (panel c)

**Cy5-DNA Endo/lysosome** (panel e)

## The hybrid lipoplex disturbed GTP-RhoA turnover in the autophagy flux

Several reports have confirmed that the transport of subcellular vesicles is highly dependent on the cytoskeleton[22,31], starting from the encapsulation of nanoparticles or viruses at the cell membrane, through endosomes/lysosomes, and finally translocating into the nucleus. Given the promising results in pDNA transfection and the effective intracellular delivery processes (including enhanced cellular

uptake, accelerated lysosome escape, and increased nuclear delivery) associated with the hybrid lipoplex, we speculated that these outcomes are linked to cytoskeletal remodeling. Thus, we aimed to elucidate the correlation and mechanisms between the pDNA delivery process and cytoskeletal remodeling mediated by the hybrid HS@Au lipoplex.

The Rho GTPase family (RhoA, Rac, and Cdc42) plays crucial roles in dynamically regulating cytoskeletal polymerization and

**Fig. 3 | Enhanced gene transfection and intracellular fate of hybrid lipoplex.** **a** Representative fluorescence images of pEGFP transfection on HepG2 cells (*n* = 3 independent experimental cell lines with similar results) and **b** semi-quantitative analysis of the mean fluorescence intensity (MFI) using ImageJ software, *n* = 5 fields from 3 independent experimental cell lines. The scale bar is 200 μm. **c** Representative images of cellular uptake for different lipoplexes in HepG2 cells after 1, 2, 4, and 6 h incubation (*n* = 3 independent experimental cell lines with similar results) and **d** semi-quantitation (MFI) calculated by the intensity of internalized Cy5-DNA, *n* = 5 fields from 3 independent experimental cell lines. Blue: Hoechst 33342 stained nucleus, Green: CellMask™ Green stained cell membranes, and Red: Cy5-labeled DNA. The scale bar is 5 μm. **e** Endo/lysosome escape of different lipoplexes observed by CLSM. (Red: Cy5-labeled DNA, Green: LysoTracker Green DND-26 stained endo/lysosome). *n* = 3 independent experimental cell lines with similar results. The scale bar is 5 μm. **f** Pearson's correlation value (Rr) reflected co-localization of Cy5 signal and lysotracker signal in **e**. *n* = 5 fields from 3 independent experimental cell lines. **g** The distance between Cy5-labeled DNA and the nucleus in HepG2 cells after incubation with different lipoplexes for 1, 2, 4, and 6 h. *n* = 30 dots of 3 fields from 3 independent experimental cell lines. **h** Nuclei delivery evaluation for the RLS, RLS/HS, and RLS/HS@Au after 6 h transfections. Representative 3D reconstruction images of lipoplexes-labeled HepG2 cells using the Imaris 9.0.1 software. White arrows represent Cy5-DNA entering the nucleus. *n* = 3 independent experimental cell lines with similar results. (Blue: Hoechst 33342 stained nucleus, Red: Cy5-labeled DNA). The scale bars are 10 μm and 2 μm, respectively. **i** Percentage of the cells with Cy5-DNA in the nucleus, calculated from 50 randomly selected cells in each group. *n* = 3 independent experimental cell lines. A two-sided unpaired Student's t-test was used for the comparisons in (**b**, **d**, **f**, **g**, **i**). The data are mean ± SD. *p* values < 0.05 were considered statistically significant. Source data are provided as a Source Data file.

depolymerization. Particularly, GTP-RhoA significantly influences vesicle endocytosis and intracellular transport by enhancing actin polymerization (Supplementary Fig. 15)[32,33]. We first assessed the expression level of GTP-RhoA using pull-down assays after treatment with different lipoplexes (Fig. 4a and Supplementary Fig. 16). As a result, RLS/HS@Au and free HS@Au significantly upregulated GTP-RhoA compared to the control and nonhybrid RLS groups. Furthermore, GTP-RhoA was found to be more concentrated on the plasma membrane of HepG2 cells in the hybridization groups, as demonstrated by ice-cold TCA fixation and immunostaining (Fig. 4b and Supplementary Fig. 17). These results indicate that hybridization promotes the accumulation of membrane-associated GTP-RhoA, which is responsible for actin polymerization[34].

To further investigate the correlation between cytoskeletal remodeling and pDNA transfection, we used the RhoA inhibitor C3 transferase. Treatment with C3 resulted in a significant decrease in EGFP expression efficiency: approximately 64% in the RLS/HS@Au group and 43% in the RLS group compared to the control group without C3 treatment (Fig. 4c, d, and Supplementary Fig. 18). This demonstrates that GTP-RhoA activity significantly influences DNA transfection, with the negative impact being more obvious in the RLS/HS@Au group. In other words, disrupting GTP-RhoA-mediated cytoskeletal remodeling markedly diminished the transfection advantage conferred by HS@Au hybridization.

Autophagy regulates the intracellular degradation of GTP-RhoA, preventing excessive polymerization of F-actin[34]. Notably, AuNPs can disrupt autophagic flux[35,36]. We hypothesized that the accumulation of GTP-RhoA in the RLS/HS@Au group may result from impaired autophagic flux. To investigate this, we used a tandem fluorescent-tagged LC3-EGFP-mCherry reporter plasmid. The mCherry fluorophore is more resistant to acidic pH than EGFP, allowing non-acidic autophagosomes to be labeled by both EGFP and mCherry, while acidic autolysosomes would show punctate red labeling due to EGFP quenching.

We observed that the number of yellow puncta (mCherry⁺EGFP⁺) in the RLS/HS@Au group was comparable to that in the RLS group under non-starvation conditions, but significantly decreased under starvation (Fig. 4e and Supplementary Fig. 19). Given that gold nanoparticles can alkalinize lysosomal pH, the yellow puncta likely included both autophagosomes and alkalinized autolysosomes. To confirm that, the colocalization of lysosomal marker LAMP1 with yellow puncta was detected (Fig. 4f). In the hybridization group (RLS/HS@Au), many bright white puncta were observed, indicating co-localization of LAMP1-positive (blue fluorescence) and LC3-GFP-mCherry-positive (yellow fluorescence) vesicles. In contrast, the yellow puncta in the RLS group rarely colocalized with the blue LAMP1 signal. These findings suggested that the yellow puncta in the RLS/HS@Au-treated cells predominantly represent alkalinized autolysosomes, while those in the RLS-treated cells mainly indicate autophagosomes.

To further characterize the disturbed autophagic flux, we conducted TEM observations and noted a decrease in autophagosomes (white arrows) after RLS/HS@Au or HS@Au treatment compared to untreated cells (Fig. 4g, Supplementary Figs. 20, 21). AuNPs were observed accumulating in endosomes (red arrow) and autolysosomes (yellow arrow), somehow illustrating their cellular transport mechanism (Fig. 4g).

Next, we measured the expression levels of LC3-II and p62, two well-known autophagy markers, with or without the autophagy inhibitors chloroquine (CQ) and NH₄Cl (Fig. 4h, Supplementary Figs. 22, 23). The hybridization induced LC3-II and p62 accumulation without inhibitors. Since CQ and NH₄Cl primarily inhibit autolysosomal proteolysis, we expected a higher accumulation of LC3-II and p62. However, in the presence of CQ or NH₄Cl, the addition of RLS/HS@Au resulted in lower LC3-II levels compared to controls, suggesting that RLS/HS@Au treatment initially reduced LC3-II generation, indicating decreased autophagosome formation (Fig. 4h and Supplementary Figs. 22, 23).

As an additional autophagy marker, p62, an autophagosome cargo protein, degrades in autolysosomes. In the presence of CQ or NH₄Cl, RLS/HS@Au also resulted in increased p62 accumulation compared to controls, indicating impaired autolysosomal proteolysis. Cells treated with only HS@Au further verified the impaired autophagic flux: as HS@Au concentration increased, LC3-II levels gradually decreased while p62 levels increased (Supplementary Fig. 24). Collectively, these findings demonstrate that RLS/HS@Au compromised autophagic flux by impairing early autophagosome formation, as evidenced by decreased LC3-II, increased p62, and reduced autophagosome formation.

Followingly, a self-quenched DQ™ Red BSA regent was applied to further detect the proteolysis activity in lysosomes because it could recover its high fluorescence after proteolysis in acidic compartments. Our results showed that RLS/HS@Au and free HS@Au induced less fluorescence signaling from DQ-BSA hydrolysis (Fig. 4i and Supplementary Fig. 25) compared to the RLS and RLS/HS groups, indicating weakened lysosomal degradation activity associated with HS@Au. The differences in fluorescence intensity between RLS/HS@Au and free HS@Au may be attributed to varying cellular uptake of AuNPs (Supplementary Fig. 26). These findings indicated that hybridization not only impairs early autophagosome formation but also affects late lysosomal proteolysis during the autophagic flux (Fig. 4j).

In summary, RLS/HS@Au transmits both "outside" mechanical signals (the stiffness of lipoplexes) and "inside" biochemical signals (the autophagy/RhoA axis) to cells, leading to cytoskeletal rearrangement and ultimately altering their intracellular fates.

## The hybridization elevated transcytosis and transportation of lipoplexes in vitro and in vivo

Cytoskeletal rearrangement can impact both intracellular transport and intercellular communication, known as transcytosis[18,37,38]. To

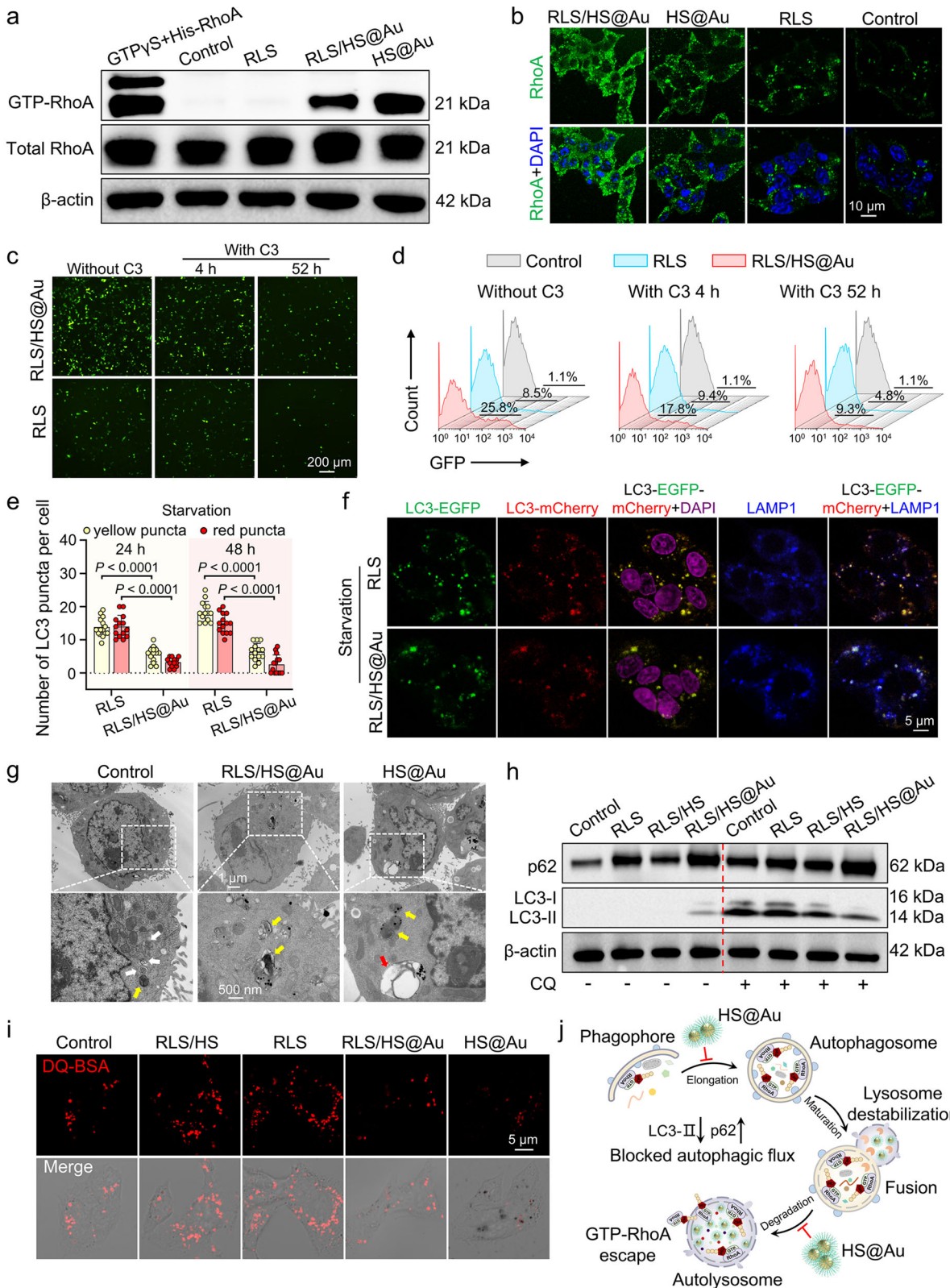

investigate this, we examined transcytosis between different cell populations. The first cell batch was treated with lipoplexes containing Cy5-labeled DNA or pEGFP, and the second batch was incubated with the culture medium supernatant from the first batch (Fig. 5a). As expected, RLS/HS@Au displayed the highest levels of intercellular delivery compared to non-hybridized groups (Fig. 5a, b). The second cell batch in the RLS/HS@Au group exhibited approximately twice the

Cy5 fluorescence intensity of the RLS and RLS/HS groups. Similarly, EGFP expression in the second cell batch for the RLS/HS@Au group showed approximately a 10-fold increase in MFI compared to the other two groups, likely due to enhanced exocytosis from the first cell batch (Fig. 5b).

Encouraged by the promising transcytosis results with the hybrid lipoplex, we extended the investigation to a 3D HepG2 tumor spheroid

**Fig. 4 | Exploration of mechanism for cytoskeletal remodeling by hybrid lipoplex. a** Pull-down assay of GTP-RhoA in HepG2 cells using RHOTEKIN binding. Cells were incubated with different lipoplexes or free HS@Au for 24 h. β-actin was a loading control, while GTPγS and His-RhoA acted as high-affinity positive controls. **b** Immunofluorescence of RhoA in cells after treatment with various lipoplexes or free HS@Au for 24 h. $n = 3$ independent experimental cell lines with similar results. Green: RhoA, Blue: DAPI. **c** Representative EGFP images and **d** flow cytometry analysis of HepG2 cells transfected with different lipoplexes, with or without C3 transferase for 48 h. $n = 3$ independent experimental cell lines. **e** Mean count of yellow and red puncta after LC3-EGFP-mCherry plasmid transfection for 24 and 48 h. $n = 14$ cells from 3 independent experimental cell lines. The data are mean ± SD. **f** Representative confocal images of cells transfected with various lipoplexes containing LC3-EGFP-mCherry for 24 h, followed by LAMP1 immunofluorescence staining. Channels: Green (LC3-EGFP), Red (LC3-mCherry), Blue (LAMP1), Violet (DAPI-labeled nuclei), Yellow (EGFP merged mCherry), and White (yellow merged

blue). $n = 3$ independent experimental cell lines with similar results.
**g** Representative TEM image of autophagic flux induced by RLS/HS@Au or free HS@Au on HepG2 cells. The red, white, and yellow arrows indicated endosomes containing Au nanoparticles, autophagosomes, and autolysosomes, respectively. $n = 25$ cells from 3 independent experimental cell lines with similar results.
**h** Western blot analysis of autophagy-related proteins in HepG2 cells treated with various lipoplexes with or without chloroquine for 24 h. β-actin was the loading control. **i** Lysosomal proteolytic activity analysis of HepG2 cells treated with various lipoplexes or HS@Au for 24 h. The activity was evaluated by the red fluorescence recovery of derivative-quenched bovine serum albumin (DQ-BSA). Serum-free MEM was the positive control. $n = 3$ independent experimental cell lines.
**j** Illustration of the GTP-RhoA accumulation due to autophagic flux suppression by the HS@Au. Two-sided unpaired Student's t test was used for the comparisons in (**e**). $p$ values < 0.05 were considered statistically significant. Source data are provided as a Source Data file.

model to assess its penetration effect. Tumor spheroids were incubated with various lipoplexes containing Cy5-DNA, followed by 3D scanning with CLSM. Notably, RLS/HS@Au demonstrated deep penetration throughout the entire spheroid, while RLS/HS and RLS were mostly restricted to the spheroid periphery. These findings suggest that RLS/HS@Au significantly enhances tumor spheroid penetration (Fig. 5c, d).

To determine if transcytosis-induced penetration could extend to solid tumors, we used a dorsal skinfold window chamber inserted into the skin of mice to construct a HepG2 tumor model (Fig. 5e). At 120 min after the intravenous injection of various Cy5-labeled lipoplexes, only the RLS/HS@Au lipoplex demonstrated significant extravasation from the blood vessels. In contrast, most of the Cy5-DNA compressed by RLS/HS remained within the blood vessels. Quantitative analysis of Cy5 fluorescence intensity revealed that RLS/HS@Au exhibited a stronger Cy5 signal than RLS/HS up to 200 μm from the vessel periphery into the tumor parenchyma (Fig. 5f). Therefore, upon delivery to xenografted solid tumors via blood circulation, the hybrid lipoplex not only enhanced intercellular penetration within tumor tissue but also facilitated extravasation from blood vessels. Transcytosis, which facilitated the extravasation of nanoparticles from blood vessels and deeper tumor infiltration, has been recognized as a key mechanism to overcome limited nanoparticle accumulation typically associated with the enhanced permeability and retention effect[39,40].

### The hybrid lipoplex enhanced specific gene expression in xenografted tumors

As verified above, the hybrid lipoplex demonstrated efficient cytoskeletal rearrangement, significantly improved intra- and intercellular DNA delivery, and enhanced tumor penetration. Encouraged by these findings, we further explored its potential for in vivo gene delivery. Tumor-bearing mice were administered varied lipoplexes containing pGL3 via intravenous injection, and luciferase expression levels were assessed using a live imaging system. Notably, the RLS/HS@Au/pGL3 lipoplex resulted in the highest luciferase expression at the tumor site, showing approximately 20-fold higher than the RLS/HS lipoplex, which yielded minimal expression. Additionally, the inclusion of HS@Au in the lipoplex increased gene expression specificity to the tumor tissue, achieving 78% localization with RLS/HS@Au treatment compared to only 17% with RLS/HS (Fig. 6a, b, e).

Meanwhile, observation of the Cy5-DNA fluorescent signal was used to monitor the distribution of lipoplex throughout the body. Unlike the localized luciferase expression at the tumor site, the Cy5 signal was broadly distributed. Notably, hybridization increased lipoplex accumulation in xenografted tumors, with a 7-fold elevation in Cy5 signal compared to RLS/HS (Fig. 6c, d, f). These results suggest that the high gene expression of the hybrid lipoplex in tumor tissues is

likely due to two main factors: increased accumulation within the tumor and enhanced gene transfection efficiency.

We observed that RLS/HS@Au/pGL3 elicited twice the luciferin signal of RLS/HS in the liver, while the Cy5 signal intensity from the RLS/HS@Au group was similar to that of RLS/HS (Fig. 6e, f). This indicates that both lipoplex formulations (RLS/HS@Au and RLS/HS) accumulate at similar levels in the liver, but the hybrid lipoplex achieves greater gene expression due to its resistance to lysosomal digestion by liver macrophages. Furthermore, few signals were detected in other organs (Fig. 6d), suggesting that the lipoplexes primarily undergo liver-dominated metabolism.

### Antitumor effect of the hybrid lipoplex targeting the p53/MDM2 pathway

Motivated by the specificity and high efficiency of this delivery system, we conducted a therapeutic experiment to validate its effectiveness by employing the p53 plasmid, the first gene approved for clinical use. However, p53 often demonstrates unsatisfactory antitumor effects in practice, as its activity is mainly inhibited by excessive expression of MDM2. Treatment strategies aimed at restoring p53 function by blocking the MDM2 axis with small-molecule inhibitors represent a promising approach for tumor therapy[41]. Therefore, we co-delivered the p53 plasmid and SP141 (an MDM2 degrader)[42] to enhance the antitumor activity of p53.

The RLS lipid could load SP141 during the self-assembly process and subsequently complex with DNA and HS@Au through electrostatic interactions. The co-delivery system (RLS/HS@Au/SP141/p53) displayed a spherical morphology and a uniform particle size at the nanoscale (approximately 165 nm) with a negative charge of -14 mV (Supplementary Fig. 27). The drug loading capacity (LC) and encapsulation efficiency (EE) of SP141 in the co-delivery system were 0.95% and 96.6%, respectively (Supplementary Fig. 28 and Supplementary Table 1). Notably, the drug loading did not affect DNA condensation or dithiothreitol (DTT)-triggered release capacity (Supplementary Fig. 6). The release rate of SP141 in PBS was slow, with a cumulative release of 33% at 48 h. However, when the co-delivery system was exposed to a buffer containing glutathione (GSH) for 48 h, the release of SP141 reached 89%, indicating a GSH-triggered release behavior similar to that of the p53 plasmid (Fig. 7a).

Next, the co-delivery effects on p53 expression were verified by western blotting (Fig. 7b, Supplementary Fig. 29). The MDM2 protein level decreased, while the p53 level significantly increased in the RLS/HS@Au/SP141/p53 compared to RLS/HS@Au/p53 without SP141. Additionally, the expression level of p21, a p53 target gene, was elevated, indicating p53-dependent cell cycle arrest. Immunofluorescence detection also showed upregulation of p53 and downregulation of MDM2 in HepG2 cells after treatment with RLS/HS@Au/SP141/p53 (Fig. 7c). Cytotoxicity data indicated that the synergistic antiproliferative effect

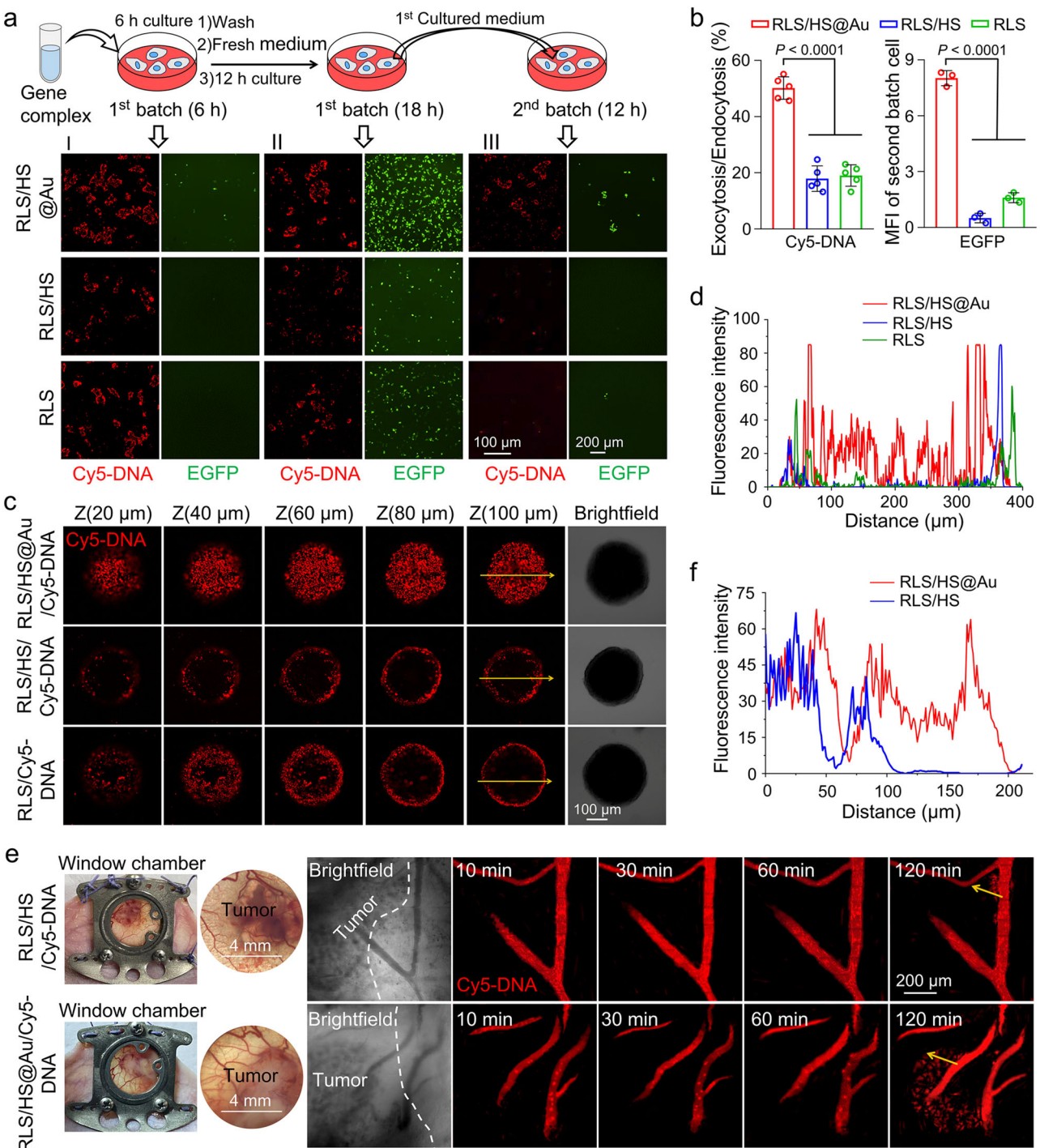

**Fig. 5 | Activation of the transcytosis and transportation of RLS/HS@Au. a** The intercellular transport of Cy5-labeled DNA and transfection of pEGFP by RLS/HS@Au via endocytosis-and-exocytosis. The procedure of the two experiments is similar: (I) Cells (the first batch) were incubated with various lipoplexes for 6 h and observed by CLSM. The culture medium was refreshed for another 12 h culture and harvested as condition medium; (II) Cells (the first batch) were imaged; (III) The condition medium was used to culture new cells (the second batch) for 12 h and observed by CLSM. (Red: Cy5-labeled DNA, Green: EGFP). n = 3 independent experimental cell lines with similar results. **b** Evaluation of exocytosis and endocytosis through the ratio of Cy5 MFI between the 2nd-batch and 1st-batch of cells (n = 5 fields from 3 independent experimental cell lines), and the EGFP MFI in the 2nd-batch of cells was calculated with ImageJ software (n = 3 fields from 3 independent experimental cell lines). The data are mean ± SD. **c** The penetration of hybrid lipolexes (containing Cy5-DNA) in HepG2 multicellular tumor spheroid. The

multicellular tumor spheroid was cultured with lipoplexes for 6 h and visualized using CLSM in Z-stacks with 20 μm intervals. n = 3 independent experiments with similar results. The Scale bar is 100 μm. **d** Fluorescence intensity of various lipoplexes contained Cy5-DNA versus depth of tumor spheroid along the yellow arrows in Fig. 4c. **e** In vivo real-time observation of the lipoplexes contained Cy5-DNA penetration from the blood vessels of tumors through a dorsal window chamber under stereomicroscopy. Representative distribution images of hybrid lipoplexes contained Cy5-DNA in tumors after i.v. injection, observed by a CLSM at different times (20 μg Cy5-DNA per mouse). (Red: Cy5-labeled DNA). n = 3 mice. **f** Fluorescence intensity changes from the tumor vessel to an in-depth region (along the yellow arrows in Fig. 4e) in the selected area at 2 h post-intravenous injection. Two-sided unpaired Student's t test was used for the comparisons in (**b**). p values < 0.05 were considered statistically significant. Source data are provided as a Source Data file.

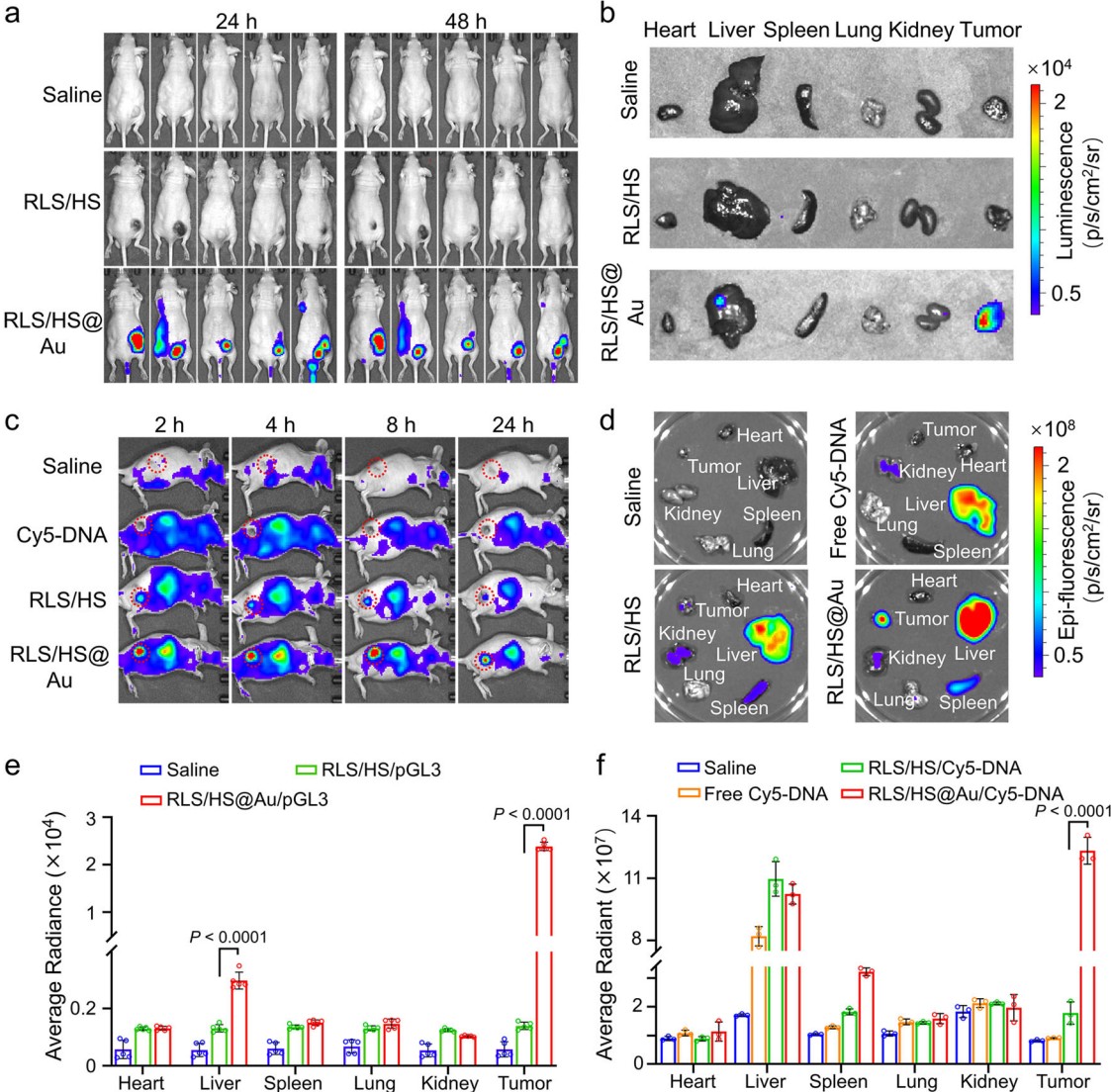

**Fig. 6 | In vivo transfection and biodistribution of hybrid lipoplexes in tumor model after intravenous injection. a** The pGL3 transfection of hybrid lipoplexes (RLS/HS@Au) in subcutaneous HepG2 xenograft mice, evaluated by bioluminescence imaging from an in vivo imaging system at 24 and 48 h after intraperitoneal administration of D-luciferin. $n = 5$ mice. **b** Representative luciferase expression in dissected organs of mice at 48 h after intravenous injection of different lipoplexes (20 μg pGL3/mouse). Representative images of in vivo **c** and ex vivo **d** biodistribution of indicated lipoplexes (containing 20 μg Cy5-DNA) in tumor-bearing nude mice at 2, 4, 8, and 24 h after intravenous injection. $n = 3$ mice. The red circle indicates the location of the tumors. **e** Bioluminescence quantification in major organs after administration of saline, RLS/HS/pGL3, and RLS/HS@Au/pGL3, respectively. $n = 5$ mice. **f** Fluorescent intensity of major organs at 24 h after intravenous injection of saline, naked plasmid DNA, RLS/HS, and RLS/HS@Au containing Cy5-labeled DNA. $n = 3$ mice. A two-sided unpaired Student's t-test was used for the comparisons in (**e**) and (**f**). The data in (**e** and **f**) are mean ± SD. $p$ values < 0.05 were considered statistically significant. Source data are provided as a Source Data file.

(Q) of the p53 plasmid and SP141 was 1.71 (Q > 1.15 indicates synergy) (Fig. 7d and Supplementary Fig. 30). The annexin V-FITC/Propidium Iodide (PI) apoptosis assay revealed that the co-delivery of the p53 plasmid and SP141 resulted in the highest apoptosis level (51.3%) compared to RLS/HS@Au/SP141 (32.3%) and RLS/HS@Au/p53 (25.6%), further demonstrating the combined therapeutic potential of RLS/HS@Au/SP141/p53 (Fig. 7e, f).

Subsequently, the antitumor efficacy of the combination therapy was evaluated using a HepG2 tumor-bearing mouse model. Mice were treated with intravenous injections of saline, RLS/HS@Au/*p53*, and the co-delivery system (RLS/HS@Au/SP141/*p53*) over seven cycles, after which tumor growth and survival were monitored (Fig. 8a). RLS/HS@Au/*p53* alone modestly inhibited tumor growth by 53%, while the addition of SP141 (RLS/HS@Au/SP141/*p53*) resulted in a greater tumor inhibition of 89.3% by day 18 (Fig. 8b, c). Notably, the co-delivery

system group demonstrated a significant delay in tumor growth and an improved survival benefit, with a median overall survival of 45 days (Fig. 8d and Supplementary Fig. 31). Furthermore, hematoxylin and eosin (H&E) staining in this group revealed extensive nuclear shrinkage and fragmentation (Fig. 8e). Immunohistochemical staining showed notable downregulation of MDM2 and accumulation of p53, while the TUNEL assay indicated a high number of apoptotic cells (Fig. 8e). Additionally, GTP-RhoA accumulation and cytoskeletal remodeling were confirmed in the HepG2 tumor model in vivo (Fig. 8f, g). To further assess therapeutic efficacy, we used a subcutaneous A549 lung carcinoma model, where the combined treatment of RLS/HS@Au/SP141/*p53* again resulted in significantly reduced tumor growth compared to *p53* treatment alone, supporting the therapeutic potential of targeting the p53/MDM2 pathway (Fig. 8h, i and Supplementary Figs. 32, 33).

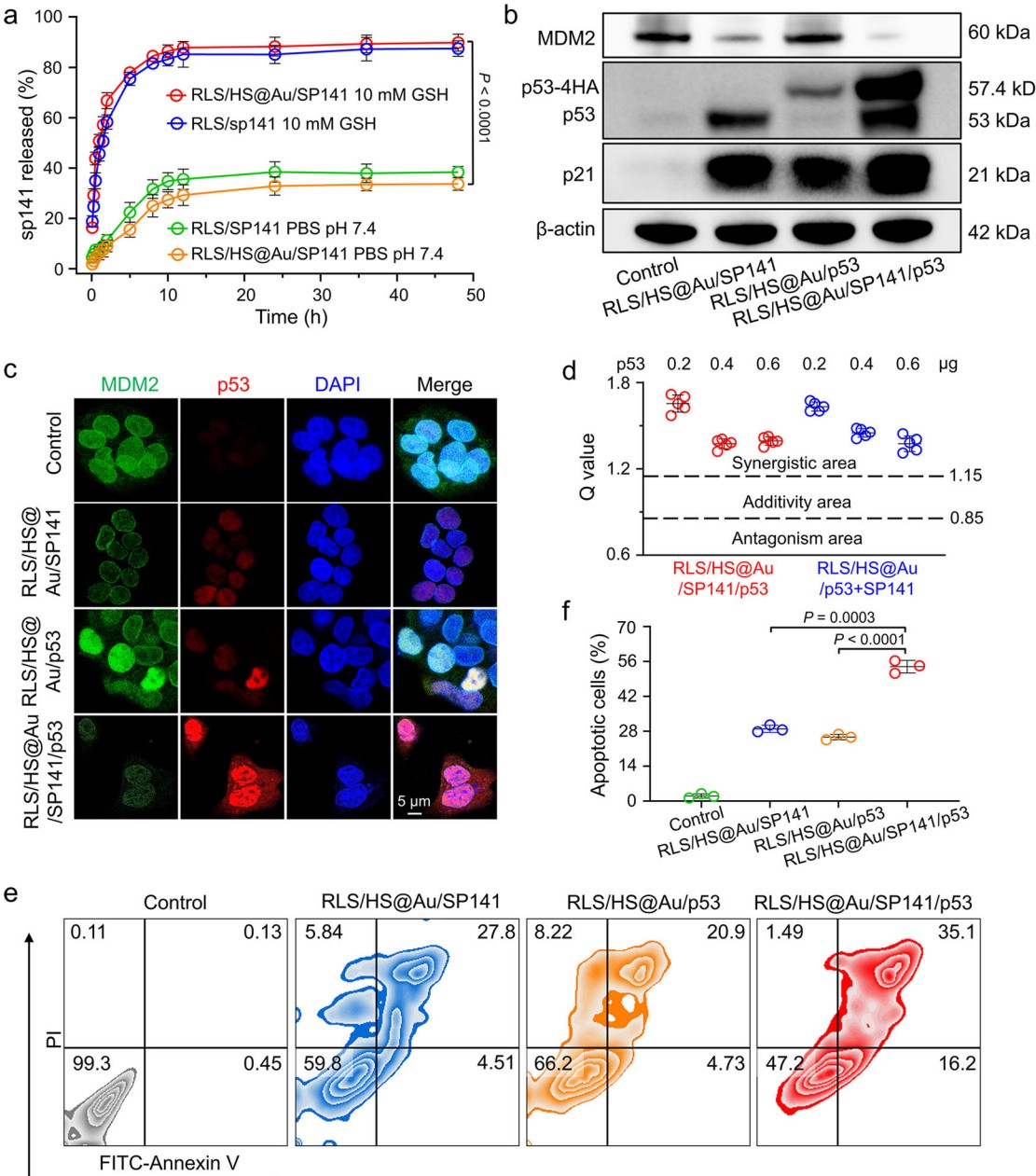

**Fig. 7 | In vitro antitumor effect and p53 activation function of hybrid lipoplexes with SP141 and p53 loading (RLS/HS@Au/SP141/p53) in HepG2 cells.**
**a** The release profile of SP141 from RLS/SP141 and RLS/HS@Au/SP141 lipoplexes analyzed by an ultraviolet spectrophotometer. *n* = 3 independent experimental units. **b** The expression levels of MDM2, p53, and p21 determined by western blot. HepG2 cells were exposed to various lipoplexes (at 1 μg/mL DNA) for 24 h, and the β-actin was used as a loading control. **c** Detection of MDM2 (green) and p53 (red) levels in cells through immunofluorescent staining after 24 h-incubation of various lipoplexes (with 1 μg/mL DNA). The nuclei of cells were stained by DAPI (blue). *n* = 3 independent experimental cell lines with similar results. The scale bar is 5 μm.

**d** Evaluation of the synergistic inhibition effect of p53 plasmid and SP141 on HepG2 cells. Q values were calculated from the cytotoxicity data for effect categorization: antagonism (Q < 0.85), additivity (Q < 1.15), or synergy (Q > 1.15). *n* = 5 independent experimental cell lines. **e** Apoptosis of HepG2 cells treated with different lipoplexes by flow cytometry detection with Annexin V-FITC and propidium iodide (PI) double-staining. f Histogram analysis of the cell apoptosis (%) by FlowJo software. *n* = 3 independent experimental cell lines. A two-sided unpaired Student's t-test was used for the comparisons in (**f**). The data in (**a**, **d**, and **f**) are mean ± SD. *p* values < 0.05 were considered statistically significant. Source data are provided as a Source Data file.

## Safety evaluation of the hybrid lipoplex

Safety evaluation was initially conducted on tumor-bearing mice. No significant change in body weight was observed across any treatment groups (Supplementary Figs. 33c, 34). Hepatic function indicators, including aspartate aminotransferase (AST) and alanine amino-transferase (ALT), remained within normal limits, with minimal differences in kidney function metrics such as creatinine (CREA) and blood urea nitrogen (BUN) between control and treatment groups

(Supplementary Fig. 35). Histological analyses revealed no notable differences in major organs across the treatment groups (Supplementary Fig. 36). Collectively, these findings demonstrated the acceptable biosafety profiles of RLS/HS@Au.

To further validate the safety of RLS/HS@Au/pDNA, additional tests were conducted on both female and male Balb/c mice without xenografted tumors. Mice were administered either a low dose (20 μg pDNA/mouse) or a high dose (40 μg pDNA/mouse) via intravenous

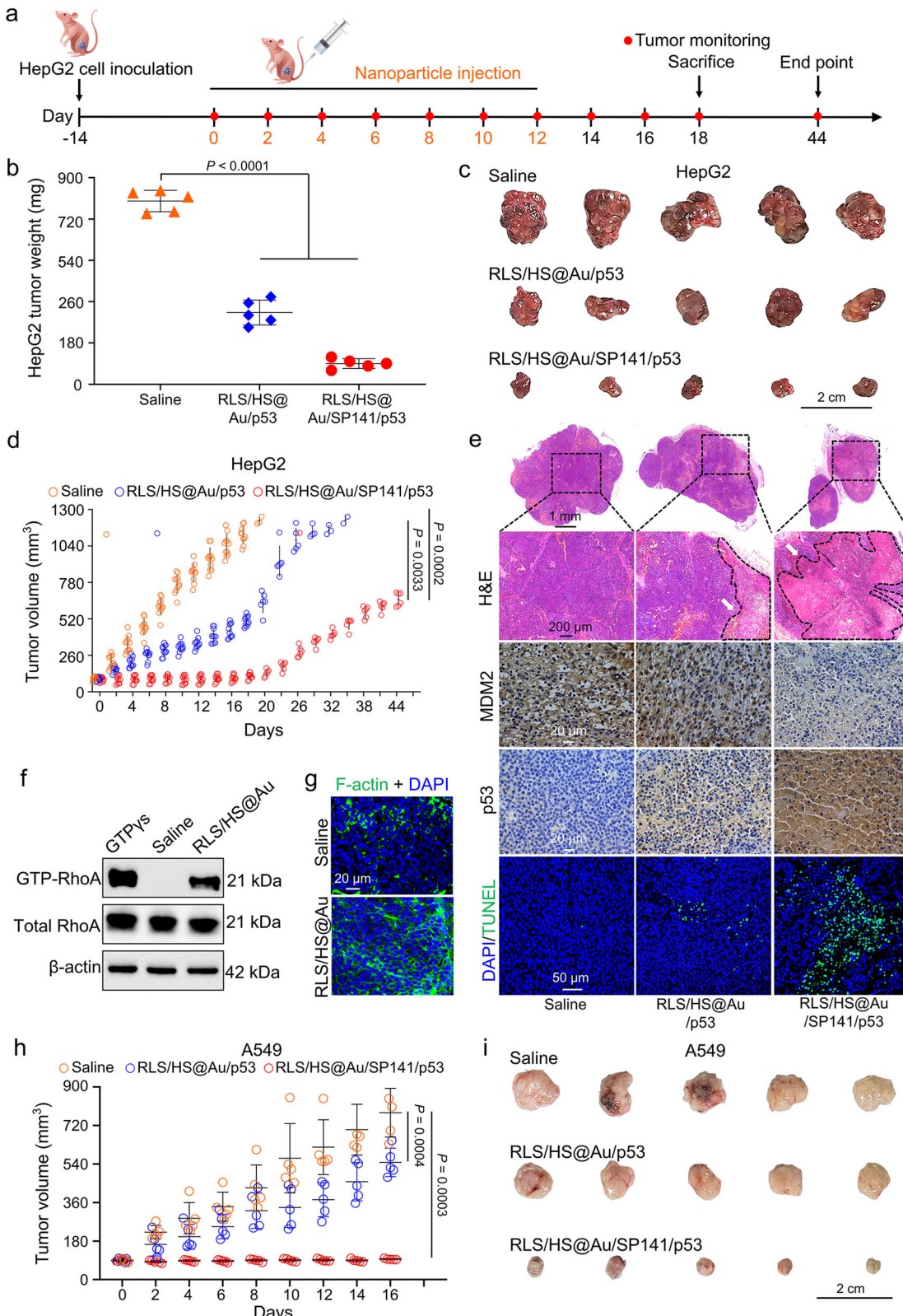

injection over seven doses at 1-day intervals (Supplementary Fig. 37a). No significant differences in body weight were noted between treated and saline control groups throughout the experiment (Supplementary Fig. 37b). On day 17 after injection, histopathological analyses of major organs, genotoxicity assessment, hematological parameters, and serum biochemistry tests were performed. The arrangement and structures of cells in all sectioned and stained organs (brain, heart,

lung, liver, spleen, kidney, and intestine) appeared normal, with no inflammatory or pathological changes observed (Supplementary Fig. 37c, d). No statistically significant differences in micronucleus formation rates were detected between the control and experimental groups (Supplementary Fig. 37e, f, g). Blood biochemical indices remained within normal ranges (Supplementary Fig. 37h, i, and Supplementary Table 2).

**Fig. 8 | Anticancer efficacy of hybrid lipoplexes (RLS/HS@Au/SP141/p53) in HepG2 and A549 tumor-bearing Balb/c athymic nude mice and validation of mechanisms of cytoskeleton remodeling in vivo. a** Establishment of the subcutaneous HepG2 tumor model and dosing regimen. **b** HepG2 tumor weight and **c** photographs of the excised tumors from different treatment groups. *n* = 5 mice. The scale bar is 2 cm. **d** Tumor growth curves of mice receiving hybrid lipoplexes treatments. *n* = 11 mice. **e** H&E staining of HepG2 tumors extracted from mice after the indicated treatments. The white arrow represents the necrotic area of the tumor. The scale bars are 1 mm and 200 μm, respectively. The HepG2 xenograft tumor sections were further analyzed for p53 and MDM2 protein expression with immunohistochemistry. The scale bar is 20 μm. Apoptotic events were determined by the TdT-mediated dUTP nick end labeling (TUNEL) assay. The scale bar is 50 μm. *n* = 3 mice. **f** Pull-down assay of GTP-RhoA in HepG2 tumor-bearing mice treated with tail vein injections of RLS/HS@Au/pDNA (pDNA 20 μg/mouse). The β-actin was used as a loading control, and GTPγs was the positive control. **g** Immunofluorescence staining of F-actin (green) in HepG2 tumors sections. *n* = 3 mice. The scale bar is 20 μm. **h** Tumor growth curves of A549 tumor-bearing Balb/c nude mice receiving hybrid lipoplexes treatments. *n* = 5 mice. **i** Photographs of the excised tumors from different treatment groups. The scale bar is 2 cm. Two-sided unpaired Student's t test was used for the comparisons in (**b**). The data in (**b, d,** and **h**) are mean ± SD. *p* values < 0.05 were considered statistically significant. Source data are provided as a Source Data file.

Moreover, developmental toxicity and teratogenicity of RLS/HS@Au/pDNA were evaluated in zebrafish embryos (Supplementary Fig. 38 and Supplementary Table 3), showing no differences in survival, hatching, spontaneous contraction, total coiling contractions, or malformation rates. These findings demonstrate that RLS/HS@Au/pDNA does not induce significant systemic toxicity.

The metabolism and distribution of RLS/HS@Au (20 μg pDNA/mouse) were evaluated in healthy Balb/c mice after intravenous injection. Inductively coupled plasma-atomic emission spectrometry (ICP-AES)-based gold content measurement revealed that over 95% of the total gold accumulated in the liver and spleen on days 15, 31, and 61 (Supplementary Fig. 39). This indicates that Au NPs were mainly metabolized in the liver and spleen, with minimal accumulation in other tissues over time.

## Discussion

Over the past five years, gene therapy has made significant progress, offering new treatments for intractable diseases. However, current pDNA-based gene delivery approaches remain largely confined to preclinical trials due to a lack of effective and safe delivery systems[6]. Developing more efficient nonviral gene vectors that mimic the infection mechanisms of natural viruses is a promising direction. Following this concept, we have designed a series of nano-delivery systems to promote pDNA delivery efficiency by mimicking viral nanostructures[13], enabling cascade targeting[14,17], promoting membrane penetration[15,16], and achieving transformations triggered by hierarchical stimulators. However, their transfection efficiency remains significantly lower than that of viruses. Meanwhile, more studies have revealed that most viruses utilize cytoskeletal networks to complete their infection processes, from entry through assembly to egress[18,24]. Moreover, inorganic nanoparticles have been identified to modulate the cytoskeleton by transmitting "outside-in" mechanical signals[25,26] and "inside-out" biochemical signals[27]. In this work, we introduce inorganic nanoparticles into cationic lipids to create a hybrid lipoplex, for further improving gene transfection efficiency through cytoskeletal regulation.

In this study, we selected an arginine-rich cationic lipid (RLS) and HS@Au for hybridization, as RLS has shown superior transfection efficacy compared to commercial reagents[15,43], and AuNPs have been reported to enhance transfection by modifying endocytosis pathways[44]. After optimizing the preparation methods and formulations, the hybrid lipoplexes exhibited improved serum stability over their nonhybrid counterparts (Fig. 2d). The introduction of negatively charged HS@Au likely reduced serum protein adsorption, thereby enhancing stability[45,46]. More importantly, the lipoplexes demonstrated an almost twofold increase in Young's modulus after hybridization (Fig. 2e, f). Cells treated with these "harder" RLS/HS@Au lipoplexes showed increased F-actin polymerization during the initial endocytosis phase (1–4 h) (Fig. 2g–i), suggesting that these "harder" lipoplexes engage the cytoskeleton more effectively, aiding in their endocytosis and intracellular transport.

Additionally, the hybrid lipoplexes were largely colocalized with F-actin, exhibiting a more perinuclear distribution within cells. These findings indicated that the "harder" lipoplexes leverage "outside-in" mechanical signaling to promote cellular endocytosis, subsequently utilizing F-actin as a conduit for endosome transport toward the nucleus. This is advantageous for pDNA delivery and transfection efficiency. Additionally, the activated RhoA and enhanced F-actin polymerization promoted cell migration (Fig. 2h), likely due to dynamic changes in the actin cytoskeleton, with RhoA acting as a key regulator throughout the continuous cycles of cell protrusion, adhesion, and contraction[47,48]. However, the precise mechanisms and signaling pathways involved remain unclear.

We then assessed the pDNA transfection efficiency and intracellular dynamics of the hybrid lipoplex. Incorporating HS@Au significantly enhanced gene transfection in vitro (Fig. 2a, b), achieving a 12-fold increase in EGFP expression compared to Lipofectamine 2000 at 24 h post-transfection. Additionally, green fluorescence was visible as early as 6 h after transfection, which did not appear in other groups (Fig. 5a). This rapid gene expression may stem from the enhanced endocytosis initiation capabilities of the hybrid lipoplex (Fig. 3c–i). Typically, nanoparticle endocytosis includes pathways like clathrin-mediated endocytosis, caveolae-mediated endocytosis (CVME), clathrin- and caveolae-independent endocytosis (CCIE), and macropinocytosis (MP)[49,50]. Among these, the CVME, CCIE, and MP pathways were triggered by and dependent on cytoskeletal reorganization[51]. Moreover, RLS/HS@Au may activate more actin-dependent endocytosis pathways[44].

Actin filaments play a crucial role in sensing extracellular mechanical signals, such as substrate stiffness[52,53]. Therefore, we hypothesize that the "harder" RLS/HS@Au structure may directly activate mechanical-sensing proteins at the nanoparticle-cell interface (e.g., integrins, focal adhesions, and various ion channels), subsequently stimulating downstream signaling pathways (e.g., the Rho family small GTPases)[54] that drive for cytoskeletal reorganization. This mechanism, termed "outside-in" mechanical signaling transduction, may facilitate enhanced intracellular transport. RLS/HS@Au also showed advantages in lysosomal escape, a critical bottleneck in gene transfection, likely due to the cell-penetrating properties of the RLS lipid nanoparticle[15,16] and the acidic environment-disrupting effect of AuNPs[36]. The hybrid lipoplex successfully delivered pDNA to the nucleus, achieving closer proximity to the nucleus and being more localized inside it. While RLS/HS@Au appears to accelerate nanoparticle transport kinetics, further studies are needed to confirm this finding, given the impact of enhanced endocytosis and rapid lysosomal escape efficiency.

The hybrid lipoplex has thus far demonstrated enhanced intracellular delivery at each step, including cellular uptake, lysosomal escape, and nuclear import. This enhancement improved pDNA transfection and led to notable cytoskeletal rearrangement. The underlying mechanisms and relationships among these phenomena captured our interest. Previous studies have shown that

Rho GTPase plays a key role in vesicle transport by regulating the cytoskeleton[32,55] during cell endocytosis and actin polymerization, with GTP-RhoA turnover inside cells regulated by autophagic flux[34]. Therefore, we speculated that RLS/HS@Au might inhibit autophagic flux, leading to GTP-RhoA accumulation and subsequently enhancing F-actin polymerization. This hypothesis was supported by our findings of increased GTP-RhoA levels (Figs. 4a and 8f) and reduced gene transfection upon treatment with GTP-RhoA inhibitors (Fig. 4c, d). Furthermore, we confirmed that autophagy inhibition occurred at the early autophagosome formation and late lysosomal proteolysis (Fig. 4e–j). Inhibiting early autophagosome formation allowed more GTP-RhoA to localize at the plasma membrane (Fig. 4b), promoting actin polymerization into F-actin. Additionally, RLS/HS@Au induced greater GTP-RhoA membrane localization than HS@Au alone, even though HS@Au induced a higher overall GTP-RhoA expression (Fig. 4a, b). We speculated that GTP-RhoA sequestered in autolysosomes could escape with the help of RLS, which can alter membrane fluidity and enhance cargo penetration within the lysosome[16].

Moreover, the accumulation of AuNPs in lysosomes leads to lysosomal pH alkalinization, impairing lysosomal proteolytic capacity (Fig. 4i). Thus, RLS/HS@Au may increase the plasma membrane distribution of GTP-RhoA by inhibiting autophagosome formation, which sequesters GTP-RhoA, and by facilitating the escape of sequestered GTP-RhoA from autolysosomes. Notably, cytoskeletal remodeling appears to result from the coregulation of mechanical and biochemical signals.

Cytoskeletal remodeling affects not only intracellular but also intercellular transport, as actin filaments can provide forces for both endocytosis and exocytosis[22,37], potentially enhancing the transcellular transport of RLS/HS@Au. In our experiments, we observed increased vascular extravasation and infiltration into solid tumors (Fig. 5e), validating enhanced intercellular transport between tumor cells and between endothelial and tumor cells. This intercellular transport further promoted the specific expression of pGL3 and increased the accumulation of Cy5-labeled hybrid lipoplex in solid tumors (Fig. 6). Notably, while the distribution of RLS/HS@Au and RLS/HS in the liver showed no statistical difference, gene expression varied, suggesting that hybridization may aid the lipoplex in resisting rapid liver metabolism.

In conclusion, hybridization offered advantages by promoting cytoskeletal rearrangement through "inside" autophagy/RhoA signaling, facilitating the "outside" intra- and intercellular delivery processes, and ultimately increasing lipid nanoparticle accumulation at tumor sites.

Finally, we validated the potential of this delivery system for anti-tumor applications by co-delivering the p53 plasmid and the MDM2 inhibitor SP141 with RLS/HS@Au. It showed remarkable synergistic anti-tumor potential in vitro (Fig. 7), achieving a 90% inhibition of xenograft hepatic tumors and extending overall survival to 45 days in vivo (Fig. 8a–e and Supplementary Fig. 30). Additionally, the therapeutic potential of the p53/MDM2 targets was confirmed in the A549 lung cancer model, achieving an inhibition ratio of approximately 86% (Fig. 8h, i). However, tumor recurrence was observed after the cessation of treatment (Fig. 8d), suggesting that a rational combination of p53 targeting with other therapies (such as chemotherapy or immunotherapy) or increased dosage and treatment cycles may enhance the durability of the response. Importantly, RLS/HS@Au did not cause any apparent systemic safety concerns at the administered doses (Supplementary Fig. 37 and Supplementary Table 2). Overall, we believe RLS/HS@Au represents a promising gene delivery platform with high transfection efficiency, potentially applicable to various tumor-specific gene delivery needs and combinable with other therapeutic approaches to achieve effective tumor therapy, supporting the clinical translation of nanomaterials.

## Methods

### Ethics statement

All animal experiments were kept to a strict protocol approved by the Institutional Animal Care and Ethics Committee of Sichuan University and conducted with the guidelines for the care and use of laboratory animals of Sichuan University (ethical approval code SCU46-2401-01). The maximal tumor size/burden permitted by our ethics committee or institutional review board is 2000 mm³, and the maximal tumor size/burden was not exceeded during the experimental process.

### Materials

Tween 80 and hydrogen tetrachloroaurate (III) hydrate (HAuCl₄·X H₂O) were purchased from Adamas (Switzerland). Sodium citrate and sodium hyaluronate (HA, 40 kDa) were purchased from Sinopharm (China). Cysteine, 1-ethyl-3-[3-(dimethylamino)-propyl] carbodiimide (EDC), and N-hydroxysuccinimide (NHS) were obtained from Sigma (USA).

Roswell Park Memorial Institute 1640 medium (RPMI 1640), Dulbecco's Modified Eagle's Medium with high glucose (DMEM-HG), Minimum Essential Medium (MEM), penicillin-streptomycin, and fetal bovine serum (FBS) were purchased from Gibco (USA). Label IT® Tracker™ Intracellular Nucleic Acid Localization Kit (Cy5®, MIR7021) was obtained from Mirus Bio Corporation (USA). D-luciferin potassium salt was purchased from Yeasen (China). Enhanced green fluorescent protein-encoding plasmid (pEGFP), luciferase reporter plasmid (pGL3), and lipofectamine 2000 were obtained from Invitrogen (USA). The plasmid encoding the human wild-type p53 tumor suppressor protein tagged with hemagglutinin (p53-4HA) was a kindly gift provided by Professor Feng Bi at West China Hospital of Sichuan University. The LC3-EGFP-mCherry plasmid was a gift kindly provided by Professor Yunjiao Zhang at the South China University of Technology. All the plasmids were propagated in Trelief® 5α Chemically Competent Cell (catalog number: TSC-C01, TsingkeBiotechnology Co., Ltd.) and purified via Endo-Free Plasmid Kit from Qiagen (Germany). Lyso-Tracker Green DND-26 (L7526), CellMask™ Green Plasma Membrane stain (C37608), and bicinchoninic acid (BCA) protein assay kit (NCI3225CH) were obtained from Thermo Fisher (USA). Cell counter kit-8 (CCK-8) was purchased from Dojindo Laboratories (Japan). Antibodies against p62 (catalog number: ab56416; clone number: 2C11; lot number: GR3374761-1; 1:1000 dilution) and Phalloidin-iFluor 488 (ab176753) were from Abcam (UK). The antibody against LC3 (catalog number: NB100-2220; clone number: N/A; lot number: D155067; 1:1000 dilution) was from Novus (USA). Antibodies against p53(catalog number: sc-126; clone number: DO-1; lot number: l2722; 1:1000 dilution), MDM2 (catalog number: sc-965; clone number: SMP-14; lot number: J2922; 1:1000 dilution), and p21 (catalog number: sc-817; clone number: 187; lot number: E2620; 1:1000 dilution) were purchased from Santa Cruz Biotechnology (USA). DTT, Hoechst 33342, 4′,6-diamidino-2-phenylindole (DAPI), RIPA lysis buffer, 4% paraformaldehyde (PFA) and antibody to β-actin (catalog number: AF0003; clone number: N/A; lot number: 102723240430; 1:1000 dilution) were purchased from Beyotime Biotechnology (China). The antibody against LAMP1 (catalog number: 9091 T; clone number: D2D11; lot number: 8; 1:400 dilution), goat anti-mouse IgG-HRP antibody (catalog number: 7076P2; clone number: N/A; lot number: 36; 1:5000 dilution), goat anti-rabbit IgG-HRP antibody (catalog number: 7074P2; clone number: N/A; lot number: 32; 1:5000 dilution), anti-mouse IgG (H + L) AlexaFluor-488 (catalog number: 4408S; clone number: N/A; lot number: 22;1:2000 dilution) conjugated antibody, and goat anti-rabbit IgG (H + L) AlexaFluor-647 (catalog number: 4414S; clone number: N/A; lot number: 26; 1:2000 dilution) conjugated antibody were obtained from Cell Signaling Technology (USA). C3 Transferase Protein (C3) and Rho Activation Assay Biochem Kit were purchased from Cytoskeleton, Inc. (USA). The antibody against RhoA (catalog number: ARH05; clone number: 64D6.1.16; lot number: 003; 1:400 dilution) was from the Rho

Activation Assay Biochem Kit. DQ-BSA (D-12051) was obtained from Molecular Probes (USA). The trichloroacetic acid (TCA), chloroquine (CQ), NH₄Cl, and SP141 were obtained from MedChemExpress (USA). Annexin V-FITC/PI apoptosis detection kit was obtained from Oriscience Biotechnology (China). The pentobarbital sodium was purchased from Ailu Biological Tech (China).

## Cell culture

The human hepatoma cell lines (HepG2, catalog number: SCSP-510), the lung carcinoma cell line (A549, catalog number: SCSP-503), human cervical carcinoma cell lines (HeLa, catalog number: SCSP-504), breast tumor cell lines (4T1, catalog number: TCM32), and fibroblast cell lines (NIH-3T3, catalog number: SCSP-515) were obtained from the Chinese Academy of Science Cell Bank for Type Culture Collection (China). These cell lines were authenticated using STR analysis and the last authentication dates are as follows: HepG2 on May 19, 2024; A549 on May 30, 2023; HeLa on September 28, 2023; 4T1 on November 13, 2023; and NIH-3T3 on August 15, 2023. All cell lines tested negative for mycoplasma contamination examined by qPCR-based assay. Cells used in this study were all cultured in the medium containing 10% FBS, 100 units/mL penicillin, and 100 μg/mL streptomycin at 37 °C in a humidified atmosphere containing 5% $CO_2$. The HepG2 and HeLa cells were maintained in MEM. The 4T1 and A549 cells were in RPMI 1640. NIH-3T3 cells were in DMEM-HG.

## Preparation and characterization of hyaluronic acid-modified gold nanoparticles (HA-S@Au)

The HA-S@Au were prepared using in situ and ligand exchange methods. In the in-situ method, HS was synthesized[56]. Briefly, HA (300 mg), NHS (35 mg), and EDC (100 mg) were successively dissolved in water with pH of 4.5 and stirred for 2 h. Next, cysteine (600 mg) was added and reacted overnight at 30 °C. The reaction solution was dialyzed (MWCO 3500 Da) and freeze-dried to obtain the HS, characterized by nuclear magnetic resonance hydrogen spectroscopy (¹H NMR, Bruker Avance II NMR spectrometer, 400 MHz, Germany), Ellman's assay (DTNB), and gel permeation chromatography (GPC, Waters 1515, USA). Then, the purified HS (12 mg) was redissolved in water and heated to 100 °C, followed by dropwise adding HAuCl₄·X H₂O (2 mg). The solution was cooled, centrifuged (15800 g for 15 min), and rinsed to yield HS@Au.

The HS@Au was also prepared through the traditional ligand exchange method[57]. In brief, the water solution of HAuCl₄·X H₂O (5 mg) was boiled, added with sodium citrate (0.5 mg), and reacted for 10 min. Then, the reaction solution was cooled, centrifuged (15800 g, 15 min), and rinsed to gain the sodium citrate-coated AuNPs (SC@Au). Afterward, the HS (12 mg) and SC@Au (2 mg) were mixed and stirred for 4 h. The final product (HS@Au_le) was collected by centrifugation (2400 g, 10 min).

The above-prepared gold nanoparticles (HS@Au, HS@Au_le, and SC@Au) were quantified by ICP-AES (5100 SVDV, USA). In short, samples were dissolved in aqua regia for 24 h, then the clarified liquid was collected and diluted to 10 mL with water and analyzed with ICP-AES to determine the concentration. The size and ζ potential were characterized by dynamic light scattering (DLS) using a Zetasizer Nano ZS (Malvern Instruments, UK), and the morphologies were observed by transmission electron microscopy (TEM, JEM-1011, Japan). Absorption spectra of the AuNPs were measured by a UV-visible spectrophotometer (UV 2600, Japan). The amount of coating on Au NPs was determined with a thermal gravimetric analyzer (TGA, Rigaku TG-DTA 8122, Japan).

## Preparation and characterization of pDNA loaded or p53/SP141 co-loaded hybrid lipoplexes

For the preparation of pDNA-loaded or *p53*/SP141 co-loaded hybrid lipoplexes, RLS was synthesized and assembled using the solvent injection method[43]. The obtained assemblies were mixed with DNA at an N/P ratio of 20, and incubated for 20 min, followed by the addition of HS or HS@Au with another 20 min incubation to prepare hybrid ternary lipoplexes at different mass ratios (Au or HS versus DNA). For *p53*/SP141 co-loaded hybrid lipoplexes, RLS was first mixed with SP141 (at a mass ratio of 10: 0.1) in methanol (100 μL) and then added dropwise into deionized water (0.9 mL). The solution was continuously stirred at room temperature overnight, and centrifuged at 400 g for 5 min. The supernatant was collected and mixed with p53 plasmid and HS@Au to fabricate *p53*/SP141 co-loaded hybrid lipoplexes. The drug loading capacity (LC) and encapsulation efficiency (EE) were calculated by dissolving the lipoplexes in methanol (1:5, v/v). The absorbance of SP141 was detected at 295 nm using a UV-visible spectrophotometer. LC and EE were defined as follows:

$$LC(\%) = \frac{M_{SP141\ in\ supernatant}}{M_{SP141\ in\ supernatant} + M_{RLS}} \times 100 \quad (1)$$

$$EE(\%) = \frac{M_{SP141\ in\ supernatant}}{M_{Total\ SP141}} \times 100 \quad (2)$$

where $M_{SP141\ in\ supernatant}$ represented the mass of SP141 in the supernatant, $M_{Total\ SP141}$ was the mass of SP141 initially added, and $M_{RLS}$ was the mass of RLS initially added.

The size and ζ zeta potential of various lipoplexes were detected by DLS, and the morphologies were observed by TEM. The measurement of Young's modulus was conducted by AFM (Multimode 8, Bruker). For RLS, the mica sheet is untreated. But for RLS/HS@Au, the mica sheet underwent immersion in polyethyleneimine solutions to be modified with a positive charge. Altogether, 50 μL of lipoplexes mixtures (RLS/DNA or RLS/HS@Au/DNA) were dropped onto the mica sheet and preserved for 20 min to confirm their adherence. Subsequently, the AFM images and force curve measurement were conducted using tapping mode, and Young's modulus was additionally determined through NanoScope Analysis software (Ver. 1.5).

In addition, lipoplexes (containing 200 ng DNA) were incubated with DTT (10 mmol/L) in MEM medium for 2 h at 37 °C to evaluate the gene release. All samples were loaded onto 1% agarose gel for electrophoresis (90 V, 1 h). The gel was stained with GelRed and observed by the Molecular Imager ChemiDoc XRS+ (Bio-Rad, USA).

Next, the lipoplexes protection of DNA was evaluated. Different lipoplexes were incubated with DNase (200 U/mL) at 37 °C for 15 min, and EDTA buffer was added to inactivate the enzyme. After that, heparin (4 mg/mL) was added and incubated for 2 h to release the compressed DNA for further electrophoresis evaluation. To investigate the stability, the size changes of various DNA lipoplexes were measured in PBS, culture medium with 10% FBS, and 10% plasma at 0, 4, 12, 24, and 48 h. Furthermore, the long-term stability of the hybrid nanoparticles in an aqueous solution was also assessed using DLS in one month.

For in vitro drug release from SP141-loaded or *p53*/SP141 co-loaded hybrid lipoplexes. The lipoplexes solution (RLS, 3 mg/mL, 3 mL) was transferred in a dialysis bag (MWCO 1000 Da) and immersed in PBS (20 mL, containing 0.1% Tween 80 (v/v), pH 7.4) with or without GSH (10 mmol/L) at 37 °C with gentle shaking. At pre-determined time points, 700 μL of samples were taken out and replaced with fresh release medium. The SP141 content in withdrawn samples was determined using a UV-visible spectrophotometer according to the established standard curve.

## The effect of hybrid lipoplexes or HS@Au on the cytoskeleton

HepG2 cells ($4 \times 10^4$ cells/well) were plated into 35 × 12 mm confocal dishes and incubated with various gene lipoplexes (200 ng Cy5-DNA) or free HS@Au (50 μg/mL) for 4 or 24 h. Cells

were fixed with 4% PFA for 15 min and permeabilized with 0.1% Triton-X 100 for 3 min and stained with FITC-Phalloidin or Rhodamine-phalloidin (100 nmol/L, 30 min) and DAPI (100 nmol/L, 1 min). Then cells were washed and photographed by CLSM. The number of stress fibers and perinuclear actin cap were quantified by counting individual fibers in images.

For random migration of single cells, HepG2 ($1 \times 10^5$ cells/well) was seeded in the 12-well plates and cultured in a live-cell incubation chamber with various lipoplexes (containing 1 µg DNA). Live-cell time-lapse imaging was conducted using a DMI6000B Leica system for 24 h at 10 min intervals. Cell images were analyzed in ImageJ to obtain the (x, y) coordinate, and the migration trajectories and distances were calculated.

## In vitro pEGFP transfection and cytotoxicity assay

Transfection of pEGFP was studied in HepG2, HeLa, 4T1, and NIH 3T3 cells seeded in 96-well plates ($1 \times 10^4$ cells/well). Cells were transfected with DNA-loaded lipoplexes (200 ng pEGFP) for 4 h, then incubated in fresh medium for an additional 44 h. Lipo 2000 was added according to the manufacturer's protocol as a positive control. The EGFP expression was measured by an inverted fluorescence microscope (Olympus, Japan), and semi-quantitatively analyzed using the ImageJ. The cytotoxicity of gene lipoplexes was determined by CCK-8 assay. Briefly, cells were seeded in 96-well plates and treated with indicated lipoplexes. After incubation with fresh serum-free medium containing 10% CCK-8 for 2 h, absorbance at 450 nm was measured using a microplate reader (Bio-Rad 550, USA).

## Cellular trafficking of various lipoplexes

Cy5-labeled DNA (Cy5-DNA) was used to track and image intra- or inter-cellular behaviors. Labeling was performed using the Label IT® Kit according to the manufacturer's protocol. Briefly, pGL3 (20 µg, 0.5 µg/µL) was incubated with the Label IT tracker reagent at 37 °C for 3 h. After adding ice-cold EtOH and 5 mol/L NaCl, Cy5-DNA was precipitated overnight at −20 °C, followed by centrifugation (15800 g, 1 h, 4 °C) and purification.

For cellular uptake imaging, cells were plated into 35 × 12 mm confocal dishes and incubated with various lipoplexes (200 ng Cy5-DNA) for 1, 2, 4, and 6 h. Then cells were counterstained with CellMask™ Green (5 µg/mL, 10 min) for the cell membrane and Hochest 33342 (5 µg/mL, 15 min) for the nucleus. After washing off uninternalized complexes, cells were observed by CLSM (LSM 880, Zeiss) with excitation wavelengths of 633, 522, and 350 nm and emission wavelengths of 670, 535, and 461 nm, respectively. ICP-AES was used to assess the endocytosed AuNPs. HepG2 cells treated with nanoparticles (RLS/HS@Au/DNA, 15.4 µg Au; HS@Au, 50 µg Au) for 24 h were trypsinized, counted and centrifuged (95 g, 2 min), and analyzed by ICP-AES.

To determine endosome/lysosome escape, cells were stained with LysoTracker Green DND-26 (50 nmol/L, 30 min) after incubation with different gene lipoplexes (200 ng Cy5-DNA) for 2, 4, and 6 h. Intracellular localization of endosome/lysosome and Cy5-DNA were observed by CLSM (LSM 880, Zeiss).

Nuclear delivery study of DNA was studied on HepG2 cells with the same treatments as above. The distance between Cy5-DNA and the nucleus at indicated time points after lipoplexes treatments was evaluated by ZEN 2.3 (Blue edition, Zeiss) software. Nuclear localization of Cy5-DNA was observed by CLSM in XYZ-3D-stack at 0.5 µm intervals, followed by the 3D construction with Imaris 9.0.1 software.

## The pull-down assay and chemical inhibition of GTP-RhoA

The pull-down assay of GTP-RhoA was performed according to the instructions of Rho Activation Assay Biochem Kit. In brief, HepG2 cells were seeded in 6-well plates ($2 \times 10^5$ cells/well) and treated with different lipoplexes (containing 2 µg pEGFP) or HS@Au (50 µg/ml) for

48 h. After washing, cells were lysed and the supernatant was collected and quantified by BCA assay. 1.5 mg of cell lysate was affinity-precipitated with 50 µg of GST-RHOTEKIN-RHO binding domain (RBD) fusion proteins pre-coupled to glutathione-agarose beads at 4 °C for 1 h. Beads were washed and the bound GTP-RhoA was eluted with SDS buffer. The achieved samples and total RhoA (in whole cell lysates) were loaded onto 12% SDS-PAGE followed by anti-RhoA western blotting. Bands were visualized with an ECL kit (1705060, Bio-Rad) and ChemiDoc XRS + System.

C3 transferase was used to chemically inhibit GTP-RhoA activity. HepG2 cells were cultured in a 12-well plate and treated with C3 transferase (1 µg/mL) in serum-free medium for 4 h. Cells were then incubated with different gene lipoplexes (1 µg pEGFP) in the absence or the presence of C3 in medium for 4 h. Finally, the culture medium was refreshed with or without of C3. After 44 h, EGFP expression was observed by fluorescence microscope. Cells were then washed, trypsinized, collected, and subjected to the flow cytometry analysis (BD Accuri C6, USA).

For immunofluorescence staining of RhoA, cells were seeded on glass coverslips and treated with various liposomes (1 µg pDNA/well) or free HS@Au (50 µg/mL) for 24 h, fixed with ice-cold 10% TCA for 15 min. They were then washed and permeabilized in 0.3% Triton X-100 for 5 min. After that, cells were incubated with the antibody against RhoA and DAPI for CLSM observation (LSM880, Zeiss).

## Effects on autophagy flux

For the DQ-BSA assay, HepG2 cells were plated in a 35 × 12 mm confocal dish and treated with various gene lipoplexes (containing 3 µg DNA) or free HS@Au (50 µg/mL) for 24 h. After washing, cells were incubated with DQ-BSA (10 µg/mL) for 3 h at 37 °C and imaged by CLSM (LSM880, Zeiss).

For detection of p62 and LC-3 expression, HepG2 cells were seeded in 12-well plates and treated with different concentrations of HS@Au or gene lipoplexes (3 µg DNA) with or without chloroquine (10 µmol/L) for 24 h. Afterward, cells were washed with PBS and lysed by RIPA lysis buffer containing a protein inhibitor mixture (Roche, Switzerland) for 30 min at 4 °C. After centrifugation (15800 g, 10 min), the collected supernatant was loaded onto 12% SDS-PAGE gel. Then proteins were transferred to a PVDF membrane (Bio-Rad), followed by incubation with primary and secondary antibodies. Protein signals were detected using an imaging system.

To visualize autophagic flux, HepG2 cells were transfected with various lipoplexes containing 200 ng LC3-EGFP-mCherry plasmid. LC3-EGFP-mCherry-expressing cells were obtained after a 24 h or 48 h transfection period. For starvation, HepG2 cells were cultured in EBSS buffer for 5 h after lipoplexes treatment. To identify the location of the lysosome, after transfection, cells were fixed in 4% PFA, permeabilized with 0.1% Triton X-100, and blocked with 3% BSA. Cells were then incubated with rabbit anti-LAMP1 antibody at 4 °C overnight, and subsequently incubated with secondary goat anti-rabbit IgG (H + L) AlexaFluor-647 conjugated antibody, counterstained with DAPI, and imaged.

For observation of autophagosome and autolysosome, HepG2 cells were seeded in 6-well plates, treated with various gene lipoplexes (3 µg DNA) or free HS@Au (50 µg/mL) for 24 h, washed with PBS, collected by digestion, and centrifuged following TEM sample preparation. They were fixed with 3% glutaraldehyde, dehydrated through graded acetone, and embedded. Ultrathin sections were cut and stained with uranyl acetate and lead citrate. Sections were examined using TEM (JEM-1400-FLASH, Japan).

## Transcytosis transportation of hybrid lipoplexes

HepG2 cells were plated in 35 × 12 mm confocal dishes. The first batch of cells was treated with different gene lipoplexes containing Cy5-DNA (1 µg) or pEGFP (600 ng) for 6 h, washed with PBS, and then imaged.

These cells were cultured in a fresh medium for another 12 h and imaged. Meanwhile, the culture medium was harvested and added to a new plate of cells (regarded as the second batch) followed by a 12 h incubation and fluorescent imaging.

To prepare multicellular tumor spheroids, HepG2 cells were suspended in fresh MEM medium (containing 0.12% w/v methylcellulose) at $4 \times 10^5$ cells/mL density. 25 µL of cell suspensions were dropped on the lids of the tissue culture dish, followed by adding 20 mL PBS in a well to maintain moisture. After 72 h, spheroids were formed and moved to a 96-well plate coated with agarose (1% w/v in PBS). Each well contains a single spheroid and is incubated for an additional 72 h for maturation. Then the HepG2 multicellular tumor spheroids were transferred to a 35 × 12 mm confocal dish and incubated with different gene lipoplexes containing 1 µg Cy5-DNA for 6 h. Images were captured using CLSM in XYZ-3D-stack at 20-µm intervals from top to equator.

### Western blot and immunofluorescence assay of p53 and MDM2

HepG2 cells were treated with *p53*-loaded, *pGL3*/SP141 co-loaded, or *p53*/SP141 co-loaded hybrid lipoplexes for 24 h. They were digested, collected, and lysed for western blotting as in LC3 and p62 detection. For the immunofluorescent detection of p53 and MDM2, HepG2 cells were grown on sterile coverslips in 12-well plates and exposed to various gene lipoplexes for 24 h. After similar treatments in LAMP-1 staining, the cells were incubated with the primary and secondary antibodies, counterstained with DAPI, and imaged by CLSM.

### In vitro antitumor activity through reactivation of p53 functions in HepG2 cells

For in vitro tumor suppression of *p53*/SP141 co-loaded hybrid lipoplexes, cells were seeded in 96-well plates and incubated overnight. Then, various lipoplexes (0.2 µg p53 plasmid) were added to each well. After 4 h incubation, the medium was displaced by fresh medium and the cells were cultured for another 44 h. The CCK-8 assay procedure was consistent with the toxicity assay above. The inhibition rate was calculated as follows:

$$\text{Inhibition rate(\%)} = \left[ 1 - \frac{(\text{OD}_{sample} - \text{OD}_{blank})}{(\text{OD}_{control} - \text{OD}_{blank})} \right] \times 100 \qquad (3)$$

where $\text{OD}_{control}$ and $\text{OD}_{sample}$ are the absorbances in the absence and presence of lipoplexes, respectively and $\text{OD}_{blank}$ is the absorbance of the medium.

The combination effect of *p53* and SP141 was analyzed by a "Q" value through the following formula[58]:

$$Q = \frac{I_{p53 + SP141}}{I_{p53} + I_{SP141} - I_{p53} \times I_{SP141}} \qquad (4)$$

where $I_{p53}$, $I_{SP141}$, and $I_{p53+SP141}$ are the tumor inhibition rates of only p53-loaded, pGL3/SP141 co-loaded, and p53/SP141 co-loaded hybrid lipoplexes, respectively.

For apoptosis assay by Annexin V-FITC/PI staining, HepG2 cells were seeded in 12-well plates and treated with various lipoplexes for 24 h. After trypsinization without EDTA, cells were collected, washed, and detected by flow cytometer according to the Annexin V-FITC kit protocol.

### In vivo real-time penetration of hybrid lipoplexes

Male Balb/c nude mice (4-6 weeks) were purchased from Gem-Pharmatech LLC. (China). The mice were inoculated with $2 \times 10^6$ HepG2 cells adjacent to an abdominal blood vessel. When the tumor reached a size of 30 mm³, mice were intraperitoneally anesthetized with 1% pentobarbital sodium (50 mg/kg), and a dorsal window chamber (APJ Trading Co. Inc, USA) was implanted in the back. The skin on one side

of the chamber was peeled off to approximately 1 cm in diameter, while the other side (containing the tumor) was intact. Finally, a coverslip (12 mm diameter) was placed over and fixed on the peeled skin. Blood vessels around the tumor were observed by CLSM (A1R MP+, Nikon). Cy5-DNA loaded lipoplexes (20 µg DNA per mouse) were i.v. injected. Tumor imaging was performed at timed intervals, and fluorescence intensity from blood vessels to the tumor was measured using ImageJ. Tumor and blood vessel morphology were imaged by stereomicroscopy (SteREO Discovery V20, Zeiss). In addition, all mice used in this study were housed under a 12-hour light/12-hour dark cycle in an animal facility under specific pathogen-free conditions, and maintained at 25 °C with humidity levels between 40% to 70%.

### In vivo and ex vivo fluorescent imaging in tumor-bearing mice

The male Balb/c nude mice were subcutaneously inoculated with $1 \times 10^7$ HepG2 cells (100 µL) next to the right hind leg. Once the tumor volume reached about 100 mm³, the mice were i.v. injected with Cy5-DNA loaded lipoplexes (20 µg DNA/mouse). Whole-body optical imaging was performed at 2 to 24 h post-injection on IVIS Lumina Series III imaging system (Perkin Elmer, USA). After 24 h injection, the mice were sacrificed and organ and tumor tissues were excised and imaged, followed by fluorescence quantification using the IVIS Spectrum Software.

### In vivo luciferase transfection to HepG2 tumors

When the tumor volume grew to 100 mm³, Balb/c nude mice were divided into three groups ($n = 5$ mice/group) randomly that received either Saline, RLS/HS@Au/*pGL3*, or RLS/HS/*pGL3* lipoplexes treatment. Normal saline was used as the negative control group. In each group, a dosage of 20 µg pGL3 was injected intravenously. At 24 and 48 h, D-luciferin potassium salt (150 mg/kg, 200 µL) was administered intraperitoneally and imaged using an IVIS Lumina Series III imaging system (Perkin Elmer, USA). Their ex vivo images were taken using the same settings.

### In vivo anticancer therapy

Once the HepG2 tumors reached about 100 mm³, Balb/c nude mice were randomly assigned to three groups ($n = 5$ mice/group). Saline, *p53*-loaded and *p53*/SP141 co-loaded hybrid lipoplexes were i.v. injected every 2 days for a total of seven times at a *p53* dosage of 20 µg. The width and length of the tumors and the body weight of mice were measured during the treatments. The tumor volume was calculated using the following equation: tumor volume = length × width × width/2. Mice were sacrificed at 6 days post the seventh treatment. Whole blood samples were collected and subjected to analysis of AST, ALT, BUN, and CREA for hepatic and renal function evaluation.

The tumors and major organs were collected and fixed in 4% PFA. After being sliced by a 5 µm section, the samples were subjected to further hematoxylin and eosin (H&E), immunohistology, and TUNEL and F-actin staining analysis. In vivo, tumor tissue homogenates were lysed in cell lysis buffer (100 mg tumor tissue per 1 mL cell lysis buffer), and the obtained cell lysates were used for the pull-down of GTP-RhoA and western blot of Total RhoA analysis as described above.

For survival studies, mice ($n = 6$ mice/group) were treated as above and maintained without treatments until the endpoint. The endpoint of a mouse was defined as the tumor volume reaching 1200 mm³ or the necrotic area larger than half of the tumor diameter.

In the A549 xenograft model, the same therapeutic procedure was used as that in the HepG2 xenograft model. All animals were monitored for activity, physical condition, body weight, and tumor growth. Mice were sacrificed on 4th day after the seventh treatment, and the isolated tumors were weighed.

### Evaluation of RLS/HS@Au/pDNA biosafety in mice and zebrafish

Male and female Balb/c mice (4–6 weeks) were purchased from Dossy Experimental Animals Co., Ltd. (Chengdu, China). The Balb/c mice

were divided into 3 groups ($n = 3$ female mice + 3 male mice per group), including low dose group (LD, 20 pDNA/mouse), high dose group (HD, 40 μg pDNA/mouse), and saline group. Body weight was detected from day 0 (1 day before the first injection) to the end of the study (day 31), then serum and blood were isolated for serum biochemistry estimation and hematological analysis. The major organs were isolated and sectioned for H&E staining. Mouse bone marrow was harvested for micronucleus formation analysis to assess genotoxicity.

Wild-type zebrafish were obtained from the China Zebrafish Resource Center and bred under standard environmental conditions. Zebrafish embryos were collected and subsequently characterized for developmental toxicity and teratogenicity after treatment with RLS/HS@Au/pDNA at different exposure time points. The survival rates were measured as the percentages of surviving zebrafish embryos for 96 h. The hatching rates were recorded every 24 h zebrafish embryos. At the 18-somite stage, the side-to-side contraction of the tail was observed by stereomicroscope (SteREO Discovery. V20, Zeiss). For the coiling contraction experiment, total contractions within 2 min were counted at 24 hours post fertilization (hpf) and displayed as the number of times per second (Hz).

To quantify the biodistribution of Au, 4–6 weeks-old healthy Balb/c mice were administered intravenously with RLS/HS@Au at 20 μg pDNA/mouse (seven times, one-day interval). After 15, 31, and 61 days, the mice were euthanized and the heart, liver, spleen, kidney, lung, brain, and intestine were isolated and weighed. Before elemental quantification, the tissues were digested with $HNO_3$ and $H_2O_2$ at 150 °C, the resultant Au solutions were diluted with 2% $HNO_3$ and the samples were analyzed by ICP-AES.

## Statistics and reproducibility

All data were presented as mean ± SD. All statistical analyses were performed using GraphPad Prism 9.0 software and Origin 8.0. Two-tailed Student's t test and One-way analysis of variance (ANOVA) with a Tukey post hoc test were used for the statistical comparison between the two groups and among multiple groups, respectively. The survival study was analyzed through Two-sided log-rank (Mantel-Cox) test. The probability value was represented by $p$, and $p$ values < 0.05 were considered statistically significant. The statistical methods used in each experiment and the corresponding statistical analysis results were listed in the figure legends. No statistical method was used to predetermine the sample size. No data were excluded from the analyses; The experiments were not randomized; The Investigators were not blinded to allocation during experiments and outcome assessment.

## Reporting summary

Further information on research design is available in the Nature Portfolio Reporting Summary linked to this article.

# Data availability

The NMR analysis of RLS, HA-SH, and HA can be found in Supplementary Fig. 1a and Supplementary Fig. 2a. All other data supporting the findings of this study are available within the article, supplementary information, or source data file. Source data are provided with this paper.

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

## Acknowledgements

We warmly thank Prof. Zhirong Zhang and Prof. Ernst Wagner for their insightful suggestions and engaging discussions on gene delivery. Appreciation also goes to Prof. Youqing Shen and Prof. Guowei Wang for sharing their experiences on transcytosis studies utilizing cellular and animal models, Prof. Matthias Barz and Dr. Heyang Zhang for improving lipid nanoparticles (LNPs) stability, Prof. Yingkun Guo for supports in flow cytometry and husbandry and Jiao Lu's technical expertise in CLSM and AFM detection. Our appreciation extends to Meiju Xie, Xi Wu and Jiahui Yang, Chenghui Li from the Analytical &Testing Center of Sichuan University for help with TEM, ICP-AES, and CLSM detection, respectively. This work was supported by the National Natural Science Foundation of China (NSFC, No. 81873921 to YN, 51933011 to XS and 82372027 to RJ), the Sino-German Cooperation Group Project (GZ1512 to YN), the Science and Technology Project in Sichuan Province (2023YFH0060 to YN, 25NSFSC2351 to NY and 25QNJJ4015 to RJ), the Chengdu Science and Technology Program (2020-GH02-00007-HZ to YN), Interdisciplinary innovation project from West China Hospital of Stomatology, Sichuan University (RD-03-202305 to YN).

## Author contributions

Y.N. and R.J. conceived and guided the study. X.H. and R.J. designed the experiments and provided insight into the final interpretation of the findings. X.H. and Y.W. performed the experiments and conducted data analysis. R.W. carried out the synthesis and characterization of HA-SH and AuNPs. Y.P. participated in the design of the initial schematic. X.H. and Y.W. co-wrote the manuscript. R.J., Y.N., and X.S. revised the manuscript and commented on it.

## Competing interests

The authors declare no competing interests.
