## [Peer Review File · Nature Communications]

REVIEWER COMMENTS

Reviewer #1 (nanomedicine):

In this project, the authors designed a hybrid lipoplex (RLS/HS@Au) assembled from cationic lipid (RLS), hyaluronic acid derivatives coated gold nanoparticles (HS@Au), and pDNA. They found that the “harder” RLS/HS@Au compelled cytoskeleton rearrangements by “outside-in” mechanical and “inside-out” biochemical signaling. The hybrid lipoplex impaired lysosomal degradation and further promoted autophagy/RhoA signaling-induced cytoskeleton rearrangement to alter the intracellular fate of lipoplexes. The rearrangement also enhanced intercellular delivery via transcytosis, increased the penetration in solid tumors, and achieved 20-fold higher gene expression in vivo. RLS/HS@Au was used to co-deliver p53 plasmid along with MDM2 inhibitor sp141 to hepatocellular carcinoma, showing a remarkable synergistic anti-tumor effect. The study is a complete and complicated work with new points, but the presented data can't support the conclusion very well. I recommend this paper only can be published on Nature Communication after major revisions.

1. As pDNA was anchored outside the RLS, its stability and activity may be affected during circulation.
2. In Figure 2f, the TEM images revealed that Au nanoparticles were not uniform outside the RLS. The authors should optimize the experiments to make the distribution of Au nanoparticle uniform.
3. In this project, it seems that Au nanoparticles are not the necessary or specific. Can other inorganic nanoparticles, such as mesoporous silica, MnO₂, CuS, make the PLS harder. Please compare and verify the effects of different harder nanoparticle on the PLS.
4. For the transfection and distribution studies, the photoacoustic imaging (PA) based on the Au nanoparticle should also be added.
5. The mechanism exploration on cytoskeletal remodeling by GTP-RhoA accumulation in vivo should also be investigated.
6. Antitumor effect and p53 activation function should also be investigated on different tumor models and humanized PDX model.
7. In Figure 8, the indicators of MDM2 and p53 should also be evaluated by flow cytometry.

Reviewer #2 (autophagy RhoA signaling):

The paper is well-written, and the referee appreciates the authors' efforts in explaining their strategy and interpretation of the results through schemas. However, I must disagree with their autophagy/RhoA conclusions.

The authors have observed that their RLS/HS@Au nanoparticles can hinder the lysosomal degradation process (Fig 4f) and promote the formation of actin stress fibers (Fig. 2), increasing plasmid transfection efficiency. I agree with their interpretation that the nanoparticles may interrupt the autophagy flux at the autophagosome maturation step, leading to a decrease in autophagosomes and an increase in autolysosomes (Fig 1 and Fig 4). However, this should lead to the accumulation of all autophagy substrates, including LC3-II, SQSTM1, and RHOA-GTP, within the autolysosomes. One important consequence would be the sequestration of RHOA-GTP within the autolysosomes, preventing it from cross-linking the actin cytoskeleton. This would be at odds with the observed LC3-II decrease (Fig. 4g, line 253) and stress fiber formation (Fig. 2).

Only the blockage of autophagy at the initiation step would undoubtedly result in a decrease in LC3-II levels, as observed by the authors (Fig. 4g, see comment), and an increase in RHOA-GTP accumulation, which will then be accessible and responsible for the reticulation of the cytoskeleton. However, this contradicts the observed accumulation of autolysosomes (Fig. 4i).

I suggest conducting several experiments to differentiate between blockages at the autophagy initiation or maturation steps. The blockage at the initiation step would be characterized by an accumulation of RHOA-GTP at the plasma cell membrane and reticulation of ACTIN stress fibers. On the other hand, the blockage at the maturation stage would lead to punctate accumulation and sequestration of RHOA-GTP within the autolysosome.

- To achieve this, the authors could use their nanoparticles with the LC3-EGF-mcherry reporter plasmid. They should expect to observe diffuse cytosolic yellow labeling (LC3-EGF-mcherry) when autophagy is blocked at the initiation step. Alternatively, they should expect to see punctate red labeling (LC3-mCherry) with the loss of EGFP green fluorescence at acidic pH when autophagy is blocked at the autolysosomal step.

- It would be interesting to know precisely where RhoA-GTP is located in cells when exposed to nanoparticles. I recommend using a method where RHOA is labeled after fixation with ice-cold 10% TCA for 15 minutes. This method preserves membrane-associated RHOA-GTP, not cytoplasmic RHOA-GDP (doi: 10.4161/auto.27198).

- I recommend measuring autophagic flux using E64d pepstatin or NH₄Cl instead of chloroquine. Chloroquine is too potent in inducing LC3-II and SQSTM1, making it challenging to modulate their levels with nanoparticles (Fig. 4g and Fig. 4h).

Please use the term "autophagosomes" instead of "autophagic vacuoles" (line 260)

The variations seen in the WB of LC3-2 (Fig 4g) could potentially be due to technical issues such as bubble formation during the WB transfer or SDS-Page migration process. I suggest that authors perform triplicate experiments to eliminate any potential ambiguity.

Please quantify the accumulation of autophagic vesicles by the TEM analysis of 25 cells (Only one cell is presented in Fig 4i).

Reviewer #3 (p53-MDM2, cancer therapy):

The article by Hu et al. describes the development of a hybrid lipid nanoparticle (RLS/HS@Au) designed for efficient and targeted delivery of plasmid DNA (pDNA) to solid tumors. By combining cationic lipid RLS, hyaluronic acid (HA)-derivatized gold nanoparticles (HS@Au), and pDNA, the researchers sought to mimic viral strategies that exploit cellular cytoskeleton machinery for enhanced intracellular transport. They report that the RLS/HS@Au lipoplex elicits "outside-in" mechanical signals and "inside-out" biochemical cues leading to cytoskeleton rearrangement. This novel formulation effectively impairs lysosomal degradation, thereby promoting autophagy/RhoA signaling pathways which, in turn, facilitates cytoskeleton alterations that favor transcytosis and improved penetration within solid tumors. The data are rich and the figures are informative and delicate. However, the molecular mechanism seems not known or not thoroughly investigated. Overall, the manuscript can be a tour de force and can be up to the bar for publication, only if all below issues are resolved.

Major issues:

Although the study shows enhanced gene expression and tumor-specific targeting, comprehensive toxicity profiles for RLS/HS@Au are not fully discussed. It would be beneficial to explore the potential toxicity and long-term biodistribution of the RLS/HS@Au system to ensure its safety and suitability for clinical applications. For example, a thorough assessment of cytotoxicity across multiple cell lines, along with in vivo systemic toxicity and genotoxicity tests would be crucial.

Additionally, elucidating the detailed molecular mechanisms underlying the enhancement of autophagic flux and RhoA activation due to nanoparticle interactions could further strengthen the understanding of this innovative gene delivery platform. At least the authors should discuss the possible detailed mechanisms and show the most critical evidence/s.

The study provides data on the immediate cytotoxic effects and efficacy but what is the the long-term stability of these nanoparticles and their long-term toxicological effects?

Minor issues:

How this new approach compares with current existing gene delivery systems, for example in terms of efficiency, safety, and feasibility?

In the abstract, the authors claimed that the hybrid lipoplex led to 20-fold higher gene expression with appr. 80% tissue specificity in vivo. How did you quantify the numbers?

Fig. 2g, it seems almost all the Cy5-DNA are localized in the cytosol but not the nucleus.

Fig. 3e, why the endo/lysosome intensity in RLS/HS and RLS groups are much lower than that in RLS/HS@Au group?

Fig. 3h, it is recommended to use Z-stack 3D fig to better demonstrate the Cy5-DNA intensity in the nucleus.

POINT-BY-POINT RESPONSE TO THE REVIEWER'S COMMENTS

REVIEWER COMMENTS

Reviewer #1 (nanomedicine):

In this project, the authors designed a hybrid lipoplex (RLS/HS@Au) assembled from cationic lipid (RLS), hyaluronic acid derivatives coated gold nanoparticles (HS@Au), and pDNA. They found that the “harder” RLS/HS@Au compelled cytoskeleton rearrangements by “outside-in” mechanical and “inside-out” biochemical signaling. The hybrid lipoplex impaired lysosomal degradation and further promoted autophagy/RhoA signaling-induced cytoskeleton rearrangement to alter the intracellular fate of lipoplexes. The rearrangement also enhanced intercellular delivery via transcytosis, increased the penetration in solid tumors, and achieved 20-fold higher gene expression *in vivo*. RLS/HS@Au was used to co-deliver p53 plasmid along with MDM2 inhibitor sp141 to hepatocellular carcinoma, showing a remarkable synergistic anti-tumor effect. The study is a complete and complicated work with new points, but the presented data can't support the conclusion very well. I recommend this paper only can be published on Nature Communication after major revisions.

> We greatly appreciate the positive comments on our work and giving us the chance to improve our manuscript.

Q1. As pDNA was anchored outside the RLS, its stability and activity may be affected during circulation.

A1: Thank you very much for raising this question. We fully agree with the reviewer that the stability and activity of RLS/DNA lipoplexes might be challenged during circulation. Thus, in our preparation, the RLS/DNA lipoplexes were coated with hyaluronic acid-decorated gold nanoparticles (HS@Au), which could neutralize the positive charge of the lipoplexes and provide more protection to genes^{1,2}.

According to your question, we conducted more experiments to confirm the stability and activity of our hybrid lipoplexes. The investigation included the integrality of pDNA in the presence of DNase (Supplementary Fig. 7), size and ζ potential changes under harsher conditions (10% FBS culture medium, PBS buffer (pH 7.4), and 10% plasma, Supplementary Fig. 8), as well as gene transfection efficiency in high concentrated serum (10%-50%, Fig.3a, b and Supplementary Fig.12)³. The corresponding revision have been made in the main text including results and method (page 6, line 111; page 9, line 161; page 42, line 708; page 44, line 750) and Supplementary Information with a bright yellow mark.

From the above data, we could find that when the RLS/DNA lipoplexes were coated with hyaluronic acid-decorated gold nanoparticles (HS@Au), their stability and transfection activity was improved, benefiting the stability and activity of lipoplexes during circulation, and resulting in the final promotion of *in vivo* gene transfection (Fig. 6a, b, e).

Supplementary Figure 7. Resistance capacity of various lipoplexes (RLS, RLS/HS, RLS/HS@Au,

RLS/HS@Au_{1e}, RLS/sp141, RLS/HS@Au/sp141) against DNase indicated by agarose gel electrophoresis. Different lipoplexes were incubated with DNase (200 U/mL) and/or heparin (4 mg/mL).

Fig. 2 d Particle size and ζ potential changes of different lipoplexes in the culture medium with or without 10% FBS for 24 h.

Supplementary Figure 8. Size changes of different nanoparticles (RLS, RLS/HS, and RLS/HS@Au) by DLS measurement in the indicated conditions, including PBS buffer (pH 7.4), culture medium with 10% FBS, and 10% plasma.

Fig. 3 pEGFP transfection of RLS/HS@Au and RLS/HS on HepG2 cells in 10% FBS culture medium. **a** Evaluation of pEGFP transfection on HepG2 cells by fluorescence microscopy images and **b** corresponding semi-quantitative analysis of the mean fluorescence intensity (MFI) using ImageJ software. The scale bar is 200 μ m. Data are mean \pm S.D. * p < 0.05.

Supplementary Figure 12. *In vitro* transfection of RLS, RLS/HS, and RLS/HS@Au lipoplexes on HepG2 cells in culture medium with different serum concentrations. Fluorescence microscopy images of the pEGFP transfection in 0% (a), 20% (c), and 50% (e) serum. The corresponding semi-quantitative analysis of the mean fluorescence MFI (b, d, and f) was performed by ImageJ software. The scale bars are 200 μm . Data are mean \pm S.D. * $p < 0.05$.

Fig. 6 *In vivo* transfection of hybrid lipoplexes in tumor model after intravenous injections (i.v.). **a** The pGL3 transfection of hybrid lipoplexes (RLS/HS@Au) in subcutaneous HepG2 xenograft mice, evaluated by bioluminescence imaging from an *in vivo* imaging system at 24 and 48 h after intraperitoneal administration of D-luciferin. (n = 5). **b** Representative luciferase expression in dissected organs of mice at 48 h after intravenous injection of different lipoplexes (20 μg pGL3/mouse). **e** Bioluminescence quantification in major organs after administration of saline, RLS/HS/pGL3, and RLS/HS@Au/pGL3, respectively.

1. He Y, Cheng G, Xie L, Nie Y, He B, Gu Z. Polyethyleneimine/DNA polyplexes with reduction-sensitive hyaluronic acid derivatives shielding for targeted gene delivery. *Biomaterials* **34**, 1235-1245 (2013).
2. He Y, Nie Y, Cheng G, Xie L, Shen Y, Gu Z. Viral mimicking ternary polyplexes: a reduction-controlled hierarchical unpacking vector for gene delivery. *Adv Mater* **26**, 1534-1540 (2014).
3. Jiang Q, *et al.* Specially-Made Lipid-Based Assemblies for Improving Transmembrane Gene Delivery: Comparison of Basic Amino Acid Residue Rich Periphery. *Mol Pharm* **13**, 1809-1821 (2016).

Q2. In Figure 2f, the TEM images revealed that Au nanoparticles were not uniform outside the RLS. The authors should optimize the experiments to make the distribution of Au nanoparticle uniform.

A2: Thank you for this helpful suggestion. In general, non-uniform distribution is a common phenomenon in hybrid nanoparticles, which has been reported by other researchers^{1,2,3}.

We have tried the microfluidic technology to mix the multiple components of RLS/HS@Au/DNA, to improve the uniformity of the lipid nanoparticles. From the TEM observation (Fig. R1), we could find that the uniformity of gold distribution has improved, but it is still difficult to achieve a perfect one. Compared to the preparation through electrostatic interaction, the microfluidic technology resulted in a vague profile of gold nanoparticles, which may result from more encapsulation of the gold nanoparticles inside the lipid nanoparticles.

Further optimization of the uniformity is also the direction of our research, in order to promote this kind of nano-delivery system for clinical translation.

Fig. R1 The preparation of RLS/HS@Au by microfluidic technology. **a** TEM morphology of RLS/HS@Au. Arrows represent gold nanoparticles inside the lipid nanoparticles. The scale bar is 100 nm. **b** A microfluidic system.

1. Hu Y, Wen C, Song L, Zhao N, Xu FJ. Multifunctional hetero-nanostructures of hydroxyl-rich polycation wrapped cellulose-gold hybrids for combined cancer therapy. *J Control Release* **255**, 154-163 (2017).
2. Ding X, *et al.* Designing Aptamer-Gold Nanoparticle-Loaded pH-Sensitive Liposomes Encapsulate Morin for Treating Cancer. *Nanoscale Res Lett* **15**, 68 (2020).
3. Larrea A, Clemente A, Luque-Michel E, Sebastian V. Efficient production of hybrid bio-nanomaterials by continuous microchannel emulsification: Dye-doped SiO₂ and Au-PLGA nanoparticles. *Chemical Engineering Journal* **316**, 663-672 (2017).

Q3. In this project, it seems that Au nanoparticles are not the necessary or specific. Can other inorganic nanoparticles, such as mesoporous silica, MnO₂, CuS, make the RLS harder. Please compare and verify the effects of different harder nanoparticle on the RLS.

A3: This is an important comment. Our research group has been engaged in the design of gene delivery nanocarriers for about 15 years¹⁻⁶ and has used iron oxide⁷, carbon dots⁸, cerium oxide (unpublished data), and gold nanoparticles for hybridization. The iron oxide and carbon dots have been confirmed to promote gene delivery mainly due to the rise of the extra magnetic field-induced mechanical force and photothermally-induced temperature. Before this study, we did not consider the “hardness” of the inorganic particles.

We fully agree with your opinion that other inorganic nanoparticles, such as mesoporous silica (MSN), MnO₂, and CuS, could make the RLS harder. The “outside-in” mechanical signal from all these harder nanoparticles might be the same. Thus, we determined the hardness of each kind of nanoparticle after the incorporation of different inorganic nanoparticles (Oleic acid-modified cerium dioxide, OA@CeO₂; Oleic

acid-modified iron tetroxide, OA@Fe₃O₄; mesoporous silica, MSN) in RLS, showing expected hardness enhancement (Fig. R2) consistent with reported data in the literature^{9,10}.

But such "hardened" nanoparticles do not always result in increased transfection (Fig. R3). The addition of OA@CeO₂ (Fig. R3a, c) and OA@Fe₃O₄ (Fig. R3b, d) could promote gene transfection of RLS at appropriate concentrations, but MSN did not (Fig. R3b, e). It is probably because not all hybrid particles can cause the dual signals of mechanical "outside-in" and biochemical "inside-out" as explored in the study, due to the varied chemical ingredients of the particles. In our future work, more inorganic particles will be compared and investigated for their different biochemical signals inside the cells.

Fig. R2 Morphology and Young's modulus of RLS, RLS/OA@CeO₂, RLS/OA@Fe₃O₄, and RLS/MSN measured by atomic force microscope (AFM). The scale bar is 100 nm.

Fig. R3 Gene transfection of differently “hardened” hybrid nanoparticles. **a** Fluorescence microscopy images of pEGFP and EGFP mRNA transfection of RLS/OA@CeO₂ on SH-SY5Y and BV2 cells (unpublished data). **b** Fluorescence microscopy images of pEGFP transfection of RLS/OA@Fe₃O₄ and RLS/MSN on HepG2 cells (unpublished data). The scale bar is 200 μm. **c** Semi-quantitative analysis of the mean fluorescence intensity (MFI) from the microscopy images (a) of pEGFP and EGFP mRNA transfection of RLS/OA@CeO₂ on SH-SY5Y and BV2 cells using ImageJ software. **d,e** Semi-quantitative analysis of the mean fluorescence intensity (MFI) from the microscopy images (b) of pEGFP transfection of RLS/OA@Fe₃O₄ and RLS/MSN on HepG2 cells using ImageJ software. **p* < 0.05, and NS means no significance.

1. Nie Y, Zhang ZR, Duan YR. Combined use of polycationic peptide and biodegradable macromolecular polymer as a novel gene delivery system: a preliminary study. *Drug Deliv* **13**, 441-446 (2006).
2. Nie Y, Günther M, Gu Z, Wagner E. Pyridylhydrazone-based PEGylation for pH-reversible lipopolyplex shielding. *Biomaterials* **32**, 858-869 (2011).
3. Nie Y, Schaffert D, Rödl W, Ogris M, Wagner E, Günther M. Dual-targeted polyplexes: one step towards a synthetic virus for cancer gene therapy. *J Control Release* **152**, 127-134 (2011).
4. He Y, Nie Y, Cheng G, Xie L, Shen Y, Gu Z. Viral mimicking ternary polyplexes: a reduction-controlled hierarchical unpacking vector for gene delivery. *Adv Mater* **26**, 1534-1540 (2014).
5. Bi Q, *et al.* Mucus-penetrating nonviral gene vaccine processed in the epithelium for inducing advanced vaginal mucosal immune responses. *Acta Pharm Sin B* **13**, 1287-1302 (2023).
6. Zhao Y, *et al.* A DNA vaccine (EG95-PT1/2/3-IL2) encoding multi-epitope antigen and IL-2 provokes efficient and long-term immunity to echinococcosis. *J Control Release* **361**, 402-416 (2023).
7. Hu A, *et al.* A parallel and cascade control system: magnetofection of miR125b for synergistic tumor-association macrophage polarization regulation and tumor cell suppression in breast cancer treatment. *Nanoscale* **12**, 22615-22627 (2020).
8. Luo T, *et al.* Iron doped carbon dots based nanohybrids as a tetramodal imaging agent for gene delivery promotion and photothermal-chemodynamic cancer synergistic theranostics. *Materials & Design* **208**, (2021).
9. Zhou C, *et al.* Lipid-coated nano-calcium-phosphate (LNCP) for gene delivery. *Int J Pharm* **392**, 201-208 (2010).
10. Zou D, *et al.* Nanoparticle elasticity regulates the formation of cell membrane-coated nanoparticles and their nano-bio interactions. *Proc Natl Acad Sci U S A* **120**, e2214757120 (2023).

Q4. For the transfection and distribution studies, the photoacoustic imaging (PAI) based on the Au nanoparticle should also be added.

A4: Thanks for your kind suggestions. We have performed photoacoustic imaging (PAI), but no obvious signal could be detected *in vitro* likely due to the small size of AuNPs (46 nm) (Fig. R4).

According to the reference¹, the efficiency of photoacoustic imaging (PAI) depends on the generation of an acoustic wave from the absorption of near-infrared (NIR) energy by endogenous chromophores or exogenous contrast agents. The ultrasound wave can be detected with a transducer that converts the acoustic waves to electric signals, which are then processed into a visualized image. A high visible contrast can only be obtained with particles absorbing in the biological window (650-1300 nm) *in vivo*, which could offer PA signals with a relatively low background and allow for deeper penetration².

Due to the strong and tunable optical absorption that results from the surface plasmon resonance (SPR) effect of AuNPs, they have been widely used as PA contrast agents in recent years. However, not all gold nanoparticles can trigger a strong photoacoustic signal. For the spherical AuNPs, only the large size (> 50 nm) has strong NIR absorption and could achieve deep penetration and distinct PA effect³. AuNPs in this study were ~46 nm, which is just at the threshold. The optimal absorption spectra of HS@Au were around 530 nm (Supplementary Fig. 4), which is not in the near-infrared (NIR) optical window (650 ~ 1300 nm). As a result, no obvious PA signal was detected.

Fig. R4 *In vitro* photoacoustic imaging of HS@Au at different concentrations (10, 50, and 100 µg/mL).

1. Li W, Chen X. Gold nanoparticles for photoacoustic imaging. *Nanomedicine (Lond)* **10**, 299-320 (2015).
2. Chen Y-S, Zhao Y, Yoon SJ, Gambhir SS, Emelianov S. Miniature gold nanorods for photoacoustic molecular imaging in the second near-infrared optical window. *Nature Nanotechnology* **14**, 465-472 (2019).
3. Cheng X, Sun R, Yin L, Chai Z, Shi H, Gao M. Light-Triggered Assembly of Gold Nanoparticles for Photothermal Therapy and Photoacoustic Imaging of Tumors *In Vivo*. *Adv Mater* **29**, (2017).

Q5: The mechanism exploration on cytoskeletal remodeling by GTP-RhoA accumulation *in vivo* should also be investigated.

A5: As per your suggestion, we have made a pull-down analysis of GTP-RhoA (Fig. 8f) and an immunostaining analysis of F-actin (Fig. 8g) in the HepG2 tumor-bearing animal model. The RLS/HS@Au induced GTP-RhoA accumulation as well as cytoskeleton remodeling *in vivo*, consistent with the *in vitro* data. The corresponding revision (in results and method) has been made in the main text with a bright yellow mark (page 27, line 441; page 31, line 472; page 52, line 926).

Fig. 8 f Pull-down assay of GTP-RhoA in HepG2 tumor-bearing mice treated with tail vein injections of RLS/HS@Au/pDNA (pDNA 20 µg/mouse). The β-actin was used as a loading control, and GTPγs was the positive control. **g** Immunofluorescence staining of F-actin (green) in HepG2 tumors sections. The scale bar is 20 µm.

Q6: Antitumor effect and p53 activation function should also be investigated on different tumor models and humanized PDX model.

A6: Thank you for the valuable suggestion. As per your suggestion, we have evaluated the cytotoxicity of RLS/HS@Au/sp141/p53 on other three kinds of cell lines (A549, MCF-7, and B16 cells) using CCK-8 assay. The co-delivery of p53 plasmid and sp141 showed better synergistic anti-tumor efficacy on MCF-7 and A549 (Fig. R5), but less effective for B16 cells. It might be because B16 cells with lower endogenous MDM2 expression were less sensitive to the combination of p53 and sp141^{1,2}.

Then, we established an A549 xenograft tumor model in Balb/c nude mice for the evaluation of the synergistic treatment of RLS/HS@Au/p53/sp141. The *i.v.* injection of saline or RLS/HS@Au/p53 for seven cycles was used as a comparison. At the end of treatment (on the 16th day), the combination formulations

(RLS/HS@Au/sp141/p53) showed a more efficient tumor inhibition (86.2%) than RLS/HS@Au/p53 (31.2%) (Fig. 8h, i and Supplementary Fig. 33). The corresponding revision (in results and method) has been made in the main text (page 27, line 443; page 52, line 932) and Supplementary Information with bright yellow mark. In addition, we have communicated with our collaborators at West China Hospital of Sichuan University, and found that the construction of the PDX model generally takes more than 1 year^{3,4}, including 2-4 months for tumor extraction from patient and mice transplantation, as well as 2-3 months for treatment and evaluation. It might impact the innovation and timeliness of the manuscript.

Fig. R5 The cytotoxicity of various lipoplexes on MCF-7, A549, and B16 cells measured by CCK-8 assay. The dosage of p53 plasmid and sp141 were 2 $\mu\text{g}/\text{mL}$ and 1 $\mu\text{mol}/\text{L}$, respectively. Data are mean \pm S.D.

Fig. 8 Anticancer efficacy of hybrid lipoplexes (RLS/HS@Au/sp141/p53) in A549 tumor-bearing Balb/c nude mice. **h** Tumor growth curves of A549 tumor-bearing Balb/c nude mice receiving hybrid lipoplexes treatments. **i** Photographs of the excised tumors from different treatment groups. The scale bar is 2 cm.

Supplementary Figure 33. **a** Establishment of the subcutaneous A549 tumor model and dosing regimen. **b** The weight of the A549 tumors excised from different treatment groups. * $p < 0.05$. **c** Body weight change of Balb/c nude mice with A549 xenografts in different groups. Data are mean \pm S.D.

- Melnikova VO, Bolshakov SV, Walker C, Ananthaswamy HN. Genomic alterations in spontaneous and carcinogen-induced murine melanoma cell lines. *Oncogene* 23, 2347-2356 (2004).
- Wang W, et al. A novel inhibitor of MDM2 oncogene blocks metastasis of hepatocellular carcinoma and overcomes chemoresistance. *Genes Dis* 6, 419-430 (2019).
- Jung J, Seol HS, Chang S. The Generation and Application of Patient-Derived Xenograft Model for Cancer Research. *Cancer Res Treat* 50, 1-10 (2018).
- Yoshida GJ. Applications of patient-derived tumor xenograft models and tumor organoids. *J Hematol Oncol* 13, 4 (2020).

Q7. In Figure 8, the indicators of MDM2 and p53 should also be evaluated by flow cytometry.

A7: Many thanks for this suggestion. Flow cytometry (FCM) is a specialized cell analysis technique developed on the basis of immunofluorescence technology, which is the most representative and advanced instrument for analyzing cells. Thus, we conducted an *in vivo* flow cytometry analysis of p53 and MDM2. Unfortunately, no specific fluorescent signals were detected. We speculated that it might be because the antibodies against p53 not only target human tumor tissues but also mouse tissues, thus leading to undesired signals. p53 is conserved across multiple species, the antibodies against which are difficult to specifically differentiate between human and murine p53 proteins, resulting in more non-specific adsorption (non-tumor cells). Moreover, we consulted the manufacturers and were informed that their products were not validated for *in vivo* flow cytometry analysis.

Instead, we quantified the expression of p53 and MDM2 in the A549 xenograft model by western blot. Consistent with the *in vitro* data (Fig. 7b), p53 expression was elevated and MDM2 expression was decreased in the combination therapy group (Fig. R6).

Fig. R6 The expression of MDM2 and p53 proteins in tumor homogenates from the A549 xenograft tumors by western blot.

Reviewer #2 (autophagy RhoA signaling):

The paper is well-written, and the referee appreciates the authors' efforts in explaining their strategy and interpretation of the results through schemas. However, I must disagree with their autophagy/RhoA conclusions.

The authors have observed that their RLS/HS@Au nanoparticles can hinder the lysosomal degradation process (Fig 4f) and promote the formation of actin stress fibers (Fig. 2), increasing plasmid transfection efficiency. I agree with their interpretation that the nanoparticles may interrupt the autophagy flux at the autophagosome maturation step, leading to a decrease in autophagosomes and an increase in autolysosomes (Fig 1 and Fig 4). However, this should lead to the accumulation of all autophagy substrates, including LC3-II, SQSTM1, and RHOA-GTP, within the autolysosomes. One important consequence would be the sequestration of RHOA-GTP within the autolysosomes, preventing it from cross-linking the actin cytoskeleton. This would be at odds with the observed LC3-II decrease (Fig. 4g, line 253) and stress fiber formation (Fig. 2).

Only the blockage of autophagy at the initiation step would undoubtedly result in a decrease in LC3-II levels, as observed by the authors (Fig. 4g, see comment), and an increase in RHOA-GTP accumulation, which will then be accessible and responsible for the reticulation of the cytoskeleton. However, this contradicts the observed accumulation of autolysosomes (Fig. 4i).

> We greatly appreciate such important and professional comments on autophagy RhoA signaling, which help us further improve our manuscript and clear the contradicts.

Q1: I suggest conducting several experiments to differentiate between blockages at the autophagy initiation or maturation steps. The blockage at the initiation step would be characterized by an accumulation of RHOA-GTP at the plasma cell membrane and reticulation of ACTIN stress fibers. On the other hand, the blockage at the maturation stage would lead to punctate accumulation and sequestration of RHOA-GTP within the autolysosome.

To achieve this, the authors could use their nanoparticles with the LC3-EGFP-mCherry reporter plasmid. They should expect to observe diffuse cytosolic yellow labeling (LC3-EGFP-mCherry) when autophagy is blocked at the initiation step. Alternatively, they should expect to see punctate red labeling (LC3-mCherry) with the loss of EGFP green fluorescence at acidic pH when autophagy is blocked at the autolysosome step.

It would be interesting to know precisely where RhoA-GTP is located in cells when exposed to nanoparticles. I recommend using a method where RHOA is labeled after fixation with ice-cold 10% TCA for 15 minutes. This method preserves membrane-associated RHOA-GTP, not cytoplasmic RHOA-GDP (doi: 10.4161/auto.27198).

A1: Thank you for your valuable suggestions, which strongly help improve this work. As you suggested, we conducted several experiments and tried to differentiate between blockages at the autophagy initiation or maturation steps, as well as to analyze the distribution of GTP-RhoA inside cells. We performed cellular transfection of LC3-EGFP-mCherry reporter plasmid (Fig. 4e, f and Supplementary Fig.19), localization of GTP-RhoA (Fig. 4b and Supplementary Fig. 17), and more observations of autophagic flux in TEM (Supplementary Fig. 20, 21, more details in Q5).

We could observe diffuse cytosolic yellow labeling (LC3-EGFP-mCherry) and punctate red labeling (LC3-mCherry), after the transfection of the LC3-EGFP-mCherry reporter plasmid with RLS/HS@Au. It was found that the number of yellow puncta (mCherry⁺EGFP⁺) in the RLS/HS@Au group was similar with the nonhybrid group, and the red puncta (mCherry⁺EGFP⁻) decreased (Supplementary Fig.19). This result

seems to be inconsistent with the decrease in autophagosomes and increase in autolysosomes from the following TEM observation (Fig. 4g and Supplementary Fig. 20, 21). We speculated that the differences might result from the alkalization of autolysosomes by AuNP¹ (Fig. 4i), leading to the failure of EGFP quenching^{2,3}. In other words, the observed yellow puncta in the RLS/HS@Au group included both autophagosomes and autolysosomes, while the true number of autolysosomes was difficult to estimate from the red fluorescence. It indicated the reduced autophagosomes after RLS/HS@Au treatment, which was further confirmed by the significant decrease of yellow puncta (mCherry⁺EGFP⁺) under starvation (Fig. 4e, f). Anyway, the results from transfection of LC3-EGFP-mCherry reporter plasmid indicated that RLS/HS@Au might compromise autophagic flux at the initial stage.

Further, after fixation with ice-cold TCA as you suggested, more membrane-associated GTP-RhoA were observed in RLS/HS@Au and HS@Au groups than in the nonhybrid RLS group (Fig. 4b). It was concluded that the blockage of the initiation autophagy by AuNP would result in an increase in GTP-RhoA accumulation in the plasma membrane, further induce cytoskeletal remodeling and benefit gene delivery. Prof. Liang's group¹ reported similar results with us (Fig. 4h), that the AuNPs could inhibit autophagy initiation with decreased LC3-II accumulation compared to the control group in the presence of chloroquine. However, the mechanism behind AuNPs-mediated suppression of autophagy initiation remains unclear. It didn't affect the activation of mTORC1/p70S6K signaling¹ (ACS Nano, 2011; 5(11):8629). So, the mechanism of AuNPs-mediated suppression of autophagy initiation will be a focus of our future investigations.

Interestingly, RLS/HS@Au compelled more membrane-associated GTP-RhoA than HS@Au groups (Fig 4b), although HS@Au induced more GTP-RhoA accumulation in pull-down assay (Fig 4a). We speculated that RLS might help more GTP-RhoA escape from lysosome because it can interfere with membrane fluidity and penetrate membrane⁴. In addition, the AuNPs accumulation in lysosomes was verified to induce lysosomal pH alkalization¹, which impairs lysosome degradation capacity. The impairment of lysosomes by AuNPs and the transmembrane function of RLS might together provide an opportunity for autolysosome escape of GTP-RhoA, preventing degradation.

Thus, in our opinion, RLS/HS@Au may increase the membrane distribution of GTP-RhoA in two ways: by blocking autophagy initiation responsible for GTP-RhoA sequestration and by aiding the escape of sequestered GTP-RhoA from autolysosomes through pH alkalization and interference with lysosomal membrane fluidity.

The related revision (in results and method) has been made in the main text (page 14, line 233; page 14, line 248; page 18, line 289; page 18, line 293; page 46, line 804; page 47, line 821) and Supplementary Information with bright yellow mark.

Supplementary Figure 19. a Confocal image of HepG2 cells expressing LC3-EGFP-mCherry after treatment with various gene lipoplexes for 24 and 48 h. Green tunnel: LC3-EGFP; Red tunnel: LC3-mCherry; Yellow tunnel: the merge of EGFP and mCherry, Blue: DAPI. The scale bar is 10 μm. **b** Mean number of yellow puncta and red puncta.

Fig. 4 Monitoring of the autophagic flux in the starvation conditions. **e** Confocal images of HepG2 cells expressing LC3-EGFP-mCherry after treatment with various gene lipoplexes for 24 and 48 h in starvation conditions. HepG2 cells were cultured in EBSS buffer for 5 h after lipoplexes treatment. Green tunnel: LC3-EGFP; Red tunnel: LC3-mCherry; Yellow tunnel: the merge of EGFP and mCherry, Blue: DAPI. The scale bar is 10 μm. **f** Mean number of yellow puncta and red puncta.

Fig. 4 b Immunofluorescence staining of RhoA in HepG2 cells fixed with 10% ice-cold TCA after treatment with various lipoplexes (1 μg pDNA/well) or free HS@Au (50 μg/mL) for 24 h. Green: RhoA, Blue: DAPI. The scale bar is 10 μm.

Supplementary Figure 17. The mean fluorescence intensity of RhoA was quantified by gray intensity analysis in ImageJ software. Data are mean ± S.D. * $p < 0.05$.

1. Ma X, *et al.* Gold nanoparticles induce autophagosome accumulation through size-dependent nanoparticle uptake and lysosome impairment. *ACS Nano* **5**, 8629-8639 (2011).
2. Klionsky DJ, *et al.* Guidelines for the use and interpretation of assays for monitoring autophagy (4th edition)(1). *Autophagy* **17**, 1-382 (2021).
3. Lee JH, *et al.* Faulty autolysosome acidification in Alzheimer's disease mouse models induces autophagic build-up of A β in neurons, yielding senile plaques. *Nat Neurosci* **25**, 688-701 (2022).
4. Wei Y, *et al.* A cationic lipid with advanced membrane fusion performance for pDNA and mRNA delivery. *J Mater Chem B* **11**, 2095-2107 (2023).

Q2: I recommend measuring autophagic flux using E64d pepstatin or NH₄Cl instead of chloroquine. Chloroquine is too potent in inducing LC3-II and SQSTM1, making it challenging to modulate their levels with nanoparticles (Fig. 4g and Fig. 4h).

A2: Many thanks for your important suggestion. According to that, we measured the disturbance of autophagic flux by RLS/HS@Au using NH₄Cl instead of chloroquine. The corresponding revision has been made in the main text and Supplementary Information with a bright yellow mark. More details were shown as follows:

"Next, the expression levels of LC3-II and p62, two well-known autophagy indicators, were detected with or without autophagy inhibitors chloroquine (CQ) and NH₄Cl (Fig. 4h, Supplementary Fig. 22, 23). The hybridization would induce LC3-II and p62 accumulation without inhibitors. Under the blockage of CQ and NH₄Cl, the RLS/HS@Au treated group displayed decreased accumulation of LC3-II compared with the untreated and non-hybrid treated group (Fig. 4h and Supplementary Fig.22, 23) ..." (page 15, line 261).

Supplementary Figure 23. Western blot analysis of autophagy-related proteins in HepG2 cells treated with various lipoplexes (at DNA of 1 μ g/mL) in the absence or presence of autophagic flux inhibitor NH₄Cl (10 mmol/L) for 24 h. The β -actin was used as a loading control.

Q3: Please use the term "autophagosomes" instead of "autophagic vacuoles" (line 260)

A3: Much appreciate your important suggestion. In the revision, we have used the term "autophagosomes" instead of "autophagic vacuoles" (page 15, line 258; page 18, line 300).

Q4: The variations seen in the WB of LC3-2 (Fig 4g) could potentially be due to technical issues such as bubble formation during the WB transfer or SDS-Page migration process. I suggest that authors perform triplicate experiments to eliminate any potential ambiguity.

A4: Thank you for your suggestion. There could potentially be technical issues such as bubble formation during the WB transfer or SDS-Page migration process. We improved our experimental technique and repeated the western blot of LC3-II three times (Fig. 4h, Supplementary Fig. 22, and Fig. R7), and replaced the previous figure with a new figure.

Fig. 4 h Western blot analysis of autophagy-related proteins in HepG2 cells treated with various lipoplexes (at DNA of 1 $\mu\text{g}/\text{mL}$) in the absence or presence of autophagic flux inhibitor chloroquine (CQ, 10 $\mu\text{mol}/\text{L}$) for 24 h. The β -actin was used as a loading control.

Supplementary Figure 22. The expression levels of LC3-II and p62 were quantified via gray intensity analysis (normalized to β -actin). Data from three independent experiments are expressed as mean \pm S.D.

Fig. R7 Original western blot data of LC3-II, p62, and β -actin with three triplicate experiments.

Q5: Please quantify the accumulation of autophagic vesicles by the TEM analysis of 25 cells (Only one cell is presented in Fig 4i).

A5: We appreciate the valuable advice for TEM analysis. Please see the additional images of another 24 different cells in the Supporting Information (Supplementary Fig. 20, 21), which were also shown as follows:

Control

RLS/HS@Au

HS@Au

Supplementary Figure 20. TEM observation of autophagic flux induced by RLS/HS@Au at DNA of 1 $\mu\text{g}/\text{mL}$ or free HS@Au (50 $\mu\text{g Au}/\text{mL}$) on HepG2 cells. The white and yellow arrows indicated autophagosomes and autolysosomes, respectively. The scale bars are 1 μm and 500 nm, respectively.

Supplementary Figure 21. The average number of autophagosomes and autolysosomes calculated from 25 cells in each group.

Reviewer #3 (p53-MDM2, cancer therapy):

The article by Hu et al. describes the development of a hybrid lipid nanoparticle (RLS/HS@Au) designed for efficient and targeted delivery of plasmid DNA (pDNA) to solid tumors. By combining cationic lipid RLS, hyaluronic acid (HA)-derivatized gold nanoparticles (HS@Au), and pDNA, the researchers sought to mimic viral strategies that exploit cellular cytoskeleton machinery for enhanced intracellular transport. They report that the RLS/HS@Au lipoplex elicits “outside-in” mechanical signals and “inside-out” biochemical cues leading to cytoskeleton rearrangement. This novel formulation effectively impairs lysosomal degradation, thereby promoting autophagy/RhoA signaling pathways which, in turn, facilitates cytoskeleton alterations that favor transcytosis and improved penetration within solid tumors. The data are rich and the figures are informative and delicate. However, the molecular mechanism seems not known or not thoroughly investigated. Overall, the manuscript can be a tour de force and can be up to the bar for publication, only if all below issues are resolved.

> We greatly appreciate the positive comments on our research work, and the suggestion which help us to improve the manuscript.

Major issues:

Q1: Although the study shows enhanced gene expression and tumor-specific targeting, comprehensive toxicity profiles for RLS/HS@Au are not fully discussed. It would be beneficial to explore the potential toxicity and long-term biodistribution of the RLS/HS@Au system to ensure its safety and suitability for clinical applications. For example, a thorough assessment of cytotoxicity across multiple cell lines, along with *in vivo* systemic toxicity and genotoxicity tests would be crucial.

A1: We appreciate your positive comments on our enhanced gene expression and targeting, and thanks for your expectation of the possibility of clinical application.

As suggested by the reviewer, safety is the most pressing issue concerning the clinical application of therapeutic agents. Here, we have added a thorough assessment of cytotoxicity across multiple cell lines (Supplementary Fig. 14), *in vivo* long-term systemic toxicity and genotoxicity tests (Supplementary Fig. 37 and Supplementary Table 2) in Balb/c mice, as well as a long-term *in vivo* biodistribution of the RLS/HS@Au system by ICP-AES analysis of the gold content in various organ tissues (Supplementary

Fig. 39). At the same time, we provided development toxicity and teratogenicity in zebrafish embryos (Supplementary Fig. 38 and Supplementary Table 3). The corresponding revision (in results and method) has been made in the main text (page 32, line 486; page 52, line 937) and Supplementary Information with bright yellow mark.

RLS/HS@Au lipoplexes demonstrated acceptable cytocompatibility in HepG2, HeLa, 4T1, and NIH-3T3 cells (Supplementary Fig.14).

In terms of mice, we repeated intravenous injection of RLS/HS@Au/pDNA (seven times, at 1-day intervals) to investigate any severe tissue damage and explore the biodistribution of RLS/HS@Au. It was found that the blood biochemical indexes were kept at normal levels (Supplementary Table 2), after administration of either a low or high dose of RLS/HS@Au with 20 or 40 μg pDNA. There were no obvious pathological changes after hybrid lipoplex treatment (Supplementary Fig. 37c, d), demonstrating no irreversible damage and no severe systemic toxicity. We evaluated the *in vivo* mouse micronucleus assay of RLS/HS@Au, and found no statistical differences in micronucleus formation rate in each group (Supplementary Fig. 37e, f, g), suggesting no significant genotoxicity.

The long-term biodistribution of RLS/HS@Au/pDNA was indicated by gold content in various tissues (Supplementary Fig. 39). It showed that about 46% of gold nanoparticles still mainly accumulated in the liver and spleen at 61 days, suggesting the slow metabolism and excretion of RLS/HS@Au through liver and spleen.

Besides, there were no significant differences in survival, hatching rates, and the spontaneous contraction of zebrafish after cocubation with RLS/HS@Au or non-treated controls (Supplementary Fig. 38 and Supplementary Table 3).

All the above data indicated an acceptable biocompatibility of RLS/HS@Au nanoparticle.

Supplementary Figure 14. Assessment of cytotoxicity of hybrid lipoplexes on multiple cell lines. Cytotoxicity of various gene lipoplexes in HepG2 (a), HeLa(b), 4T1 (c), and NIH-3T3 (d) cells with different Au to pDNA ratios (w/w). The N/P ratio was fixed at 20 and the final concentration of pEGFP was fixed at 2 $\mu\text{g}/\text{mL}$. Data are mean \pm S.D.

Supplementary Figure 37. Evaluation of biosafety of hybrid lipoplexes in Balb/c mice. **a** Schematic illustration of safety evaluation of RLS/HS@Au in mice. Balb/c mice were randomly divided into 3 groups, including saline-treatment control (Saline), low-dose (20 µg pDNA/mouse, LD), and high-dose (40 µg pDNA/mouse, HD) treated groups. Each group contained 3 female and 3 male mice. RLS/HS@Au/pDNA was intravenously injected seven times on days 1, 3, 5, 7, 9, 11, and 13. **b** Body weight changes of female and male mice throughout the study. **c,d** Representative images of H&E stained organ sections isolated from female mice (**c**) and male mice (**d**) on day 31. The scale bar is 50 µm. **e,f,g** Representative microscopic images of micronuclei formation from female mice (**e**) and male mice (**f**) on day 31, and corresponding quantification of the percentage of micronucleus (**g**) from 200 polychromatic erythrocytes (PCE) in each group. Red arrows represent the micronucleus formed in the PCE. The scale bar is 20 µm. NS means no significance, Data are mean ± S.D. **h, i** Blood biochemistry analysis of ALT, AST, CREA, and UREA in serum of female mice (**h**) and male mice (**i**) on day 31. (n = 3)

Supplementary table 2. Hematological parameters of mice treated with RLS/HS@Au at the termination of the study.

Parameters	Groups (Female)			Groups (Male)			Reference value
	Control	LD	HD	Control	LD	HD	
WBC ($\times 10^9/L$)	4.46 \pm 1.05	4.72 \pm 0.48	5.68 \pm 2.54	4.43 \pm 0.70	5.15 \pm 1.05	7.45 \pm 3.83	0.80 - 10.60
Neu%	23.10 \pm 3.40	23.87 \pm 2.31	22.7 \pm 3.54	47.63 \pm 7.00	32.03 \pm 7.03	40.30 \pm 3.89	6.5 - 50.0
Lym%	71.47 \pm 4.63	69.57 \pm 4.42	76.1 \pm 1.02	48.77 \pm 6.50	63.97 \pm 6.88	55.43 \pm 5.40	40.0 - 92.0
Mon%	4.07 \pm 1.43	2.20 \pm 0.14	3.57 \pm 1.60	2.70 \pm 0.64	2.67 \pm 0.31	3.1 \pm 0.51	0.9 - 18.0
Eos%	1.37 \pm 0.52	1.00 \pm 0.42	0.97 \pm 0.25	0.90 \pm 0.22	1.33 \pm 0.33	0.50 \pm 0.08	0.0 - 7.5
Bas%	0.00 \pm 0.00	0.03 \pm 0.05	0.00 \pm 0.00	0.00 \pm 0.00	0.00 \pm 0.00	0.00 \pm 0.00	0.0 - 1.5
RBC ($\times 10^{12}/L$)	10.11 \pm 0.07	10.64 \pm 0.35	9.89 \pm 0.75	11.37 \pm 0.09	10.41 \pm 0.86	11.15 \pm 0.52	6.50 - 11.50
HGB (g/L)	161.67 \pm 2.62	161.33 \pm 0.47	162.67 \pm 3.40	155.33 \pm 3.86	168.00 \pm 6.48	147.67 \pm 4.11	110 - 165
HCT (%)	45.67 \pm 0.46	47.63 \pm 0.78	45.53 \pm 2.36	49.87 \pm 0.46	45.63 \pm 3.83	47.13 \pm 1.47	35.0 - 55.0
MCV (fL)	45.17 \pm 0.69	44.83 \pm 0.90	44.63 \pm 0.97	43.87 \pm 0.60	43.87 \pm 0.52	42.33 \pm 1.03	41.0 - 55.0
MCH (pg)	16.00 \pm 0.36	15.87 \pm 0.37	15.53 \pm 0.41	15.43 \pm 0.31	15.53 \pm 0.21	15.10 \pm 0.37	13.0 - 18.0
MCHC (g/L)	353.67 \pm 5.25	353.67 \pm 0.94	348.33 \pm 2.62	352.00 \pm 6.16	353.67 \pm 2.05	356.00 \pm 2.83	300 - 360
RDW-CV (%)	15.10 \pm 0.36	15.27 \pm 0.17	16.17 \pm 1.31	15.23 \pm 0.42	15.93 \pm 0.83	16.40 \pm 0.62	12.0 - 19.0
RDW-SD (%)	30.70 \pm 0.37	30.53 \pm 0.71	32.27 \pm 1.96	29.93 \pm 1.36	31.23 \pm 1.27	31.37 \pm 0.45	23.0 - 39.0
PLT ($\times 10^{11}/L$)	10.49 \pm 0.36	10.88 \pm 0.80	10.68 \pm 1.49	10.10 \pm 0.53	10.88 \pm 1.49	9.11 \pm 1.04	4.00 - 16.00
MPV (fL)	6.20 \pm 0.16	6.00 \pm 0.08	6.03 \pm 0.25	6.07 \pm 0.12	5.93 \pm 0.12	5.97 \pm 0.17	4.0 - 6.2
PDW (fL)	16.63 \pm 0.12	16.33 \pm 0.09	16.30 \pm 0.22	16.40 \pm 0.22	16.33 \pm 0.19	16.33 \pm 0.26	12.0 - 17.5
PCT (%)	0.65 \pm 0.01	0.65 \pm 0.04	0.64 \pm 0.07	0.64 \pm 0.01	0.65 \pm 0.10	0.54 \pm 0.07	0.100 - 0.780

WBC, White blood cell counts; Neu, Neutrophils; Lym, Lymphocytes; Mon, Monocytes; Eos, Eosinophils; Bas, Basophils; RBC, Red blood cell counts; HGB, Hemoglobin; HCT, Hematocrit; MCV, Mean corpuscular volume; MCH, Mean corpuscular hemoglobin; MCHC, Mean corpuscular hemoglobin concentration; RDW-CV, Red cell distribution width-coefficient variation; RDW-SD, Red cell distribution width-standard deviation; PLT, Platelets; MPV, Mean platelet volume; PDW, Platelet distribution width; PCT, Plateletcrit. Data are mean \pm S.D. (n = 3).

Supplementary Figure 38. Developmental toxicity and teratogenicity in zebrafish embryos. **a** Images of zebrafish embryos after injection with various lipoplexes at 48, 72, and 96 hours post-fertilization (hpf). **b** Survival rate and **c** hatching rate of zebrafish embryos at 24, 48, 72, and 96 hpf. **d** The spontaneous contraction of the embryos examined at the 18-somite stage. **e** Total coiling contractions within 1 min counted at 24 hpf.

Supplementary table 3. Malformation rate induced by various lipoplexes in zebrafish larvae.

Name	Malformation of tail	Scoliosis	Pericardial edema
Blank	0	1	1
RLS	1	0	0
RLS/HS@Au	0	0	0

Supplementary Figure 39. Distribution of RLS/HS@Au in different organ tissues of Balb/c mice at 15, 31, and 61 days, expressed as the percentage of gold mass in the total injected dose. * $p < 0.05$, $n = 3$.

Q2: Additionally, elucidating the detailed molecular mechanisms underlying the enhancement of autophagic flux and RhoA activation due to nanoparticle interactions could further strengthen the understanding of this innovative gene delivery platform. At least the authors should discuss the possible detailed mechanisms and show the most critical evidence/s.

A2: Thanks for your kind suggestion. To elucidate the mechanism more clearly, we have conducted more experiments to strengthen the understanding of this innovative gene delivery platform. The supplemental experiments included transfection of the LC3-EGFP-mCherry reporter plasmid (Fig. 4e, f and Supplementary Fig. 19), localization of RhoA-GTP (Fig. 4b), more TEM observations of autophagic flux (Supplementary Fig. 20, 21), *in vivo* detection of the RhoA-GTP expression and F-actin polymerization (Fig. 8h, i). Combined with that, we have added more possible detailed mechanism discussion and showed the most critical evidence as follows:

- 1) The membrane distribution of the RhoA-GTP has been confirmed, as important proof to clarify the “inside-out” biochemical (autophagy/RhoA) signaling-induced cytoskeleton rearrangement, which would also decide the dynamic polymerization of F-actin¹⁻⁴. From the supplemented results (Fig. 4b), we could conclude that the membrane distribution of the RhoA-GTP increased after RLS/HS@Au treatment.
- 2) The accumulation of RhoA-GTP was induced by the inhibition of autophagic flux by RLS/HS@Au. Considering the activated RhoA-GTP inside cells was often recycled by the autophagosome formed at the initial stage of autophagic flux⁵, the inhibition of autophagy initiation was confirmed by LC3-EGFP-mCherry reporter plasmid transfection, TEM observation, and western blot detected LC3-I/II expression (Fig. 4e-h) after RLS/HS@Au treatment in our supplemented experiments.

Interestingly, we also found that RLS/HS@Au compelled more membrane-associated RhoA-GTP than HS@Au groups (Fig. 4b). It was speculated that RLS might help more GTP-RhoA escape from lysosome because it can interfere with membrane fluidity and penetrate membrane as we previously reported⁶. The AuNPs accumulation in lysosomes was verified to induce lysosomal pH alkalinization (Fig. 4 i), which impairs lysosome degradation capacity⁷. These destabilizations of lysosomes by AuNPs and the transmembrane function of RLS might together provide an opportunity for GTP-RhoA to escape from autolysosomes without being degraded. Besides, the increased GTP-RhoA accumulation and F-actin polymerization were also observed in the *in vivo* tumor tissues (Fig. 8f, g).

Thus, we concluded that RLS/HS@Au may increase the membrane distribution of GTP-RhoA in two

ways: ① blocking the autophagy at the initial stage which is responsible for GTP-RhoA sequestration; ② aiding the escape of sequestered GTP-RhoA from autolysosomes through pH alkalization and interference with lysosomal membrane fluidity (Fig. 4j).

Fig. 4 Mechanism exploration on cytoskeletal remodeling by GTP-RhoA accumulation. **a** The pull-down assay of GTP-RhoA accumulation in HepG2 cells using RHOTEKIN binding. Cells were added with different lipoplexes (1 $\mu\text{g}/\text{mL}$ DNA) or free HS@Au (50 μg Au/mL) for 24 h incubation. Bound proteins (top) and total cell lysates (middle) were analyzed by western blotting. The β -actin was used as a loading control, while GTP γ S was the positive control with high affinity to RHOTEKIN. **b** Immunofluorescence staining of RhoA in HepG2 cells fixed with 10% ice-cold TCA after treatment with various lipoplexes (1 μg pDNA/well) or free HS@Au (50 $\mu\text{g}/\text{mL}$) for 24 h. Green: RhoA, Blue: DAPI. The scale bar is 10 μm . **c** EGFP fluorescence microscopy images and **d** quantitative analysis by flow cytometry of HepG2 cells transfected with different lipoplexes in the absence or presence of C3 transferase (RhoA inhibitor, 1 $\mu\text{g}/\text{mL}$) at 48 h. **e** Confocal images of HepG2 cells expressing LC3-EGFP-mCherry after treatment with various gene lipoplexes for 24 and 48 h. In the starvation conditions, HepG2 cells were cultured in EBSS buffer for 5 h after lipoplexes treatment. Green tunnel: LC3-EGFP; Red tunnel: LC3-mCherry; Yellow tunnel: the merge of EGFP and mCherry, Blue: DAPI. The scale bar is 10 μm . **f** Mean number of yellow puncta and red puncta. **g** TEM observation of autophagic flux induced by RLS/HS@Au at DNA of 1 $\mu\text{g}/\text{mL}$ or free HS@Au (50 μg Au/mL) on HepG2 cells. The red, white, and yellow arrows indicated endosomes containing Au nanoparticles, autophagosomes, and autolysosomes, respectively. The scale bars are 1 μm and 500 nm, respectively. **h** Western blot analysis of autophagy-related proteins in HepG2 cells treated with various lipoplexes (at DNA of 1 $\mu\text{g}/\text{mL}$) in the absence or presence of autophagy flux inhibitor chloroquine (CQ, 10 $\mu\text{mol}/\text{L}$) for 24 h. The β -actin was used as a loading control. **i** Lysosomal proteolytic activity analysis of HepG2 cells treated with various lipoplexes (with 3 $\mu\text{g}/\text{mL}$ DNA) or HS@Au (at 50 μg Au/mL) for 24 h. The activity was evaluated by the red fluorescence recovery of derivative-quenched bovine serum albumin (DQ-BSA, 10 $\mu\text{g}/\text{mL}$) in HepG2 cells. Serum-free MEM was used as a positive control since starvation could increase the lysosomal proteolytic activity and cause bright red fluorescence. **j** Illustration of the GTP-RhoA accumulation due to autophagic flux suppression by the HS@Au.

Fig. 8 **f** Pull-down assay of GTP-RhoA in HepG2 tumor-bearing mice treated with tail vein injection of RLS/HS@Au/pDNA (pDNA 20 $\mu\text{g}/\text{mouse}$). The β -actin was used as a loading control, and GTP γ S was the positive control. **g** Immunofluorescence staining of F-actin (green) in HepG2 tumors sections. The scale bar is 20 μm .

1. Begum R, Nur EKMS, Zaman MA. The role of Rho GTPases in the regulation of the rearrangement of actin cytoskeleton and cell movement. *Exp Mol Med* 36, 358-366 (2004).
2. Hall A. Rho GTPases and the actin cytoskeleton. *Science* 279, 509-514 (1998).
3. Symons M, Rusk N. Control of vesicular trafficking by Rho GTPases. *Curr Biol* 13, R409-418 (2003).
4. Chen Y, et al. Cullin mediates degradation of RhoA through evolutionarily conserved BTB adaptors to control actin cytoskeleton structure and cell movement. *Mol Cell* 35, 841-855 (2009).
5. Belaid A, et al. Autophagy plays a critical role in the degradation of active RHOA, the control of cell cytokinesis, and genomic stability. *Cancer Res* 73, 4311-4322 (2013).
6. Wei Y, et al. A cationic lipid with advanced membrane fusion performance for pDNA and mRNA delivery. *J*

Mater Chem B 11, 2095-2107 (2023).

7. Ma X, *et al.* Gold nanoparticles induce autophagosome accumulation through size-dependent nanoparticle uptake and lysosome impairment. *ACS Nano* 5, 8629-8639 (2011).

Q3: The study provides data on the immediate cytotoxic effects and efficacy but what is the long-term stability of these nanoparticles and their long-term toxicological effects?

A3: Thank you for this question. According to your suggestion, we have supplemented an assessment of long-term stability of the hybrid nanoparticles (one month in aqueous solution, Supplementary Fig. 9), as well as a long-term biodistribution of the RLS/HS@Au system by ICP-AES analysis of the gold content in various organ tissues *in vivo* (Supplementary Fig. 39). Meanwhile, the *in vivo* long-term systemic toxicity and genotoxicity tests (Supplementary Fig. 37 and Supplementary table 2) in the Balb/c mice have been performed according to the suggestion from the International Organization for Standardization (ISO)-10993 guidance (Biological Evaluation of Medical Devices).

Results showed that the size of the RLS/HS@Au was stable in one month in an aqueous solution (Supplementary Fig. 9). The corresponding revision has been made in the main text (including results and method, page 6, line 115; page 42, line 713) and Supplementary Information with bright yellow mark. In the long-term systemic toxicity and genotoxicity tests, the hybrid nanoparticles were non-toxic (Supplementary Fig. 37), and the component Au was detained a lot in the liver and spleen (Supplementary Fig. 39).

The components of our designed nanoparticles are hyaluronic acid-modified Au nanoparticles (AuNPs) and lipopeptide RLS. The choice of each part has considered their potential biocompatibility. The AuNPs were one of the most widely used inorganic particles in immunology, medicine, and biotechnology applications¹⁻³. Its biosafety has been acceptable for many clinical trials containing Au nanoparticles. The first clinical trial related to AuNPs was recombinant human tumor necrosis factor (rhTNF) bound to colloidal gold, which was conducted in the year 2006 and named CYT-6091. Followingly, tumor necrosis factor- α conjugated and polyethylene glycol (PEG) decorated AuNPs entered into clinical trials of early-phase (NCT00436410) and phase I (NCT00356980). Recently, silica-coated gold nanoparticles (Aurolase) have been investigated in the clinical trials for the photothermal ablation treatment of refractory and/or recurrent tumors of the head and neck (NCT00848042) and prostate cancer (NCT02680535). Moreover, a gold base spherical nucleic acid (SNA) nanoconjugate targeting BCL2L12 (named: NU-0129) is currently in clinical trials for the treatment of patients with recurrent glioblastoma and gliosarcoma undergoing surgery (NCT03020017). These trials didn't result in any adverse events.

In terms of the toxicity of RLS, when we designed the chemical composition of RLS^{4,5}, we took into account its potential toxicity in clinical application. Thus, to minimize the toxicity, all components including the hydrophilic, hydrophobic, and linking parts are the endogenous amino acids and degradable small molecules (such as arginine, lysine, and oleic acid). They are also conjugated by biodegradable peptide bonds. All of this ahead of consideration ensures the safety and low toxicity of the material to a large extent.

Supplementary Figure 9. Size distribution of RLS/HS@Au in one month. Data are mean \pm S.D.

1. Mohammadpour R, Dobrovolskaia MA, Cheney DL, Greish KF, Ghandehari H. Subchronic and chronic toxicity evaluation of inorganic nanoparticles for delivery applications. *Adv Drug Deliv Rev* **144**, 112-132 (2019).
2. Zhu S, Li L, Gu Z, Chen C, Zhao Y. 15 Years of Small: Research Trends in Nanosafety. *Small* **16**, e2000980 (2020).
3. Kesharwani P, *et al.* Gold nanoparticles and gold nanorods in the landscape of cancer therapy. *Mol Cancer* **22**, 98 (2023).
4. Liang H, *et al.* Structure optimization of dendritic lipopeptide based gene vectors with the assistance from molecular dynamic simulation. *J Mater Chem B* **7**, 915-926 (2019).
5. Chen X, Yang J, Liang H, Jiang Q, Ke B, Nie Y. Disulfide modified self-assembly of lipopeptides with arginine-rich periphery achieve excellent gene transfection efficiency at relatively low nitrogen to phosphorus ratios. *J Mater Chem B* **5**, 1482-1497 (2017).

Minor issues:

Q4: How this new approach compares with current existing gene delivery systems, for example in terms of efficiency, safety, and feasibility?

A4: In gene therapy, a safe and efficient gene delivery system is critical to ensure therapeutic efficacy. The gene delivery system we designed aims to demonstrate a proof-of-concept for the design of efficient gene delivery systems that could manipulate the cytoskeleton through co-regulation of mechanical and biochemical signals, by mimicking behaviors of viruses.

In terms of efficiency, among the existing gene delivery systems, viral vectors are highly efficient but their immunogenicity and cytotoxicity limit their application. Non-viral vectors such as lipid nanoparticles (LNP) show great advantages over viruses in gene loading capacity, fabrication, and scale-up. However, their delivery efficiency still requires further improvement, especially for DNA.

Our gene delivery system has high transfection efficiency and a more simplified component, employing inorganic/organic hybrid lipid nanoparticles that combine the hardness and structural stability of inorganic materials with the biocompatibility of organic materials¹. Such carriers can efficiently encapsulate genes and optimize cellular uptake and transport capacity, resulting in a significant increase in gene transfection efficiency.

In terms of biosafety, each part of the hybrid lipoplexes (hyaluronic acid-modified AuNPs and lipopeptide RLS) is biocompatible^{2,3,4}. Results of the biosafety assessment also confirmed their no long-term toxicity. The mice treated with RLS/HS@Au did not show significant weight loss, major organ abnormalities, or obvious liver dysfunction. In addition, no toxic reactions were observed at high doses (details in Reviewer 3, answer to Q1).

In summary, inorganic/organic hybrid lipid nanoparticles have shown potential as a promising approach to improve the efficiency of gene delivery. By combining the advantages of both inorganic and organic materials, such vectors are able to improve gene transfection efficiency and offer great potential and feasibility for the development of gene therapy and other genetic medicine approaches. However, further studies and evaluations are still needed to ensure the safety and efficacy of these vectors before they can be applied to clinical treatments.

1. Seaberg J, Montazerian H, Hossen MN, Bhattacharya R, Khademhosseini A, Mukherjee P. Hybrid Nanosystems for Biomedical Applications. *ACS Nano* 15, 2099-2142 (2021).
2. Liang H, *et al.* Structure optimization of dendritic lipopeptide based gene vectors with the assistance from molecular dynamic simulation. *J Mater Chem B* 7, 915-926 (2019).
3. Chen X, Yang J, Liang H, Jiang Q, Ke B, Nie Y. Disulfide modified self-assembly of lipopeptides with arginine-rich periphery achieve excellent gene transfection efficiency at relatively low nitrogen to phosphorus ratios. *J Mater Chem B* 5, 1482-1497 (2017).
4. He Y, Cheng G, Xie L, Nie Y, He B, Gu Z. Polyethyleneimine/DNA polyplexes with reduction-sensitive hyaluronic acid derivatives shielding for targeted gene delivery. *Biomaterials* 34, 1235-1245 (2013).

Q5: In the abstract, the authors claimed that the hybrid lipoplex led to 20-fold higher gene expression with appr. 80% tissue specificity *in vivo*. How did you quantify the numbers?

A5: Thanks for the question. The hybrid lipoplex (RLS/HS@Au/pGL3) group demonstrated a 20-fold increase in gene expression at the tumor site in mice, as determined by dividing the average radiance value of the RLS/HS@Au group by that of the RLS/HS group at the tumor site. Tissue specificity *in vivo* was quantified by calculating the percentage of gene expression at the tumor site relative to the total average radiance value of all organ tissues (including the tumor) in the RLS/HS@Au/pGL3 group, resulting in a value of 80% for tissue specificity *in vivo*.

Q6: Fig. 2g, it seems almost all the Cy5-DNA are localized in the cytosol but not the nucleus.

A6: Thanks for the comment. Fig. 2g was the image of the cells (with F-actin staining) after incubation with various lipoplexes containing Cy5-DNA for 4 h. You are right that, most lipoplex has not yet reached the stage of nucleus entry at that time. The nuclear distribution of Cy5-DNA could be captured at 6 h (Fig. 3h, details in Reviewer 3, answer to Q8).

Moreover, in order to see Fig. 2g more clearly, we performed a split of the fluorescence channels of the RLS/HS@Au group (Blue: nucleus, Red: Cy5-DNA, Green: F-actin) as shown in Fig. R8. We could see there is only a small amount of Cy5-DNA located in the nucleus (white arrow) in the RLS/HS@Au group.

Fig. 2g Illustration and intuitive observation of the lipoplexes distributed along F-actin in well-spread cells after 4 h transfection. (Blue: DAPI stained nucleus, Green: FITC-phalloidin stained F-actin, Red: Cy5-labeled DNA). The scale bar is 5 μm .

Fig. R8 Observation of the RLS/HS@Au/Cy5-DNA distributed along F-actin in well-spread cells after 4 h transfection. The white arrows represent Cy5-DNA in the nucleus. (Blue: DAPI stained nucleus, Green: FITC-phalloidin stained F-actin, Red: Cy5-labeled DNA). The scale bar is 5 μm .

Q7: Fig. 3e, why the endo/lysosome intensity in RLS/HS and RLS groups are much lower than that in RLS/HS@Au group?

A7: Thanks for the question. The weak endo/lysosome fluorescence signals in the RLS/HS and RLS groups in Fig. 3e might be because the amount of cellular endocytosis of these two lipoplexes was less than that of RLS/HS@Au, resulting in a lower endo/lysosome fluorescence signal. This result was similar to other reported literature^{1,2}.

1. Ren J, *et al.* Enzyme-powered nanomotors with enhanced cell uptake and lysosomal escape for combined therapy of cancer. *Applied Materials Today* **27**, (2022).
2. Zheng J, *et al.* Bifunctional Compounds as Molecular Degradators for Integrin-Facilitated Targeted Protein Degradation. *J Am Chem Soc* **144**, 21831-21836 (2022).

Q8: Fig. 3h, it is recommended to use Z-stack 3D fig to better demonstrate the Cy5-DNA intensity in the nucleus.

A8: Thanks for your suggestions. We have re-created a Z-stack 3D image using Imaris 9.0.1 software (Fig. 3h) to replace the previous Fig. 3h. The legend for Fig. 3h was described as follows:

“...Nuclei delivery evaluation for the RLS, RLS/HS, and RLS/HS@Au after 6 h transfections. 3D reconstruction of lipoplexes-labeled HepG2 cells using the Imaris 9.0.1 software. White arrows represent Cy5-DNA entering the nucleus. (Blue: Hoechst 33342 stained nucleus, Red: Cy5-labeled DNA) ...”(page 13, line 213).

Fig. 3h Nuclei delivery evaluation for the RLS, RLS/HS, and RLS/HS@Au after 6 h transfections. 3D

reconstruction of lipoplexes-labeled HepG2 cells using the Imaris 9.0.1 software. White arrows represent Cy5-DNA entering the nucleus. (Blue: Hoechst 33342 stained nucleus, Red: Cy5-labeled DNA).

REVIEWER COMMENTS

Reviewer #1:

The authors have adequately addressed the concerns raised by this reviewer.

Reviewer #2:

I am expressing my sincere appreciation for the authors' efforts in revising the manuscript. The new conclusions, particularly the recognition that RhoA-GTP accumulation may be due to a block in initiation rather than a block in maturation, are truly impressive.

However, I do have some questions regarding the results:

* Cell Migration and RLS/HS@Au (Fig. 2h, i, lanes 132-133). The increase in cell migration by 58% due to RLS/HS@Au is intriguing. Since RhoA-GTP accumulation typically inhibits migration and increases cell adhesion to the support via the reticulation of stress fibers, further discussion or clarification from the authors would be valuable.

* Yellow LC3 GFP-mCherry Puncta (Fig. 4e, lane 255). The authors claimed, "It is noteworthy that yellow puncta included both autophagosomes and alkalized autolysosomes. This confirms the interference of RLS/HS@Au in autophagosome formation, indicating that autophagic flux is compromised at the initial stage." To confirm the presence of alkalized autolysosomes, I suggest the authors perform immunofluorescence with the lysosomal marker LAMP1 in the LC3 GFP-mCherry transfected cells (the autolysosomes will be LAMP1 positive and LC3 GFP mCherry positive). Instead, I interpret this differently; the yellow puncta may correspond to LC3-I and p62 aggregates/aggregosomes, consistent with a blockage of autophagy initiation.

* CQ and NH₄Cl as "autophagy inhibitors" (lane 263, Fig. 4h, Supplementary Fig. 22, 23). CQ and NH₄Cl primarily inhibit autophagy maturation (as evidenced by the higher accumulation of LC3-II and p62 in lane 5), not initiation. Adding RLS/HS@Au decreased LC3-II levels (lane 8), suggesting that RLS/HS@Au inhibited autophagy flux at the initiation step. The text should be revised accordingly to avoid confusion.

* Editing Text on Autophagy Maturation and Autolysosome throughout the manuscript:

The authors should acknowledge that the same compound cannot simultaneously block both autophagy initiation and maturation stages. Clarification from the authors regarding this apparent contradiction would be valuable. Changing the text to focus on autophagy initiation and phagocytosis (lane 271, DQ-BSA tracks lysosomal endocytosis and not autolysosomal degradation) aligns better with the observed effects of RLS/HS@Au.

Along this line, I observed that several structures highlighted by the yellow arrows (Figure 20) in the electron microscopy slides lack the typical double membrane (of autophagosome) or membrane delimitation (of autolysosome). Instead, they resemble dense aggregates or even aggresomes, commonly associated with a blockage at the initiation stage of autophagy.

This discrepancy warrants further investigation and consideration in the context of the study. Perhaps additional analyses could shed light on the precise nature of these structures.

Reviewer #3:

I do not have other concern.

Comments and point-to-point answers:

Reviewer #1:

The authors have adequately addressed the concerns raised by this reviewer.

> Thank you very much for your support of our work.

Reviewer #2:

I am expressing my sincere appreciation for the authors' efforts in revising the manuscript. The new conclusions, particularly the recognition that RhoA-GTP accumulation may be due to a block in initiation rather than a block in maturation, are truly impressive. However, I do have some questions regarding the results:

> Thank you very much for your positive and professional suggestion regarding the autophagy-related research.

According to the common definition of the autophagy process, it can be summarized as five continuous events: initiation/nucleation, expansion, closure, fusion, and cargo degradation^{1,2}. The newly formed autophagosomes undergo 'maturation', namely the autophagosome fuses with the endosome to form an amphisome, and subsequently fuses with the lysosome to be a degradative autolysosome, or the autophagosome fuses directly with the lysosome to form a degradative autolysosome.

Thus, according to your suggestion, we have corrected and refined the description in the revision as: "RLS/HS@Au impairs early autophagosome formation and late lysosome proteolysis during the autophagic flux."

Besides, we have detected and elaborated on the induction of alkalized autolysosomes by gold nanoparticles (Fig. 4f).

1. Zhao YG, Codogno P, Zhang H. Machinery, regulation and pathophysiological implications of autophagosome maturation. *Nat Rev Mol Cell Biol* **22**, 733-750 (2021).
2. Klionsky DJ, *et al.* Guidelines for the use and interpretation of assays for monitoring autophagy (4th edition)(1). *Autophagy* **17**, 1-382 (2021).

Q1* Cell Migration and RLS/HS@Au (Fig. 2h, i, lanes 132-133). The increase in cell migration by 58% due to RLS/HS@Au is intriguing. Since RhoA-GTP accumulation typically inhibits migration and increases cell adhesion to the support via the reticulation of stress fibers, further discussion or clarification from the authors would be valuable.

A1. Thanks for your kind suggestion. First of all, we fully agree that the GTP-RhoA accumulation is closely related to cell migration via promoting actin polymerization. However, the cell migration behaviors are decided by the dynamic changes in the actin cytoskeleton which work in concert to facilitate the continuous cycles of cell protrusion, adhesion, and contraction. RhoA, acting as molecular switches, demonstrates remarkable versatility throughout the whole process. How the function of RhoA exactly switches between contraction and protrusion of the stress fibers has remained unclear, leading to an unpredictable migration behavior.

As an active state of RhoA, GTP-RhoA affects its downstream Rho kinase (ROCK). When the ROCK phosphorylates, it activates LIM kinase. Subsequent, LIM kinase phosphorylates and inhibits the actin-

severing protein cofilin, thereby increasing the stability of actin¹. Additionally, GTP-RhoA could promote actin filament polymerization by interacting with another effector, mDia1, which is responsible for actin nucleating¹.

Many studies have demonstrated a positive role for RhoA in migration, with the concept that Rac (another well-studied Rho GTPases) and RhoA were spatially separated during cell migration such that Rac was activated at the leading edge and RhoA was activated at the trailing edge². In addition, RhoA is also found to be activated at the leading edge of cellular migration, preceding that of Rac and Cdc42 (another well-studied Rho GTPases), which suggested that RhoA is implicated in membrane ruffling and lamellae formation, playing an important role in the protrusive events at the leading edge that drive cell motility³. However, how the RhoA switches from stress fibers to lamellae formation is still unclear now. As an example, Prof. Yiyao Liu's group⁴ found that "RhoA, ROCK1, and ROCK2 signaling was positively connected to breast tumor cell motility in a substrate stiffness-dependent manner." In typical migrating cells, such as triple-negative breast cancer cell line, MDA-MB-231, active RhoA binds to ROCK, mediating the formation of actin stress fibers, with the generation of the contractile force required for cell tail retraction and focal adhesion turnover. They observed that the ROCK isoforms differentially regulated the pathways of RhoA/ROCK1/p-MLC and RhoA/ROCK2/p-cofilin in a coordinate fashion to promote breast cancer cell motility in a substrate stiffness-dependent manner through integrin β 1-activated FAK signaling.

Some other reports agreed with your opinion that RhoA-GTP accumulation inhibits migration and increases cell adhesion to the support via the reticulation of stress fibers. As described in the reference⁵ you mentioned in the first peer review comments, alterations in the autophagy pathway can have a profound impact on cell motility: inhibition of autophagy degradation would promote rapid cell migration through the increased RhoA sequestration (TCIRG1 loss), whereas blocking autophagosome formation or sequestration would have the opposite effects (ATG5, ATG7, SQSTM1 shRNA). Detailly, autophagy inhibition active RhoA at the plasma membrane allowed actin polymerization into filaments and impaired the formation of cell protrusions. Meanwhile, they found that the accumulation of RhoA at the A549 tumor cell surface induced higher levels of downstream phosphorylation of myosin regulatory light chain (P-MLC), a denser ACTIN network, and consistently suppressed motility. They proposed that autophagy might be necessary for maintaining the appropriate amount of RhoA at the lamellipodia to allow cell motility.

Thus, the relationship between RhoA activation and cell migration is decided by varied factors, such as the appropriate amount, downstream effectors, and space-time distribution. In our case, the RhoA activation promoted the polymerization of stress fibers and enhanced cell migration. However, the detailed mechanism and signals involved were unclear, which would be focused on in future studies. In the revision, the related discussion was added to the manuscript with yellow highlight (page 34, line 551).

1. Lessey EC, Guilluy C, Burridge K. From mechanical force to RhoA activation. *Biochemistry* **51**, 7420-7432 (2012).
2. O'Connor K, Chen M. Dynamic functions of RhoA in tumor cell migration and invasion. *Small GTPases* **4**, 141-147 (2013).
3. O'Connor KL, Chen M, Towers LN. Integrin α 6 β 4 cooperates with LPA signaling to stimulate Rac through AKAP-Lbc-mediated RhoA activation. *Am J Physiol Cell Physiol* **302**, C605-614 (2012).
4. Peng Y, *et al.* ROCK isoforms differentially modulate cancer cell motility by mechanosensing the substrate stiffness. *Acta Biomater* **88**, 86-101 (2019).
5. Belaid A, *et al.* Autophagy and SQSTM1 on the RHOA(d) again: emerging roles of autophagy in the degradation of signaling proteins. *Autophagy* **10**, 201-208 (2014).

Q2* Yellow LC3-GFP-mCherry Puncta (Fig. 4e, lane 255). The authors claimed, "It is noteworthy that yellow puncta included both autophagosomes and alkalized autolysosomes. This confirms the

interference of RLS/HS@Au in autophagosome formation, indicating that autophagic flux is compromised at the initial stage." To confirm the presence of alkalinized autolysosomes, I suggest the authors perform immunofluorescence with the lysosomal marker LAMP1 in the LC3-GFP-mCherry transfected cells (the autolysosomes will be LAMP1 positive and LC3-GFP-mCherry positive). Instead, I interpret this differently; the yellow puncta may correspond to LC3-I and p62 aggregates/aggresomes, consistent with a blockage of autophagy initiation.

A2. Thank you for the detailed guide on the confirmation of alkalinized autolysosomes. We have performed the immunofluorescence of lysosomal marker LAMP1 after the cellular transfection of LC3-EGFP-mCherry reporter plasmid treated with various lipoplexes (Fig.4f). Many bright white puncta were observed in the hybridization group (RLS/HS@Au), resulting from the colocalization of the LAMP1-positive (blue fluorescence) and LC3-GFP-mCherry positive (yellow fluorescence) vesicles. In contrast, the yellow puncta in the RLS group rarely colocalized with the blue LAMP1 signal.

These findings suggest that the yellow puncta included alkalinized autolysosomes in RLS/HS@Au-treated cells, whereas they primarily indicate autophagosomes in RLS-treated cells.

Fig. 4f Detection of alkalinized autolysosomes. Confocal images of HepG2 cells transfected with various gene lipoplexes containing LC3-EGFP-mCherry plasmid for 24 h and 48 h followed by LAMP1 immunofluorescence staining. HepG2 cells were cultured in EBSS buffer for 5 h after lipoplexes treatment for starvation. Green tunnel: LC3-EGFP; Red tunnel: LC3-mCherry; Blue: LAMP1, Violet: Nuclei. The scale bar is 5 μm.

Besides, the phenomena were consistent with other previous studies that the accumulation of AuNPs in lysosomes could induce lysosome alkalinization (Fig.R1, inhibits lysosomal degradation activity), swelling,

and membrane permeabilization, and result in damage to autophagic flux. These findings confirmed the interference of lysosomes by Au nanoparticles^{1,2}.

(1) Prof. Liang's group tested whether AuNPs can affect lysosome pH. In their experiment, "the cells were labeled with LysoSensor Green DND-189 dye to compare the lysosome acidity. Flow cytometry analysis showed size-dependent alkalinization of lysosomes in AuNP-treated cells" (**Fig. R1A**). "Fluorescent microscopic analysis confirmed this effect" (**Fig. R1B**).

(2) Prof. Gao's group utilized the mRFP-GFP-LC3 assay and analyzed the autophagy-related proteins (LC3, beclin1, and p62) to conclude that "the PEG-AuNPs induced autophagic flux inhibition in tumor-associated macrophages" (**Fig. R2**), attributing from "the PEG-AuNPs induced lysosome alkalinization (**Fig. R2D**) and membrane permeabilization (**Fig. R2E**)".

Thus, we concluded that "the yellow puncta included both autophagosomes and alkalinized autolysosomes".

Fig. R1¹. Effect of AuNPs on lysosome pH. (A) FACS analysis of cells stained with LysoSensor Green DND-189. The effect of 10, 25, and 50 nm AuNP treatment on lysosome pH was analyzed 24 h after treatment. Black line, control cells; green line, AuNP-treated cells. (B) Representative fluorescent pictures of NRK cells treated with AuNPs for 24 h, then exposed for 30 min to 1 μ mol/L LysoSensor Green DND-189 (scale bar, 50 μ m).

Fig. R2 PEG-AuNPs block autophagic flux and cause lysosomal dysfunction in macrophages. (A) Western blotting analysis of autophagy-related proteins. (B) The images of representative immunofluorescence and quantitative analysis of the number of yellow autophagosomes (G⁺R⁺) and red autolysosomes (G⁻R⁺) of mRFP-GFP-LC3 dots in macrophages. Scale bar = 10 μm. (C) Fluorescence images of LysoSensor Green DND-189 stained macrophages. Scale bar = 50 μm. (D) Flow cytometry analysis of LysoSensor Green DND-189 stained macrophages. (E) Flow cytometry analysis of macrophages stained with acridine orange (AO). Data are expressed as mean ± SD (n = 5). *P < 0.05, **P < 0.01, ***P < 0.001

1. Ma X, *et al.* Gold nanoparticles induce autophagosome accumulation through size-dependent nanoparticle uptake and lysosome impairment. *ACS Nano* **5**, 8629-8639 (2011).
2. Zhang S, *et al.* Gold nanoparticle-directed autophagy intervention for antitumor immunotherapy via inhibiting tumor-associated macrophage M2 polarization. *Acta Pharm Sin B* **12**, 3124-3138 (2022).

Q3* CQ and NH₄Cl as “autophagy inhibitors” (lane 263, Fig. 4h, Supplementary Fig. 22, 23). CQ and NH₄Cl primarily inhibit autophagy maturation (as evidenced by the higher accumulation of LC3-II and p62 in lane 5), not initiation. Adding RLS/HS@Au decreased LC3-II levels (lane 8), suggesting that RLS/HS@Au inhibited autophagy flux at the initiation step. The text should be revised accordingly to avoid confusion.

A3 We apologize for the unclear statement of the function of the CQ and NH₄Cl. In the revision, we clarified that “As the CQ and NH₄Cl primarily inhibit autolysosome degradation, it should induce the higher accumulation of LC3-II and p62 (Fig. 4h, lane 5). However, the LC3-II levels decreased (Fig. 4h, lane 8) after adding RLS/HS@Au in the presence of CQ or NH₄Cl. It suggested that the LC3-II generation was probably decreased initially, which means the reduced autophagosome formation via RLS/HS@Au treatment”.

Q4* Editing Text on Autophagy Maturation and Autolysosome throughout the manuscript:

The authors should acknowledge that the same compound cannot simultaneously block both autophagy initiation and maturation stages. Clarification from the authors regarding this apparent contradiction would be valuable. Changing the text to focus on autophagy initiation and phagocytosis (lane 271, DQ-BSA tracks lysosomal endocytosis and not autolysosomal degradation) aligns better with the observed effects of

RLS/HS@Au.

A4 Thank you very much for this comment. We apologize for our inappropriate terms and conclusion, which may exaggerate the findings. In the revision, we revised it as “the RLS/HS@Au impairs early autophagosome formation and late lysosome proteolysis during the autophagic flux”. This expression text aligns with previous studies by other researchers. They found the same phenomena that some nanoparticles or knockdown of certain intracellular protein (YTHDF3) could simultaneously impair both autophagosome formation and lysosomal degradation function^{1,2}.

The DQ-BSA has been extensively used as a fluorophore to indicate the proteolysis in lysosomes¹⁻³, resulting from the de-quenching and releasing of bright fluorescent fragments (**Fig. R3**). DQ-BSA is a fluorogenic substrate for proteases, whose fluorescence is quenched by its labeling with BODIPY dyes. Upon hydrolysis of the DQ-BSA to single peptides by proteases, the quenching is relieved, and thus, the fluorescent signal can be taken as a proxy for proteolytic activity of lysosome, but not the phagocytosis/lysosomal endocytosis.

We considered that, as the final step in the autophagic flux, the autolysosome degradation was compromised when autophagosomes or amphisomes fused with impaired lysosomes.

Fig. R3¹ Impairment of lysosomes by AuNPs. (A) Vacuoles induced by AuNP treatment are enlarged lysosomes. NRK cells were incubated for 24 h with plain medium (control) or with 1 nM AuNPs. Inset: close-up of the enlarged lysosomes (scale bar, 10 μ m). (B) Lysosomal size was analyzed with Image pro-plus 6.0 software. The number of lysosomes analyzed was as follows: control, $n = 7437$; 10 nm, $n = 7726$; 25 nm, $n = 8548$; 50 nm, $n = 4564$. (C) TEM pictures of enlarged lysosomes after AuNP treatment (scale bar, 500 nm). (D) DQ-BSA analysis of lysosomal proteolytic activity. Accumulation of fluorescent signal, generated from lysosomal proteolysis of DQ-BSA, was much lower in AuNP-treated cells (scale bar, 10 μ m). (E) Fluorescence intensity of the brightly fluorescent fragments released by lysosomal degradation of DQ-BSA was quantified by densitometry (IOD). At least 40 cells were analyzed for each treatment. There was a statistically significant difference in lysosomal proteolytic activity between the control cells and cells treated with AuNPs. (F) Acid

phosphatase enzyme activity measurement of NRK cells treated with AuNPs for 24 h.

1. Ma X, *et al.* Gold nanoparticles induce autophagosome accumulation through size-dependent nanoparticle uptake and lysosome impairment. *ACS Nano* **5**, 8629-8639 (2011).
2. Hao W, *et al.* Autophagy induction promoted by m(6)A reader YTHDF3 through translation upregulation of FOXO3 mRNA. *Nat Commun* **13**, 5845 (2022).
3. Klionsky DJ, *et al.* Guidelines for the use and interpretation of assays for monitoring autophagy (4th edition)(1). *Autophagy* **17**, 1-382 (2021).

Q5* Along this line, I observed that several structures highlighted by the yellow arrows (Figure 20) in the electron microscopy slides lack the typical double membrane (of autophagosome) or membrane delimitation (of autolysosome). Instead, they resemble dense aggregates or even aggresomes, commonly associated with a blockage at the initiation stage of autophagy.

A5 Thanks for your kind suggestions. Thus, we investigated more references¹ containing the TEM observation of autophagy microstructure. As you said, the typical autophagosomes have double membranes or parallel double membranes visible at least in some areas and encapsulate cytoplasmic contents or organelles (Fig. R4). And the autolysosomes commonly have single-membrane structures and visible contents in different stages of degradation. These components often show amorphous areas of high electron density (Fig. R4).

Some of our previous selections of the autophagosome and autolysosome might not be so typical. It was possibly because the sample was tangentially sectioned, which results in the limiting membrane of the structures is not visible¹. In the revision, we reclassified the autophagy microstructure (only the autophagosomes and autolysosomes with typical structures are counted), showing the same trend as the results of the previous analysis (Supplementary Fig. 20, 21). The white arrow indicated autophagosome and the yellow arrow indicated the typical autolysosome after incubation with inorganic particles.

Autophagy is a highly conserved evolutionary process which would not incorporate gold nanoparticles during autophagosome formation, thus vesicles with typical double membranes and without Au nanoparticles were identified as autophagosomes. The single-membrane structure with different degradation stages of the contents (as shown by the amorphous region with high electron density) was identified as autolysosomes. We could also see that a fraction of the autolysosomes contained black and well-defined gold nanoparticles. According to reports², gold nanoparticles enter into endosomes or lysosomes through endocytosis, and subsequently fuse with autophagosomes, to eventually form autolysosomes.

Control

RLS/HS@Au

HS@Au

Supplementary Figure 20. TEM observation of autophagic flux induced by RLS/HS@Au at DNA of 1 $\mu\text{g}/\text{mL}$ or free HS@Au (50 $\mu\text{g Au}/\text{mL}$) on HepG2 cells. The white and yellow arrows indicated autophagosomes and autolysosomes, respectively. The scale bars are 1 μm and 500 nm, respectively.

Supplementary Figure 21. The average number of autophagosomes and autolysosomes calculated from 25 cells in each group.

Fig. R4¹ TEM images of autophagic vacuoles in isolated mouse hepatocytes. (A) One autophagosome or early autophagic vacuole (AVi) and one degradative autophagic vacuole (AVd) are shown. The AVi can be identified by its contents (morphologically intact cytoplasm, including ribosomes, and rough ER), and the limiting membrane that is partially visible as two bilayers separated by a narrow electron-lucent cleft, i. e., as a double membrane (arrow). The AVd can be identified by its contents, partially degraded, electron-dense rough ER. The vesicle next to the AVd is an endosomal/lysosomal structure containing 5-nm gold particles that were added to the culture medium to trace the endocytic pathway. (B) One AVi, containing rough ER and a mitochondrion, and one AVd, containing partially degraded rough ER, are shown. The AVd contains a region filled by small internal vesicles (asterisk), indicating that the AVd has fused with a multivesicular endosome. mi, mitochondrion.

1. Klionsky DJ, *et al.* Guidelines for the use and interpretation of assays for monitoring autophagy (4th edition)(1). *Autophagy* **17**, 1-382 (2021).
2. Zhou H, *et al.* Gold nanoparticles impair autophagy flux through shape-dependent endocytosis and lysosomal dysfunction. *J Mater Chem B* **6**, 8127-8136 (2018).

Q6* This discrepancy warrants further investigation and consideration in the context of the study. Perhaps additional analyses could shed light on the precise nature of these structures.

A6 We thank you for this kind suggestion. Our focus of the present study lies in the promotion of intracellular and extracellular gene delivery. We found that the hybrid nanoparticles could inhibit the autophagic flux.

We are also curious about the exact effect of nanoparticles on the process of autophagic flux, including the precise nature of these structures. However, using only one kind of material (RLS/HS@Au) is not enough for comparison. To confirm this is a common phenomenon, we have to find/synthesize/fabricate more kinds of NPs with different components, sizes, and shapes.

As we have provided in the last response, we have tried other hybrid NPs including oleic acid-modified cerium dioxide (OA@CeO₂), oleic acid-modified iron tetroxide (OA@Fe₃O₄), mesoporous silica (MSN), etc. It is another huge work in further investigation.

Reviewer #3:

I do not have other concern.

> We greatly appreciate your approbation of our work.

REVIEWERS' COMMENTS

Reviewer #2 (Remarks to the Author):

Excellent work

Thank you !